# A Nonconvex Approach for Exact and Efficient Multichannel Sparse Blind Deconvolution

**Qing Qu**
New York University
qq213@nyu.edu

**Xiao Li**
Chinese University of Hong Kong
xli@ee.cuhk.edu.hk

**Zhihui Zhu***
Johns Hopkins University
zzhu29@jhu.edu

## Abstract

We study the multi-channel sparse blind deconvolution (MCS-BD) problem, whose task is to simultaneously recover a kernel $\boldsymbol{a}$ and multiple sparse inputs $\{\boldsymbol{x}_i\}_{i=1}^p$ from their circulant convolution $\boldsymbol{y}_i = \boldsymbol{a} \circledast \boldsymbol{x}_i$ ($i = 1, \cdots, p$). We formulate the task as a nonconvex optimization problem over the sphere. Under mild statistical assumptions of the data, we prove that the vanilla Riemannian gradient descent (RGD) method, with random initializations, provably recovers both the kernel $\boldsymbol{a}$ and the signals $\{\boldsymbol{x}_i\}_{i=1}^p$ up to a signed shift ambiguity. In comparison with state-of-the-art results, our work shows significant improvements in terms of sample complexity and computational efficiency. Our theoretical results are corroborated by numerical experiments, which demonstrate superior performance of the proposed approach over the previous methods on both synthetic and real datasets.

## 1 Introduction

We study the blind deconvolution problem with multiple inputs: given *circulant* convolutions

$$\boldsymbol{y}_i = \boldsymbol{a} \circledast \boldsymbol{x}_i \in \mathbb{R}^n, \qquad i \in [p] := \{1, \ldots, p\}, \tag{1}$$

we aim to recover both the kernel $\boldsymbol{a} \in \mathbb{R}^n$ and the signals $\{\boldsymbol{x}_i\}_{i=1}^p \in \mathbb{R}^n$ using efficient methods. Blind deconvolution is an *ill-posed* problem in its most general form. Nonetheless, problems in practice often exhibits *intrinsic* low-dimensional structures, showing promises for efficient optimization. One such useful structure is the *sparsity* of the signals $\{\boldsymbol{x}_i\}_{i=1}^p$ [1]. The multichannel sparse blind deconvolution (MCS-BD) broadly appears in the context of communications [2, 3], computational imaging [4, 5], seismic imaging [6–8], neuroscience [9–13], computer vision [14–16], and more.

- **Neuroscience.** Detections of neuronal spiking activity is a prerequisite for understanding the mechanism of brain function. Calcium imaging [12,13] and functional MRI [9,11] are two widely used techniques, which record the convolution of unknown neuronal transient response and sparse spike trains. The spike detection problem can be naturally cast as a MCS-BD problem.

- **Computational (microscopy) imaging.** Super-resolution fluorescent microscopy imaging [4, 17, 18] conquers the resolution limit by solving sparse deconvolution problems. Its basic principle is using photoswitchable fluorophores that stochastically activate fluorescent molecular, creating a video sequence of sparse superpositions of point spread function (PSF). In many scenarios (especially in 3D imaging), as it is often difficult to obtain the PSF due to defocus and unknown aberrations [19], it is preferred to estimate the point-sources and PSF jointly by solving MCS-BD.

- **Image deblurring.** Sparse blind deconvolution problems also arise in natural image processing: when a blurry image is taken due to the resolution limit or malfunction of imaging procedure, it

Table 1: Comparison with existing methods for solving MCS-BD[2]

| Methods | Wang et al. [20] | Li et al. [21] | **Ours** |
|---|---|---|---|
| Assumptions | $\boldsymbol{a}$ spiky & invertible, $\boldsymbol{x}_i \sim_{i.i.d.} \mathcal{BG}(\theta)$ | $\boldsymbol{a}$ invertible, $\boldsymbol{x}_i \sim_{i.i.d.} \mathcal{BR}(\theta)$ | $\boldsymbol{a}$ invertible, $\boldsymbol{x}_i \sim_{i.i.d.} \mathcal{BG}(\theta)$ |
| Formulation | $\min_{\|\boldsymbol{q}\|_\infty = 1} \|\boldsymbol{C_q Y}\|_1$ | $\max_{\boldsymbol{q} \in \mathbb{S}^{n-1}} \|\boldsymbol{C_q PY}\|_4^4$ | $\min_{\boldsymbol{q} \in \mathbb{S}^{n-1}} H_\mu \left( \boldsymbol{C_q PY} \right)$ |
| Algorithm | interior point | *noisy* RGD | *vanilla* RGD |
| Recovery Condition | $\theta \in \mathcal{O}(1/\sqrt{n})$, $p \geqslant \widetilde{\Omega}(n)$ | $\theta \in \mathcal{O}(1)$, $p \geqslant \widetilde{\Omega}(\max\left\{n, \kappa^8\right\} \frac{n^8}{\varepsilon^8})$ | $\theta \in \mathcal{O}(1)$, $p \geqslant \widetilde{\Omega}(\max\left\{n, \frac{\kappa^8}{\mu^2}\right\} n^4)$ |
| Time Complexity | $\widetilde{\mathcal{O}}(p^4 n^5 \log(1/\varepsilon))$ | $\widetilde{\mathcal{O}}(pn^{13}/\varepsilon^8)$ | $\widetilde{\mathcal{O}}(pn^5 + pn \log\left(1/\varepsilon\right))$ |

can be modeled as a blur pattern convolved with visually plausible sharp images (whose gradient are sparse) [15, 16].

**Prior arts.** Recently, there have been a few attempts to solve MCS-BD with guaranteed performance. Wang et al. [20] formulated the task as finding the sparsest vector in a subspace problem [22]. They considered a convex objective, showing that the problem can be solved to exact solutions when $p \geqslant \Omega(n \log n)$ and the sparsity level $\theta \in \mathcal{O}(1/\sqrt{n})$. A similar approach has also been investigated by [23]. Li et al. [21] consider an $\ell^4$-maximization problem over the sphere, revealing benign global geometric structures of the problem. Correspondingly, they introduced a *noisy* Riemannian gradient descent (RGD) that solves the problem to approximate solutions in polynomial time.

These results are very inspiring but still suffer from quite a few limitations. The theory and method in [20] *only* applies to cases when $\boldsymbol{a}$ is approximately a delta function (which excludes most problems of interest) and $\{\boldsymbol{x}_i\}_{i=1}^p$ are *very* sparse. Li et al. [21] suggests that more generic kernels $\boldsymbol{a}$ can be handled via preconditioning of the data. However, due to the *heavy-tailed* behavior of $\ell^4$-loss, the sample complexity provided in [21] is quite *pessimistic*[3]. Moreover, noisy RGD is proved to converge with huge amounts of iterations [21], and it requires additional efforts to tune the noise parameters which is often unrealistic in practice. As mentioned in [21], one may use vanilla RGD which almost surely converges to a global minimum, but without guarantee on the number of iterations. On the other hand, Li et al. [21] only considered the Bernoulli-Rademacher model[4] which is quite restrictive.

**Contributions.** In this work, we introduce an efficient optimization method for solving MCS-BD. We consider a natural nonconvex formulation based on a smooth relaxation of $\ell^1$-loss. Under mild assumptions of the data, we prove the following result.

> With *random* initializations, a *vanilla* RGD efficiently finds an approximate solution, which can then be refined by a subgradient method that converges to the target solution in a *linear* rate.

We summarize our main result in Table 1. By comparison[5] with [21], our approach demonstrates *substantial* improvements for solving MCS-BD in terms of both sample and time complexity. Moreover, our experimental results imply that our analysis is still far from tight – the phase transitions suggest that $p \geqslant \Omega(\text{poly} \log(n))$ samples might be sufficient for exact recovery, which is favorable for applications (as real data in form of images can have millions of pixels, resulting in huge dimension $n$). Our analysis is inspired by recent results on orthogonal dictionary learning [24–26], but much of our theoretical analysis is tailored for MCS-BD with a few extra new ingredients. Our work is the first result provably showing that *vanilla* gradient descent type methods solve MCS-BD efficiently.

Moreover, our ideas could potentially lead to new algorithmic guarantees for other nonconvex problems such as blind gain and phase calibration [27, 28] and convolutional dictionary learning [29, 30]. The full version [31] of this work can be found at https://arxiv.org/abs/1908.10776.

## 2 Problem Formulation

To begin, we list our assumptions on the *unknown* kernel $\boldsymbol{a} \in \mathbb{R}^n$ and sparse inputs $\{\boldsymbol{x}_i\}_{i=1}^p \in \mathbb{R}^n$:

- *Invertible kernel.* We assume the kernel $\boldsymbol{a}$ to be *invertible* in the sense that its spectrum $\widehat{\boldsymbol{a}} = \boldsymbol{F}\boldsymbol{a}$ does not have zero entries, where $\widehat{\boldsymbol{a}} = \boldsymbol{F}\boldsymbol{a}$ is the discrete Fourier transform (DFT) of $\boldsymbol{a}$ with $\boldsymbol{F} \in \mathbb{C}^{n \times n}$ being the DFT matrix. Let $\boldsymbol{C}_{\boldsymbol{a}} \in \mathbb{R}^{n \times n}$ be an $n \times n$ circulant matrix whose first column is $\boldsymbol{a}$. Since this circulant matrix $\boldsymbol{C}_{\boldsymbol{a}}$ can be decomposed as $\boldsymbol{C}_{\boldsymbol{a}} = \boldsymbol{F}^* \operatorname{diag}(\widehat{\boldsymbol{a}})\, \boldsymbol{F}$ [32], it is also invertible and we define its condition number $\kappa(\boldsymbol{a}) := \max_i |\widehat{a}_i| / \min_i |\widehat{a}_i|$.

- *Random sparse signal.* The input signals $\{\boldsymbol{x}_i\}_{i=1}^p$ are *sparse*, and follow i.i.d. Bernoulli-Gaussian ($\mathcal{BG}(\theta)$) distribution:

$$\boldsymbol{x}_i = \boldsymbol{b}_i \odot \boldsymbol{g}_i, \qquad \boldsymbol{b}_i \sim_{i.i.d.} \mathcal{B}(\theta), \quad \boldsymbol{g}_i \sim_{i.i.d.} \mathcal{N}(\boldsymbol{0}, \boldsymbol{I}),$$

where $\theta \in [0, 1]$ is the Bernoulli-parameter which controls the sparsity level of $\boldsymbol{x}_i$.

As aforementioned, this assumption generalizes those used in [20, 21]. In particular, the first assumption on kernel $\boldsymbol{a}$ is much more practical than that of [20], in which $\boldsymbol{a}$ is assumed to be spiky. The second assumption is a generalization of the Bernoulli-Rademacher model adopted in [21].

Note that the MCS-BD problem exhibits intrinsic *signed scaling-shift* symmetry, i.e., for any $\alpha \neq 0$,

$$\boldsymbol{y}_i = s_{-\ell}\left[\pm\alpha\boldsymbol{a}\right] \circledast s_\ell\left[\pm\alpha^{-1}\boldsymbol{x}_i\right], \qquad i \in [p], \tag{2}$$

where $s_\ell[\cdot]$ denotes a cyclic shift operator of length $\ell$. Without loss of generality, for the rest of the paper we assume that the kernel $\boldsymbol{a}$ is normalized with $\|\boldsymbol{a}\| = 1$. Thus, we only hope to recover $\boldsymbol{a}$ and $\{\boldsymbol{x}_i\}_{i=1}^p$ up to a *signed shift ambiguity*,

**A nonconvex formulation.** Let $\boldsymbol{Y} = \begin{bmatrix} \boldsymbol{y}_1 & \boldsymbol{y}_2 & \cdots & \boldsymbol{y}_p \end{bmatrix}$ and $\boldsymbol{X} = \begin{bmatrix} \boldsymbol{x}_1 & \boldsymbol{x}_2 & \cdots & \boldsymbol{x}_p \end{bmatrix}$. We can rewrite the measurement (1) in a matrix-vector form via circulant matrices,

$$\boldsymbol{y}_i = \boldsymbol{a} \circledast \boldsymbol{x}_i = \boldsymbol{C}_{\boldsymbol{a}}\boldsymbol{x}_i, \ i \in [p] \implies \boldsymbol{Y} = \boldsymbol{C}_{\boldsymbol{a}}\boldsymbol{X},$$

Since $\boldsymbol{C}_{\boldsymbol{a}}$ is assumed to be invertible, we can define its corresponding *inverse kernel* $\boldsymbol{h} \in \mathbb{R}^n$ by $\boldsymbol{h} := \boldsymbol{F}^{-1}\widehat{\boldsymbol{a}}^{\odot-1}$ whose corresponding circulant matrix satisfies

$$\boldsymbol{C}_{\boldsymbol{h}} := \boldsymbol{F}^* \operatorname{diag}\left(\widehat{\boldsymbol{a}}^{\odot-1}\right) \boldsymbol{F} = \boldsymbol{C}_{\boldsymbol{a}}^{-1},$$

where $(\cdot)^{\odot-1}$ denotes entrywise inversion. Observing

$$\boldsymbol{C}_{\boldsymbol{h}} \cdot \boldsymbol{Y} = \underbrace{\boldsymbol{C}_{\boldsymbol{h}} \cdot \boldsymbol{C}_{\boldsymbol{a}}}_{= \boldsymbol{I}} \cdot \boldsymbol{X} = \underbrace{\boldsymbol{X}}_{\text{sparse}},$$

it leads us to consider the following objective

$$\min_{\boldsymbol{q}} \frac{1}{np} \|\boldsymbol{C}_{\boldsymbol{q}}\boldsymbol{Y}\|_0 = \frac{1}{np} \sum_{i=1}^p \|\boldsymbol{C}_{\boldsymbol{y}_i}\boldsymbol{q}\|_0, \qquad \text{s.t.} \quad \boldsymbol{q} \neq \boldsymbol{0}. \tag{3}$$

Obviously, when the solution of (3) is unique, the *only* minimizer is the inverse kernel $\boldsymbol{h}$ up to signed scaling-shift (i.e., $\boldsymbol{q}_\star = \pm\alpha s_\ell[\boldsymbol{h}]$), producing $\boldsymbol{C}_{\boldsymbol{h}}\boldsymbol{Y} = \boldsymbol{X}$ with the highest sparsity. The nonzero constraint $\boldsymbol{q} \neq \boldsymbol{0}$ is enforced simply to prevent the trivial solution $\boldsymbol{q} = \boldsymbol{0}$. Ideally, if we could solve (3) to obtain one of the target solutions $\boldsymbol{q}_\star = s_\ell[\boldsymbol{h}]$ up to a signed scaling, the kernel $\boldsymbol{a}$ and sparse signals $\{\boldsymbol{x}_i\}_{i=1}^p$ can be exactly recovered up to signed shift via

$$\boldsymbol{a}_\star = \boldsymbol{F}^{-1}\left[(\boldsymbol{F}\boldsymbol{q}_\star)^{\odot-1}\right], \qquad \boldsymbol{x}_i^\star = \boldsymbol{C}_{\boldsymbol{y}_i}\boldsymbol{q}_\star, \ (1 \leq i \leq p).$$

However, it has been known for decades that optimizing the basic $\ell_0$-formulation (3) is an NP-hard problem [33, 34]. Instead, we consider the following *nonconvex*[6] relaxation of the original problem (3):

$$\boxed{\min_{\boldsymbol{q}} \varphi_h(\boldsymbol{q}) := \frac{1}{np} \sum_{i=1}^p H_\mu\left(\boldsymbol{C}_{\boldsymbol{y}_i}\boldsymbol{P}\boldsymbol{q}\right), \qquad \text{s.t.} \quad \boldsymbol{q} \in \mathbb{S}^{n-1},} \tag{4}$$

where $H_\mu(\cdot)$ is the Huber loss [35] and $\boldsymbol{P}$ is a preconditioning matrix, both of which will be defined and discussed as follows.

[6]It is nonconvex because of the spherical constraint $\boldsymbol{q} \in \mathbb{S}^{n-1}$.

**Smooth sparsity surrogate.** It is well-known that $\ell^1$-norm serves as a natural sparsity surrogate for $\ell^0$-norm, but its nonsmoothness often makes it difficult for analysis[7]. Here, we consider the Huber loss[8] $H_\mu(\cdot)$ which is widely used in robust optimization [35]. It acts as a *smooth* sparsity surrogate of $\ell^1$ penalty and is defined as:

$$H_\mu(\boldsymbol{Z}) := \sum_{i=1}^{n} \sum_{j=1}^{p} h_\mu(Z_{ij}), \qquad h_\mu(z) := \begin{cases} |z| & |z| \geqslant \mu \\ \frac{z^2}{2\mu} + \frac{\mu}{2} & |z| < \mu \end{cases}, \tag{5}$$

where $\mu > 0$ is a smoothing scalar. Our choice $h_\mu(z)$ is first-order smooth, and behaves exactly same as the $\ell^1$-norm for $|z| \geqslant \mu$. In contrast, although the the $\ell^4$ objective in [21] is smooth, it *only* promotes sparsity in special cases. Moreover, it results in a heavy-tailed process, producing flat landscape around target solutions, and requiring substantially more samples for measure concentration. Figure 1 shows a comparison of optimization landscapes in low dimension: the Huber-loss behaves very similar to the $\ell^1$-loss, while optimizing the $\ell^4$-loss could result in large approximation error.

**Preconditioning.** An ill-conditioned kernel $\boldsymbol{a}$ can result in poor optimization landscapes (see Figure 1 for an illustration). To alleviate this effect, we introduce a preconditioning matrix $\boldsymbol{P} \in \mathbb{R}^{n \times n}$ [21,36,37], defined as follows[9]

$$\boldsymbol{P} = \left( \frac{1}{\theta n p} \sum_{i=1}^{p} \boldsymbol{C}_{\boldsymbol{y}_i}^\top \boldsymbol{C}_{\boldsymbol{y}_i} \right)^{-1/2}, \tag{6}$$

which refines the function landscapes by orthogonalizing the circulant matrix $\boldsymbol{C}_{\boldsymbol{a}}$:

$$\underbrace{\boldsymbol{C}_{\boldsymbol{a}} \boldsymbol{P}}_{\boldsymbol{R}} \approx \underbrace{\boldsymbol{C}_{\boldsymbol{a}} \left( \boldsymbol{C}_{\boldsymbol{a}}^\top \boldsymbol{C}_{\boldsymbol{a}} \right)^{-1/2}}_{\boldsymbol{Q} \text{ orthogonal}}. \tag{7}$$

Since $\boldsymbol{P} \approx (\boldsymbol{C}_{\boldsymbol{a}} \boldsymbol{C}_{\boldsymbol{a}})^{-1/2}$, $\boldsymbol{R}$ can be proved to be very close to the orthogonal matrix $\boldsymbol{Q}$. Thus, $\boldsymbol{R}$ is much more well-conditioned than $\boldsymbol{C}_{\boldsymbol{a}}$. As illustrated in Figure 1, a comparison of optimization landscapes without and with preconditioning shows that preconditioning symmetrifies the optimization landscapes and eliminates *spurious* local minimizers. Therefore, it makes the problem more amendable for optimization.

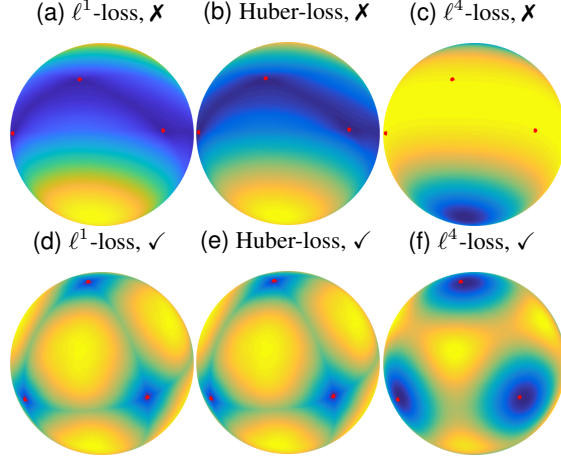

(a) $\ell^1$-loss, ✗    (b) Huber-loss, ✗    (c) $\ell^4$-loss, ✗

(d) $\ell^1$-loss, ✓    (e) Huber-loss, ✓    (f) $\ell^4$-loss, ✓

Figure 1: **Comparison of optimization landscapes for different loss functions.** Here ✗ and ✓ mean without and with the preconditioning matrix $\boldsymbol{P}$, respectively. Each figure plots the function values of the loss over $\mathbb{S}^2$, where the function values are all normalized between 0 and 1 (darker color means smaller value, and vice versa). The small red dots on the landscapes denote shifts of the ground truths.

**Constrain over the sphere $\mathbb{S}^{n-1}$.** We relax the nonconvex constraint $\boldsymbol{q} \neq \boldsymbol{0}$ in (3) by a unit norm constraint on $\boldsymbol{q}$. The norm constraint removes the scaling ambiguity as well as prevents the trivial solution $\boldsymbol{q} = \boldsymbol{0}$. Note that the choice of the norm has strong implication for computation. When $\boldsymbol{q}$ is constrained over $\ell^\infty$-norm, the $\ell^1/\ell^\infty$ optimization problem breaks beyond sparsity level $\theta \geqslant \Omega(1/\sqrt{n})$ [20]. In contrast, the sphere $\mathbb{S}^{n-1}$ is a homogeneous Riemannian manifold. It has been shown recently that optimizing over the sphere often leads to optimal sparsity $\theta \in \mathcal{O}(1)$ [21, 22, 36, 38]. Therefore, we choose to work with $\boldsymbol{q} \in \mathbb{S}^{n-1}$ and we also show similar recovery results for MCS-BD.

Next, we develop efficient first-order methods and provide guarantees for exact recovery.

# 3 Main Results and Analysis

In this section, we show that the underlying benign *first-order geometry* of the optimization landscapes of (4) *enables* efficient and exact recovery using *vanilla* gradient descent methods, even with *random* initialization. Our main result can be captured by the following theorem.

**Theorem 3.1** *We assume that the kernel $\boldsymbol{a}$ is invertible with condition number $\kappa$, and $\{\boldsymbol{x}_i\}_{i=1}^p \sim \mathcal{BG}(\theta)$. Suppose $\theta \in \left(\frac{1}{n}, \frac{1}{3}\right)$ and $\mu \leqslant c \min\left\{\theta, \frac{1}{\sqrt{n}}\right\}$. Whenever*

$$p \geqslant C \max\left\{n, \frac{\kappa^8}{\theta\mu^2\sigma_{\min}^2}\log^4 n\right\}\theta^{-2}n^4\log^3(n)\log\left(\frac{\theta n}{\mu}\right), \tag{8}$$

*w.h.p. the function (4) satisfies certain regularity conditions (see Theorem 3.2), allowing us to design an efficient vanilla first-order method. In particular, with probability at least $\frac{1}{2}$, by using a random initialization, the algorithms provably recover the target solution up to a signed shift with $\varepsilon$-precision in a linear rate*

$$\#Iter \;\leqslant\; C'\left(\theta^{-1}n^4\log\left(\frac{1}{\mu}\right) + \log(np)\log\left(\frac{1}{\varepsilon}\right)\right).$$

**Remark 1.** In the following, we explain our results in several aspects.

- *Conditions and Assumptions.* Here, as the MCS-BD problem becomes trivial[10] when $\theta \leqslant 1/n$, we only focus on the regime $\theta > 1/n$. Similar to [21], our result only requires the kernel $\boldsymbol{a}$ to be invertible and sparsity level $\theta$ to be constant. In contrast, the method in [20] only works when the kernel $\boldsymbol{a}$ is spiky and $\{\boldsymbol{x}_i\}_{i=1}^p$ are very sparse $\theta \in \mathcal{O}(1/\sqrt{n})$, excluding most problems of interest.

- *Sample Complexity.* As shown in Table 1, our sample complexity $p \geqslant \widetilde{\Omega}(\max\left\{n, \kappa^8/\mu^2\right\}n^4)$ in (8) improves upon the result $p \geqslant \widetilde{\Omega}(\max\left\{n, \kappa^8\right\}n^8/\varepsilon^8)$ in [21]. As aforementioned, this improvement partly owes to the similarity of the Huber-loss to $\ell^1$-loss, so that the Huber-loss is much less heavy-tailed than the $\ell^4$-loss studied in [21], requiring fewer samples for measure concentration. Still, our result leaves much room for improvement – we believe the sample dependency on $\theta^{-1}$ is an artifact of our analysis[11], and the phase transition in Figure 5 suggests that $p \geqslant \Omega(\text{poly}\log(n))$ samples might be sufficient for exact recovery.

- *Algorithmic Convergence.* Finally, it should be noted that the number of iteration $\widetilde{O}\left(n^4 + \log\left(1/\varepsilon\right)\right)$ for our algorithm substantially improves upon that $\widetilde{\mathcal{O}}(n^{12}/\varepsilon^2)$ of the noisy RGD in [21, Theorem IV.2]. This has been achieved via a two-stage approach: (i) we first run $\mathcal{O}(n^4)$ iterations of vanilla RGD to obtain an approximate solution; (ii) then perform a subgradient method with linear convergence to the ground-truth. Moreover, without any noise parameters to tune, vanilla RGD is more practical than the noisy RGD in [21].

## 3.1 A glimpse of high dimensional geometry

To study the optimization landscape of (5), we simplify the problem by a change of variable $\overline{\boldsymbol{q}} = \boldsymbol{Q}\boldsymbol{q}$, which rotates the space by the orthogonal matrix $\boldsymbol{Q}$ in (7). Since the rotation $\boldsymbol{Q}$ does not change the optimization landscape, by an abuse of notation of $\boldsymbol{q}$ and $\overline{\boldsymbol{q}}$, we can rewrite the problem (5) as

$$\min_{\boldsymbol{q}}\; f(\boldsymbol{q}) \;:=\; \frac{1}{np}\sum_{i=1}^p H_\mu\left(\boldsymbol{C}_{\boldsymbol{x}_i}\boldsymbol{R}\boldsymbol{Q}^{-1}\boldsymbol{q}\right), \qquad \text{s.t.} \quad \|\boldsymbol{q}\| = 1, \tag{9}$$

where we also used the fact that $\boldsymbol{C}_{\boldsymbol{y}_i}\boldsymbol{P} = \boldsymbol{C}_{\boldsymbol{x}_i}\boldsymbol{R}$ in (7). Moreover, since $\boldsymbol{R} \approx \boldsymbol{Q}$ is *near orthogonal*, by assuming $\boldsymbol{R}\boldsymbol{Q}^{-1} = \boldsymbol{I}$, for *pure* analysis purposes we can further reduce (9) to

$$\min_{\boldsymbol{q}}\; \widetilde{f}(\boldsymbol{q}) = \frac{1}{np}\sum_{i=1}^p H_\mu\left(\boldsymbol{C}_{\boldsymbol{x}_i}\boldsymbol{q}\right), \quad \text{s.t.} \quad \|\boldsymbol{q}\| = 1. \tag{10}$$

The reduction in (10) is simpler and much easier for analysis. By a similar analysis as [24, 36], it can be shown that asymptotically the landscape is highly symmetric and the standard basis vectors $\{\pm\boldsymbol{e}_i\}_{i=1}^n$ are approximately[12] the only global minimizers. Hence, as $\boldsymbol{R}\boldsymbol{Q}^{-1} \approx \boldsymbol{I}$, we can study the

landscape of $f(\boldsymbol{q})$ via studying the landscape of $\widetilde{f}(\boldsymbol{q})$ followed by a perturbation analysis. As illustrated in Figure 2, based on the target solutions of $\widetilde{f}(\boldsymbol{q})$, we partition the sphere into $2n$ symmetric regions, and consider $2n$ (disjoint) subsets of each region[13] [24, 25]

$$\mathcal{S}_\xi^{i\pm} := \left\{ \boldsymbol{q} \in \mathbb{S}^{n-1} \mid \frac{|q_i|}{\|\boldsymbol{q}_{-i}\|_\infty} \geqslant \sqrt{1+\xi},\ q_i \gtrless 0 \right\}, \quad \xi \in [0, \infty),$$

where $\boldsymbol{q}_{-i}$ is a subvector of $\boldsymbol{q}$ with $i$-th entry removed. For every $i \in [n]$, $\mathcal{S}_\xi^{i+}$ (or $\mathcal{S}_\xi^{i-}$) contains exactly one of the target solution $\boldsymbol{e}_i$ (or $-\boldsymbol{e}_i$), and all points in this set have one unique largest entry with index $i$, so that they are closer to $\boldsymbol{e}_i$ (or $-\boldsymbol{e}_i$) in $\ell^\infty$ distance than all the other standard basis vectors. As shown in Figure 2, the union of these sets form a full partition of the sphere only when $\xi = 0$. While for small $\xi > 0$, each disjoint set excludes all the saddle points and maximizers, but their union covers most measure of the sphere: when $\xi = (5\log n)^{-1}$, their union covers at least half of the sphere, and hence a random initialization falls into one of the regions $\mathcal{S}_\xi^{i\pm}$ with probability at least $1/2$ [25]. Therefore, we can only consider the optimization landscapes on the sets $\mathcal{S}_\xi^{i\pm}$, where we show the Riemannian gradient of $f(\boldsymbol{q})$

$$\operatorname{grad} f(\boldsymbol{q}) := \mathcal{P}_{\boldsymbol{q}^\perp} \nabla f(\boldsymbol{q}) = \left( \boldsymbol{I} - \boldsymbol{q}\boldsymbol{q}^\top \right) \nabla f(\boldsymbol{q})$$

satisfies the following properties in each set $\mathcal{S}_\xi^{i\pm}$. For convenience, we will simply present the results in terms of $\mathcal{S}_\xi^{i+}$, but they also hold for $\mathcal{S}_\xi^{i-}$.

**Proposition 3.2 (Regularity Condition)** *Suppose* $\theta \in \left( \frac{1}{n}, \frac{1}{3} \right)$ *and* $\mu \leqslant c\min\left\{ \theta, \frac{1}{\sqrt{n}} \right\}$. *When* $p$ *satisfies* (8), *w.h.p. over the randomness of* $\{\boldsymbol{x}_i\}_{i=1}^p$, *the Riemannian gradient of* $f(\boldsymbol{q})$ *satisfies*

$$\langle \operatorname{grad} f(\boldsymbol{q}), q_i\boldsymbol{q} - \boldsymbol{e}_i \rangle \geqslant \alpha(\boldsymbol{q}) \|\boldsymbol{q} - \boldsymbol{e}_i\|, \tag{11}$$

*for any* $\boldsymbol{q} \in \mathcal{S}_\xi^{i+}$ *with* $\sqrt{1-q_i^2} \geqslant \mu$, *where the regularity parameter is*

$$\alpha(\boldsymbol{q}) = \begin{cases} c'\theta(1-\theta)q_i & \sqrt{1-q_i^2} \in [\mu, \gamma] \\ c'\theta(1-\theta)n^{-1}q_i & \sqrt{1-q_i^2} \geqslant \gamma \end{cases}$$

*which increases as* $\boldsymbol{q}$ *gets closer to* $\boldsymbol{e}_i$. *Here* $\gamma \in [\mu, 1)$ *is some constant.*

**Remark 2.** Here, our result is stated with respect to $\boldsymbol{e}_i$ for the sake of simplicity. It should be noted that asymptotically the global minimizer of (9) is $\beta(\boldsymbol{R}\boldsymbol{Q}^{-1})^{-1}\boldsymbol{e}_i$ rather than $\boldsymbol{e}_i$, where $\beta$ is a normalization factor. Nonetheless, as $\boldsymbol{R}\boldsymbol{Q}^{-1} \approx \boldsymbol{I}$, the global optimizer $\beta(\boldsymbol{R}\boldsymbol{Q}^{-1})^{-1}\boldsymbol{e}_i$ of (9) is very close to $\boldsymbol{e}_i$, so that we can state a similar result with respect to $\beta(\boldsymbol{R}\boldsymbol{Q}^{-1})^{-1}\boldsymbol{e}_i$. The regularity condition (11) shows that any $\boldsymbol{q} \in \mathcal{S}_\xi^{i+}$ with $\sqrt{1-q_i^2} \geqslant \mu$ is not a stationary point. Similar regularity condition has been proved for phase retrieval [39], dictionary learning [25], etc. Such condition implies that the negative gradient direction coincides with the direction to the target solution. The lower bound on Riemannian gradient ensures that the iterate still makes sufficient progress to the target solution, even when it is close to the target.

To ensure convergence of RGD, we also need to show the following property, so that once initialized in $\mathcal{S}_\xi^{i+}$ the iterates of the RGD method *implicitly* regularize themselves staying in the set $\mathcal{S}_\xi^{i+}$. This ensures that the regularity condition (11) holds through the solution path of the RGD method.

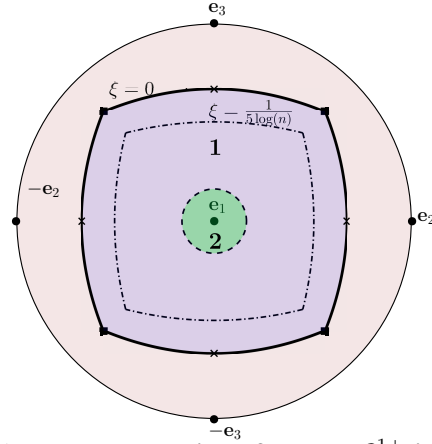

Figure 2: **Illustration of the set $\mathcal{S}_\xi^{1+}$ in 3-dimension.** Region 1 (purple region) denotes the interior of $\mathcal{S}_\xi^{1+}$ when $\xi = 0$, where it includes one unique target solution. We show the regularity condition (11) within $\mathcal{S}_\xi^{1+}$, excluding a green region of order $\mathcal{O}(\mu)$ (i.e., Region 2) due to Huber smoothing.

**Proposition 3.3 (Implicit Regularization)** *Under the same condition of Proposition 3.2, w.h.p. over the randomness of $\{\boldsymbol{x}_i\}_{i=1}^p$, the Riemannian gradient of $f(\boldsymbol{q})$ satisfies*

$$\left\langle \operatorname{grad} f(\boldsymbol{q}), \frac{1}{q_j}\boldsymbol{e}_j - \frac{1}{q_i}\boldsymbol{e}_i \right\rangle \geqslant c_4 \frac{\theta(1-\theta)}{n} \frac{\xi}{1+\xi}, \tag{12}$$

*for all $\boldsymbol{q} \in \mathcal{S}_\xi^{i+}$ and any $q_j$ such that $j \neq i$ and $q_j^2 \geqslant \frac{1}{3}q_i^2$.*

**Remark 3.** In a nutshell, (12) guarantees that the negative gradient direction points towards $\boldsymbol{e}_i$ component-wisely for relatively large components (i.e., $q_j^2 \geqslant \frac{1}{3}q_i^2$, $\forall j \neq i$). With this, we can prove that those components will not increase after gradient update, ensuring the iterates stay within the region $\mathcal{S}_\xi^{i+}$. This type of implicit regularizations for the gradient has also been discovered for many nonconvex optimization problems, such as low-rank matrix factorizations [40–43], phase retrieval [44], and neural network training [45].

### 3.2 From geometry to efficient optimization

**Phase 1: Finding an approximate solution via RGD.** Starting from a *random* initialization $\boldsymbol{q}^{(0)}$ uniformly drawn from $\mathbb{S}^{n-1}$, we solve the problem (4) via *vanilla* RGD

$$\boldsymbol{q}^{(k+1)} = \mathcal{P}_{\mathbb{S}^{n-1}} \left( \boldsymbol{q}^{(k)} - \tau \cdot \operatorname{grad} f(\boldsymbol{q}^{(k)}) \right), \tag{13}$$

where $\tau > 0$ is the stepsize, and $\mathcal{P}_{\mathbb{S}^{n-1}}(\cdot)$ is a projection operator onto the sphere $\mathbb{S}^{n-1}$.

**Proposition 3.4 (Linear convergence of gradient descent)** *Suppose Proposition 3.2 and Proposition 3.3 hold. With probability at least $1/2$, the random initialization $\boldsymbol{q}^{(0)}$ falls into one of the regions $\mathcal{S}_\xi^{i\pm}$ for some $i \in [n]$. Choosing a fixed step size $\tau \leqslant \frac{c}{n} \min\{\mu, n^{-3/2}\}$ in (13), we have*

$$\left\| \boldsymbol{q}^{(k)} - \boldsymbol{e}_i \right\| \leqslant 2\mu, \ \forall k \geqslant N := \frac{C}{\theta} n^4 \log\left(\frac{1}{\mu}\right).$$

Because of the preconditioning and smoothing via Huber loss (5), the geometry structure in Proposition 3.2 implies that the gradient descent method can only produce an approximate solution $\boldsymbol{q}_s$ up to a precision $\mathcal{O}(\mu)$. Moreover, as we can show that $\|\boldsymbol{e}_i - \beta(\boldsymbol{RQ}^{-1})^{-1}\boldsymbol{e}_i\| \leqslant \mu/2$, it does not make much difference of stating the result in terms of either $\boldsymbol{e}_i$ or $\beta(\boldsymbol{RQ}^{-1})^{-1}\boldsymbol{e}_i$. Next, we show that, by using $\boldsymbol{q}_s$ as a *warm start*, an extra linear program (LP) rounding procedure produces an exact solution $(\boldsymbol{RQ}^{-1})^{-1}\boldsymbol{e}_i$ up to a scaling factor in a few iterations.

**Phase 2: Exact solution via LP rounding.** Let $\boldsymbol{r} = \boldsymbol{q}_s$ be the solution obtain from solving RGD. We recover the exact solution by solving the following LP problem[14]

$$\min_{\boldsymbol{q}} \ \zeta(\boldsymbol{q}) := \frac{1}{np} \sum_{i=1}^p \left\| \boldsymbol{C}_{\boldsymbol{x}_i} \boldsymbol{RQ}^{-1} \boldsymbol{q} \right\|_1 \quad \text{s.t.} \quad \langle \boldsymbol{r}, \boldsymbol{q} \rangle = 1. \tag{14}$$

Since the feasible set $\langle \boldsymbol{r}, \boldsymbol{q} \rangle = 1$ is essentially the tangent space of the sphere $\mathbb{S}^{n-1}$ at $\boldsymbol{r}$, and $\boldsymbol{r} = \boldsymbol{q}_s$ is pretty close to the target solution, one should expect that the optimizer $\boldsymbol{q}_\star$ of (14) exactly recovers the inverse kernel $\boldsymbol{h}$ up to a scaled-shift. The problem (14) is *convex* and can be directly solved using standard tools such as CVX [46], but it will be time consuming for large dataset. Instead, we introduce an efficient projected subgradient method for solving (14),

$$\boldsymbol{q}^{(k+1)} = \boldsymbol{q}^{(k)} - \tau^{(k)} \mathcal{P}_{\boldsymbol{r}^\perp} \boldsymbol{g}^{(k)}, \quad \boldsymbol{g}^{(k)} = \frac{1}{np} \sum_{i=1}^p \left(\boldsymbol{RQ}^{-1}\right)^\top \boldsymbol{C}_{\boldsymbol{x}_i}^\top \operatorname{sign}\left(\boldsymbol{C}_{\boldsymbol{x}_i} \boldsymbol{RQ}^{-1} \boldsymbol{q}^{(k)}\right). \tag{15}$$

For convenience, let $\tilde{\boldsymbol{r}} := \left(\boldsymbol{RQ}^{-1}\right)^{-\top} \boldsymbol{r}$, and define the distance $d(\boldsymbol{q})$ between $\boldsymbol{q}$ and the truth

$$\operatorname{dist}(\boldsymbol{q}) := \|\boldsymbol{d}(\boldsymbol{q})\|, \quad \boldsymbol{d}(\boldsymbol{q}) := \boldsymbol{q} - \left(\boldsymbol{RQ}^{-1}\right)^{-1} \frac{\boldsymbol{e}_n}{\tilde{r}_n}.$$

**Proposition 3.5** *Suppose $\mu \leqslant \frac{1}{25}$ and let $\boldsymbol{r} = \boldsymbol{q}_s$ which satisfies $\|\boldsymbol{r} - \boldsymbol{e}_i\| \leqslant 2\mu$. Choose $\tau^{(k)} = \eta^k \tau^{(0)}$ with $\tau^{(0)} = c_1 \log^{-2}(np)$ and $\eta \in \left[\left(1 - c_2 \log^{-2}(np)\right)^{1/2}, 1\right)$. Under the same condition of Theorem 3.1, w.h.p. the sequence $\{\boldsymbol{q}^{(k)}\}$ produced by (15) with $\boldsymbol{q}^{(0)} = \boldsymbol{r}$ converges to the target solution in a linear rate, i.e.,*

$$\operatorname{dist}(\boldsymbol{q}^{(k)}) \leqslant C\eta^k, \quad \forall k = 0, 1, 2, \cdots.$$

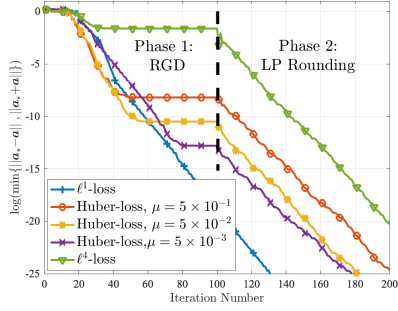

Figure 3: **Comparison of iterate convergence.** $p = 50$, $n = 200$, $\theta = 0.25$.

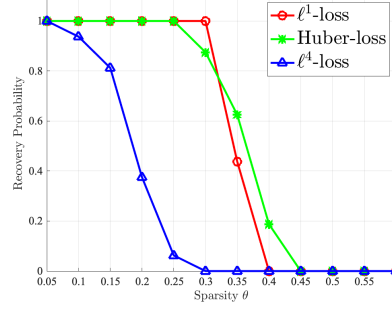

Figure 4: **Comparison of recovery probability with varying $\theta$.** $p = 50$, $n = 500$.

**Remark 4.** Unlike smooth problems, in general, subgradient methods for nonsmooth problem have to use *geometrically* diminishing stepsize to achieve linear convergence[15] [48–51]. The underlying geometry that supports the use of geometric diminishing step size and linear convergence is the so-called *sharpness* property [52] of the problem (14). In particular, for some constant $\alpha > 0$, we prove $\zeta(\boldsymbol{q})$ is sharp in the sense that

$$\zeta(\boldsymbol{q}) - \zeta\left(\left(\boldsymbol{R}\boldsymbol{Q}^{-1}\right)^{-1}\boldsymbol{e}_n/\widetilde{r}_n\right) \geqslant \alpha \cdot \mathrm{dist}(\boldsymbol{q}), \quad \forall \langle \boldsymbol{r}, \boldsymbol{q} \rangle = 1.$$

Finally, we end this section by noting that although we use matrix-vector form of convolutions in (13) and (15), all the matrix-vector multiplications can be efficiently implemented by FFT, including the preconditioning matrix in (6) which is also a circulant matrix. With FFT, the complexities of implementing one gradient update in (13) and subgradient in (15) are both $\mathcal{O}(pn\log n)$.

## 4 Experiment

**Experiments on 1D synthetic dataset.** First, we conduct a series of experiments on synthetic dataset to demonstrate the superior performance of the vanilla RGD method (13). For all synthetic experiments, we generate the measurements $\boldsymbol{y}_i = \boldsymbol{a} \circledast \boldsymbol{x}_i$ $(1 \leqslant i \leqslant p)$, with the ground truth kernel $\boldsymbol{a} \in \mathbb{R}^n$ drawn uniformly random from the sphere $\mathbb{S}^{n-1}$ (i.e., $\boldsymbol{a} \sim \mathcal{U}(\mathbb{S}^{n-1})$), and with sparse signals $\boldsymbol{x}_i \in \mathbb{R}^n, i = [p]$ drawn from i.i.d. Bernoulli-Gaussian distribution $\boldsymbol{x}_i \sim_{i.i.d.} \mathcal{BG}(\theta)$.

We compare the performances of RGD[16] with random initialization on $\ell^1$-loss, Huber-loss, and $\ell^4$-loss considered in [21]. We use line-search for adaptively choosing stepsize. For a fair comparison of optimizing all losses, we refine all solutions with the LP rounding procedure (14) optimized by subgradient descent (15), and use the same random initialization uniformly drawn from the sphere.

For judging the success of recovery, let $\boldsymbol{q}_\star$ be a solution produced by the algorithm and we define

$$\rho_{acc}(\boldsymbol{q}_\star) := \|\boldsymbol{C}_a \boldsymbol{P} \boldsymbol{q}_\star\|_\infty / \|\boldsymbol{C}_a \boldsymbol{P} \boldsymbol{q}_\star\| \in [0, 1].$$

If $\boldsymbol{q}_\star$ achieves the target solution, it should satisfy $\boldsymbol{P}\boldsymbol{q}_\star = \boldsymbol{h}$, with $\boldsymbol{h}$ being the inverse kernel of $\boldsymbol{a}$ and thus $\rho_{acc}(\boldsymbol{q}_\star) = 1$. Therefore, we should expect $\rho_{acc}(\boldsymbol{q}_\star) \approx 1$ when an algorithm produces a correct solution. For the following simulations, we assume successful recovery whenever $\rho_{acc}(\boldsymbol{q}_\star) \geqslant 0.95$.

(a) **Comparison of iterate convergence.** We first compare the convergence in terms of the distance from the iterate to the target solution for all losses using RGD. As shown in Figure 3, in Phase 1 optimizing $\ell^4$-loss can only produce an approximate solution up to precision $10^{-2}$. In contrast, optimizing Huber-loss converges much faster, and producing much more accurate solutions as $\mu$ decreases. In Phase 2, subgradient descent converges linearly to the exact solution.

(b) **Recovery with varying sparsity.** Fix $n = 500$ and $p = 50$, we compare the recovery probability with varying sparsity level $\theta$. For each $\theta$, we repeat the simulation for 15 times. As illustrated in Figure 4, optimizing Huber-loss enables successful recovery for much larger $\theta$ in comparison with that of $\ell^4$-loss. The performances of optimizing $\ell^1$-loss and Huber-loss are quite similar.

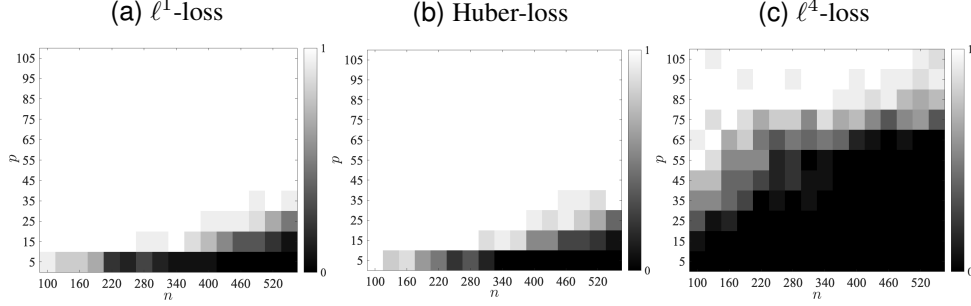

Figure 5: **Comparison of phase transition on** $(p, n)$ **with fixed** $\theta = 0.25$.

(c) **Phase transition on** $(p, n)$. Finally, we fix $\theta = 0.25$, and test the dependency of sample number $p$ on the dimension $n$ via phase transition plots. For each individual $(p, n)$, we repeat the simulation for 15 times. Whiter pixels in Figure 5 indicates higher success probability, and vice versa. As shown in Figure 5, for a given $n$, optimizing Huber-loss requires much fewer samples $p$ for recovery in comparison with that of $\ell^4$-loss. The performance of optimizing $\ell^1$-loss and Huber-loss is comparable; we conjecture sample dependency for optimizing both losses is $p \geqslant \Omega(\text{poly} \log(n))$. In contrast, optimizing $\ell^4$-loss might need $p \geqslant \Omega(n)$ samples.

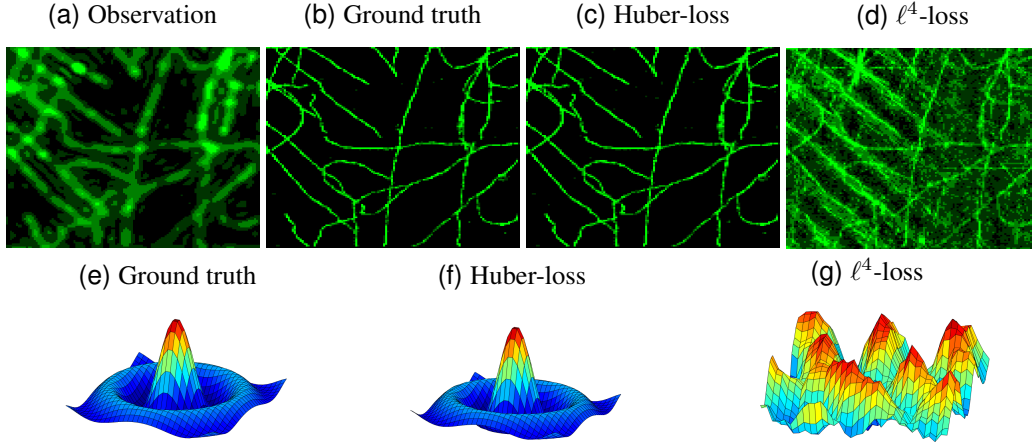

Figure 6: **STORM imaging via solving MCS-BD.** The first line shows (a) observed image, (b) ground truth, (c) recovered image by optimizing Huber-loss, and (d) by $\ell^4$-loss. The second line, (e) ground truth kernel, (f) recovered by optimizing Huber-loss, and (g) by $\ell^4$-loss.

**Real experiment on 2D super-resolution microscopy imaging.** As introduced in Section 1, stochastic optical reconstruction microscopy (STORM) is a new computation based imaging technique which breaks the resolution limits of optical fluorescence microscopy [4, 17, 18]. The basic principle is using photoswitchable florescent probes to create multiple images $Y_i = A \circledast X_i$, where $\circledast$ denotes 2D circular convolution, $A$ is PSF, and $\{X_i\}_{i=1}^p$ are sparse point-sources. In 3D imaging, the PSF $A$ is hard to estimate due to defocus and unknown aberrations [19], so that we want to jointly estimate the PSF $A$ and point sources $\{X_i\}_{i=1}^p$. Once $\{X_i\}_{i=1}^p$ are recovered, we can obtain a high resolution image by aggregating all $X_i$. We test our algorithms on this task, by using $p = 1000$ frames obtained from a standard dataset[17]. As demonstrated in Figure 6, optimizing Huber-loss using vanilla RGD can near perfectly recover both the underlying Bessel PSF and point-sources, producing accurate high resolution image. In contrast, optimizing $\ell^4$-loss [21] fails to recover the PSF, resulting in some aliasing effects of the recovered image.

## Discussion & Acknowledgement

Due to space limitation, we refer readers to Section 5 of our full paper [31] for a comprehensive discussion. QQ also would like to acknowledge the support of Microsoft PhD fellowship, and Moore-Sloan foundation fellowship. XL would like to acknowledge the support by Grant CUHK14210617 from the Hong Kong Research Grants Council. ZZ was partly supported by NSF Grant 1704458.

## Footnotes

*ZZ is also with the Department of Electrical & Computer Engineering, University of Denver.

[2]Here, (i) $\mathcal{BG}(\theta)$ and $\mathcal{BR}(\theta)$ denote Bernoulli-Gaussian and Bernoulli-Rademacher distribution, respectively; (ii) $\theta \in [0, 1]$ is the Bernoulli parameter controlling the sparsity level of $\boldsymbol{x}_i$; (iii) $\varepsilon$ denotes the recovery precision of global solution $\boldsymbol{a}_\star$, i.e., $\min_\ell \|\boldsymbol{a} - s_\ell [\boldsymbol{a}_\star]\| \leqslant \varepsilon$; (iv) $\widetilde{\mathcal{O}}$ and $\widetilde{\Omega}$ hides $\log(n)$, $\theta$ and other factors.

[3]As the tail of $\mathcal{BG}(\theta)$ distribution is heavier than that of $\mathcal{BR}(\theta)$, their sample complexity would be even worse if $\mathcal{BG}(\theta)$ model was considered.

[4]We say $\boldsymbol{x}$ obeys a Bernoulli-Rademacher distribution when $\boldsymbol{x} = \boldsymbol{b} \odot \boldsymbol{r}$ where $\odot$ denotes point-wise product, $\boldsymbol{b}$ follows i.i.d. Bernoulli distribution and $\boldsymbol{r}$ follows i.i.d. Rademacher distribution.

[5]We do not find a direct comparison with [20] meaningful, mainly due to its limitations of the kernel assumption and sparsity level $\theta \in \mathcal{O}(1/\sqrt{n})$ discussed above.

[7]The subgradient of $\ell^1$-loss is *non-Lipschitz*, which introduces tremendous difficulty in controlling suprema of random process and perturbation analysis for preconditioning.

[8]Actually, $h_\mu(\cdot)$ is a scaled and elevated version of standard Huber function $h_\mu^s(z)$, with $h_\mu(z) = \frac{1}{\mu} h_\mu^s(z) + \frac{\mu}{2}$. Hence in our framework minimizing with $h_\mu(z)$ is equivalent to minimizing with $h_\mu^s(z)$.

[9]Here, the sparsity $\theta$ serves as a normalization purpose. It is often not known ahead of time, but the scaling here does not change the optimization landscape.

[10]The problem becomes trivial when $\theta \leqslant 1/n$ because $\theta n = 1$ so that each $\boldsymbol{x}_i$ tends to be an one sparse $\delta$-function.

[11]The same $\theta^{-1}$ dependency also appears in [21, 24, 25, 36, 37].

[12]The standard basis $\{\pm\boldsymbol{e}_i\}_{i=1}^n$ are exact global solutions for $\ell^1$-loss. The Huber loss we considered here introduces small approximation errors due to its smoothing effects.

[13] Here, we define $\|\boldsymbol{q}_{-i}\|_\infty^{-1} = +\infty$ when $\|\boldsymbol{q}_{-i}\|_\infty = 0$, so that the set $\mathcal{S}_\xi^{i+}$ is compact and $\boldsymbol{e}_i$ is also contained in the set.

[14]For convenience, we state this problem in the rotated space. For the original problem (5), we should solve an equivalent problem of (14) as $\min_{\boldsymbol{q}} \zeta(\boldsymbol{q}) := \frac{1}{np} \sum_{i=1}^p \|\boldsymbol{C}_{\boldsymbol{y}_i} \boldsymbol{P}\boldsymbol{q}\|_1$, s.t. $\langle \overline{\boldsymbol{r}}, \boldsymbol{q} \rangle = 1$, with $\overline{\boldsymbol{r}} = \boldsymbol{Q}^\top \boldsymbol{r}$.

[15]Typical choices such as $\tau^{(k)} = \mathcal{O}(1/k)$ and $\tau^{(k)} = \mathcal{O}(1/\sqrt{k})$ lead to sublinear convergence [47–51].

[16]For $\ell^1$-loss, we use Riemannian subgradient method, similar to (15).

[17]Available at http://bigwww.epfl.ch/smlm/datasets/index.html?p=tubulin-conjal647.

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
