[Supplementary Material]



# Appendices

The appendices are organized as follows. In Appendix A we introduce the basic notations and problem reductions that are used throughout the main draft and the appendix. In Appendix B we describe and prove the main geometric properties of the optimization landscape for Huber loss. In Appendix C, we provide global convergence analysis for the propose Riemannian gradient descent methods for optimizing the Huber loss, and the subgradient methods for solving LP rounding. We list the basic technical tools and results in Appendix D. All the technical geometric analysis are postponed to Appendix E, Appendix F, Appendix G, and Appendix H. Finally, in Appendix I we describe the proposed optimization algorithms in full details for all $\ell^1$, Huber, and $\ell^4$ losses.

## A  Basic Notations and Problem Reductions

### A.1  Notations

Throughout this paper, all vectors/matrices are written in bold font $\boldsymbol{a}/\boldsymbol{A}$; indexed values are written as $a_i$, $A_{ij}$. We use $\boldsymbol{v}_{-i}$ to denote a subvector of $\boldsymbol{v}$ without the $i$-th entry. Zeros or ones vectors are defined as $\boldsymbol{0}_m$ or $\boldsymbol{1}_m$ with $m$ denoting its length, and $i$-th canonical basis vector defined as $\boldsymbol{e}_i$. We use $\mathbb{S}^{n-1}$ to denote an $n$-dimensional unit sphere in the Euclidean space $\mathbb{R}^n$. We use $\boldsymbol{z}^{(k)}$ to denote an optimization variable $\boldsymbol{z}$ at $k$-th iteration. We let $[m] = \{1, 2, \cdots, m\}$. Let $\boldsymbol{F}_n \in \mathbb{C}^{n \times n}$ denote a unnormalized $n \times n$ Fourier matrix, with $\|\boldsymbol{F}_n\| = \sqrt{n}$, and $\boldsymbol{F}_n^{-1} = n^{-1}\boldsymbol{F}_n^*$.

We define $\mathrm{sign}(\cdot)$ as

$$\mathrm{sign}(z) = \begin{cases} z/|z|, & z \neq 0 \\ 0, & z = 0 \end{cases}$$

**Some basic operators.**  We use $\boldsymbol{\iota}_{n \to m}$ to denote the zero-padding operator $\boldsymbol{\iota}_{n \to m} \boldsymbol{v} = \begin{bmatrix} \boldsymbol{v} \\ \boldsymbol{0}_{n-m} \end{bmatrix}$, which zero-pads a length $n$ vector $\boldsymbol{v} \in \mathbb{R}^n$ to length $m$ ($n \leq m$). Correspondingly, its adjoint operator $\boldsymbol{\iota}_{n \to m}^*$ denotes the restriction of a vector to its first $n$ coordinate (and $\boldsymbol{\iota}_{n \to m}^* = \boldsymbol{\iota}_{m \to n}$). Similarly, given a subset $\mathcal{I} \subseteq [m]$ and a vector $\boldsymbol{v} \in \mathbb{R}^{|\mathcal{I}|}$, we use $\boldsymbol{\iota}_{\mathcal{I} \to m} : \mathbb{R}^{|\mathcal{I}|} \mapsto \mathbb{R}^m$ to denote an operator that maps $\boldsymbol{v}$ to a zero-padded vector whose entries in $\mathcal{I}$ corresponding to those of $\boldsymbol{v}$.

We use $\mathcal{P}_{\boldsymbol{v}}$ and $\mathcal{P}_{\boldsymbol{v}^\perp}$ to denote projections onto $\boldsymbol{v}$ and its orthogonal complement, respectively. We let $\mathcal{P}_{\mathbb{S}^{n-1}}$ to be the $\ell^2$-normalization operator. To sum up, we have

$$\mathcal{P}_{\boldsymbol{v}^\perp}\boldsymbol{u} = \boldsymbol{u} - \frac{\boldsymbol{v}\boldsymbol{v}^\top}{\|\boldsymbol{v}\|^2}\boldsymbol{v}, \quad \mathcal{P}_{\boldsymbol{v}}\boldsymbol{u} = \frac{\boldsymbol{v}\boldsymbol{v}^\top}{\|\boldsymbol{v}\|^2}\boldsymbol{u}, \quad \mathcal{P}_{\mathbb{S}^{n-1}}\boldsymbol{v} = \frac{\boldsymbol{v}}{\|\boldsymbol{v}\|}.$$

**Circular convolution and circulant matrices.**  The convolution operator $\circledast$ is *circular* with modulo-$m$: $(\boldsymbol{a} \circledast \boldsymbol{x})_i = \sum_{j=0}^{m-1} a_j x_{i-j}$, and we use $\boxast$ to specify the *circular* convolution in 2D. For a vector $\boldsymbol{v} \in \mathbb{R}^m$, let $\mathrm{s}_\ell[\boldsymbol{v}]$ denote the cyclic shift of $\boldsymbol{v}$ with length $\ell$. Thus, we can introduce the circulant matrix $\boldsymbol{C}_{\boldsymbol{v}} \in \mathbb{R}^{m \times m}$ generated through $\boldsymbol{v} \in \mathbb{R}^m$,

$$\boldsymbol{C}_{\boldsymbol{v}} = \begin{bmatrix} v_1 & v_m & \cdots & v_3 & v_2 \\ v_2 & v_1 & v_m & & v_3 \\ \vdots & v_2 & v_1 & \ddots & \vdots \\ v_{m-1} & & \ddots & \ddots & v_m \\ v_m & v_{m-1} & \cdots & v_2 & v_1 \end{bmatrix} = \begin{bmatrix} \mathrm{s}_0\left[\boldsymbol{v}\right] & \mathrm{s}_1\left[\boldsymbol{v}\right] & \cdots & \mathrm{s}_{m-1}\left[\boldsymbol{v}\right] \end{bmatrix}. \quad (16)$$

Now the circulant convolution can also be written in a simpler matrix-vector product form. For instance, for any $\boldsymbol{u} \in \mathbb{R}^m$ and $\boldsymbol{v} \in \mathbb{R}^n$ ($n \leq m$),

$$\boldsymbol{u} \circledast \boldsymbol{v} = \boldsymbol{C}_{\boldsymbol{u}} \cdot \boldsymbol{\iota}_{n \to m}\boldsymbol{v} = \boldsymbol{C}_{\boldsymbol{\iota}_{n \to m}\boldsymbol{v}} \cdot \boldsymbol{u}.$$

In addition, the correlation between $\boldsymbol{u}$ and $\boldsymbol{v}$ can be also written in a similar form of convolution operator which reverses one vector before convolution. Let $\check{\boldsymbol{v}}$ denote a *cyclic reversal* of $\boldsymbol{v} \in \mathbb{R}^m$,

i.e., $\breve{\boldsymbol{v}} = [v_1, v_m, v_{m-1}, \cdots, v_2]^{\top}$, and define two correlation matrices $\boldsymbol{C}_{\boldsymbol{v}}^* \boldsymbol{e}_j = \mathrm{s}_j[\boldsymbol{v}]$ and $\breve{\boldsymbol{C}}_{\boldsymbol{v}} \boldsymbol{e}_j = \mathrm{s}_{-j}[\boldsymbol{v}]$. The two operators satisfy

$$\boldsymbol{C}_{\iota_{n \to m} \boldsymbol{v}}^* \boldsymbol{u} = \breve{\boldsymbol{v}} \circledast \boldsymbol{u}, \quad \breve{\boldsymbol{C}}_{\iota_{n \to m} \boldsymbol{v}} \boldsymbol{u} = \boldsymbol{v} \circledast \breve{\boldsymbol{u}}.$$

**Notation for several distributions.** We use $i.i.d.$ to denote *identically* and *independently distributed* random variables:

- we use $\mathcal{N}(\mu, \sigma^2)$ to denote Gaussian distribution with mean $\mu$ and variance $\sigma^2$;
- we use $\mathcal{U}(\mathbb{S}^{n-1})$ to denote the uniform distribution over the sphere $\mathbb{S}^{n-1}$;
- we use $\mathcal{B}(\theta)$ to denote the Bernoulli distribution with parameter $\theta$ controling the nonzero probability;
- we use $\mathcal{BG}(\theta)$ to denote Bernoulli-Gaussian distribution, i.e., if $u \sim \mathcal{BG}(\theta)$, then $u = b \cdot g$ with $b \sim \mathcal{B}(\theta)$ and $g \sim \mathcal{N}(0, 1)$;
- we use $\mathcal{BR}(\theta)$ to denote Bernoulli-Rademacher distribution, i.e., if $u \sim \mathcal{BR}(\theta)$, then $u = b \cdot r$ with $b \sim \mathcal{B}(\theta)$ and $r$ follows Rademacher distribution.

## A.2  Problem Reduction

In the following sections of the appendices, we study the optimization of

$$\min_{\boldsymbol{q}} \ \varphi_h(\boldsymbol{q}) \ := \ \frac{1}{np} \sum_{i=1}^{p} H_{\mu} \left( \boldsymbol{C}_{\boldsymbol{y}_i} \boldsymbol{P} \boldsymbol{q} \right), \qquad \text{s.t.} \quad \boldsymbol{q} \in \mathbb{S}^{n-1},$$

where

$$\boldsymbol{P} \ = \ \left( \frac{1}{\theta np} \sum_{i=1}^{p} \boldsymbol{C}_{\boldsymbol{y}_i}^{\top} \boldsymbol{C}_{\boldsymbol{y}_i} \right)^{-1/2}.$$

We simplify the problem by a change of variable $\overline{\boldsymbol{q}} = \boldsymbol{Q} \boldsymbol{q}$, which rotates the space by the orthogonal matrix $\boldsymbol{Q}$ in (7). Since the rotation $\boldsymbol{Q}$ does not change the optimization landscape, by an abuse of notation of $\boldsymbol{q}$ and $\overline{\boldsymbol{q}}$, we can rewrite the problem (5) as

$$\min_{\boldsymbol{q}} \ f(\boldsymbol{q}) \ := \ \frac{1}{np} \sum_{i=1}^{p} H_{\mu} \left( \boldsymbol{C}_{\boldsymbol{x}_i} \boldsymbol{R} \boldsymbol{Q}^{-1} \boldsymbol{q} \right), \qquad \text{s.t.} \quad \|\boldsymbol{q}\| \ = \ 1, \tag{17}$$

where

$$\boldsymbol{R} \ = \ \boldsymbol{C}_{\boldsymbol{a}} \left( \frac{1}{\theta np} \sum_{i=1}^{p} \boldsymbol{C}_{\boldsymbol{y}_i}^{\top} \boldsymbol{C}_{\boldsymbol{y}_i} \right)^{-1/2}, \quad \boldsymbol{Q} \ = \ \boldsymbol{C}_{\boldsymbol{a}} \left( \boldsymbol{C}_{\boldsymbol{a}}^{\top} \boldsymbol{C}_{\boldsymbol{a}} \right)^{-1/2},$$

and

$$\boldsymbol{R} \boldsymbol{Q}^{-1} \ = \ \boldsymbol{C}_{\boldsymbol{a}} \left( \frac{1}{\theta np} \sum_{i=1}^{p} \boldsymbol{C}_{\boldsymbol{y}_i}^{\top} \boldsymbol{C}_{\boldsymbol{y}_i} \right)^{-1/2} \left( \boldsymbol{C}_{\boldsymbol{a}}^{\top} \boldsymbol{C}_{\boldsymbol{a}} \right)^{1/2} \boldsymbol{C}_{\boldsymbol{a}}^{-1}.$$

For the reduction from the original problem to (17), we used the fact that $\boldsymbol{C}_{\boldsymbol{y}_i} \boldsymbol{P} = \boldsymbol{C}_{\boldsymbol{x}_i} \boldsymbol{R}$ in (7). Moreover, since $\boldsymbol{R} \approx \boldsymbol{Q}$ is *near orthogonal*, by assuming $\boldsymbol{R} \boldsymbol{Q}^{-1} = \boldsymbol{I}$ we can further reduce (17) to

$$\min_{\boldsymbol{q}} \ \widetilde{f}(\boldsymbol{q}) \ = \ \frac{1}{np} \sum_{i=1}^{p} H_{\mu} \left( \boldsymbol{C}_{\boldsymbol{x}_i} \boldsymbol{q} \right), \quad \text{s.t.} \quad \|\boldsymbol{q}\| = 1. \tag{18}$$

The objective (10) is simpler and much easier for analysis. By a similar analysis as [22, 32], it can be shown that asymptotically the landscape is highly symmetric and the standard basis vectors $\{\pm \boldsymbol{e}_i\}_{i=1}^{n}$ are the only global minimizers.

For the following sections of the appendices, without loss of generaltiy, we study the optimization landscape of $f(\boldsymbol{q})$ over the sphere, and proving global convergence of vanilla Riemannian gradient descent methods. We will show that $\boldsymbol{R} \boldsymbol{Q}^{-1} \approx \boldsymbol{I}$, so that we can study the landscape of $f(\boldsymbol{q})$ via studying the landscape of $\widetilde{f}(\boldsymbol{q})$ followed by a perturbation analysis.

## B Geometry: Main Results

In this part of appendix, we study the optimization landscape of $f(\boldsymbol{q})$ over regions

$$\mathcal{S}_\xi^{i\pm} := \left\{ \boldsymbol{q} \in \mathbb{S}^{n-1} \;\middle|\; \frac{|q_i|}{\|\boldsymbol{q}_{-i}\|_\infty} \geqslant \sqrt{1+\xi},\; q_i \gtrless 0 \right\}, \quad \xi \in [0, \infty).$$

We will show that the function $f(\boldsymbol{q})$ over each one of $\mathcal{S}_\xi^{i\pm}$ has benign first-order geometric structure, which enables efficient optimization via vanilla Riemannian gradient descent methods.

**Proposition B.1 (Regularity condition)** *Suppose $\theta \geqslant \frac{1}{n}$ and $\mu \leqslant c_0 \min\left\{\theta, \frac{1}{\sqrt{n}}\right\}$. There exists some numerical constant $\gamma \in (0,1)$, when the sample complexity*

$$p \geqslant C \max\left\{ n, \frac{\kappa^8}{\theta\mu^2\sigma_{\min}^2} \log^4 n \right\} \xi^{-2}\theta^{-2}n^4 \log\left(\frac{\theta n}{\mu}\right),$$

*with probability at least $1 - n^{-c_1} - c_2 np^{-c_3 n\theta}$ over the randomness of $\{\boldsymbol{x}_i\}_{i=1}^p$, we have*

$$\langle \operatorname{grad} f(\boldsymbol{q}), q_i\boldsymbol{q} - \boldsymbol{e}_i \rangle \geqslant c_4\theta(1-\theta)q_i \|\boldsymbol{q} - \boldsymbol{e}_i\|, \quad \sqrt{1-q_i^2} \in [\mu,\; \gamma], \tag{19}$$

$$\langle \operatorname{grad} f(\boldsymbol{q}), q_i\boldsymbol{q} - \boldsymbol{e}_i \rangle \geqslant c_4\theta(1-\theta)q_i n^{-1} \|\boldsymbol{q} - \boldsymbol{e}_i\|, \quad \sqrt{1-q_i^2} \in \left[\gamma,\; \sqrt{\frac{n-1}{n}}\right], \tag{20}$$

*holds for any $\boldsymbol{q} \in \mathcal{S}_\xi^{i+}$ and each index $i \in [n]$. Here, $c_0$, $c_1$, $c_2$, $c_3$, $c_4$, and $C$ are positive numerical constants.*

**Proof** Without loss of generality, it is enough to consider the case $i = n$. For all $\boldsymbol{q} \in \mathcal{S}_\xi^{n+}$, we have

$$\begin{aligned}
&\langle \operatorname{grad} f(\boldsymbol{q}), q_n\boldsymbol{q} - \boldsymbol{e}_n \rangle \\
&= \left\langle \operatorname{grad} f(\boldsymbol{q}) - \operatorname{grad} \widetilde{f}(\boldsymbol{q}) + \operatorname{grad} \widetilde{f}(\boldsymbol{q}) - \operatorname{grad} \mathbb{E}\left[\widetilde{f}(\boldsymbol{q})\right] + \operatorname{grad} \mathbb{E}\left[\widetilde{f}(\boldsymbol{q})\right], q_n\boldsymbol{q} - \boldsymbol{e}_n \right\rangle \\
&\geqslant \left\langle \operatorname{grad} \mathbb{E}\left[\widetilde{f}(\boldsymbol{q})\right], q_n\boldsymbol{q} - \boldsymbol{e}_n \right\rangle - \left|\left\langle \operatorname{grad} f(\boldsymbol{q}) - \operatorname{grad} \widetilde{f}(\boldsymbol{q}), q_n\boldsymbol{q} - \boldsymbol{e}_n \right\rangle\right| \\
&\quad - \left|\left\langle \operatorname{grad} \widetilde{f}(\boldsymbol{q}) - \operatorname{grad} \mathbb{E}\left[\widetilde{f}(\boldsymbol{q})\right], q_n\boldsymbol{q} - \boldsymbol{e}_n \right\rangle\right|.
\end{aligned}$$

From Proposition E.1, when $\theta \geqslant \frac{1}{n}$ and $\mu \leqslant c_0 \min\left\{\theta, \frac{1}{\sqrt{n}}\right\}$, we know that in the worst case scenario,

$$\left\langle \operatorname{grad} \mathbb{E}\left[\widetilde{f}(\boldsymbol{q})\right], q_n\boldsymbol{q} - \boldsymbol{e}_n \right\rangle \geqslant c_1\theta(1-\theta)\xi n^{-3/2} \|\boldsymbol{q}_{-n}\|$$

holds for all $\boldsymbol{q} \in \mathcal{S}_\xi^{n+}$. On the other hand, by Corollary G.2, when $p \geqslant C_1\theta^{-2}\xi^{-2}n^5 \log\left(\frac{\theta n}{\mu}\right)$, we have

$$\begin{aligned}
\left|\left\langle \operatorname{grad} \widetilde{f}(\boldsymbol{q}) - \operatorname{grad} \mathbb{E}\left[\widetilde{f}(\boldsymbol{q})\right], q_n\boldsymbol{q} - \boldsymbol{e}_n \right\rangle\right| &\leqslant \left\|\operatorname{grad} \widetilde{f}(\boldsymbol{q}) - \operatorname{grad} \mathbb{E}\left[\widetilde{f}(\boldsymbol{q})\right]\right\| \|q_n\boldsymbol{q} - \boldsymbol{e}_n\| \\
&\leqslant \frac{c_1}{3}\theta(1-\theta)\xi n^{-3/2} \|q_n\boldsymbol{q} - \boldsymbol{e}_n\|
\end{aligned}$$

holds for all $\boldsymbol{q} \in \mathcal{S}_\xi^{n+}$ with probability at least $1 - np^{-c_2\theta n} - n\exp\left(-c_3 n^2\right)$. Moreover, from Proposition H.1, we know that when $p \geqslant C\frac{\kappa^8 n^4}{\mu^2\theta^3\sigma_{\min}^2\xi^2} \log^4 n \log\left(\frac{\theta n}{\mu}\right)$

$$\begin{aligned}
\left|\left\langle \operatorname{grad} f(\boldsymbol{q}) - \operatorname{grad} \widetilde{f}(\boldsymbol{q}), q_n\boldsymbol{q} - \boldsymbol{e}_n \right\rangle\right| &\leqslant \|q_n\boldsymbol{q} - \boldsymbol{e}_n\| \cdot \left\|\operatorname{grad} f(\boldsymbol{q}) - \operatorname{grad} \widetilde{f}(\boldsymbol{q})\right\| \\
&\leqslant \frac{c_1}{3}\theta(1-\theta)\xi n^{-3/2} \|q_n\boldsymbol{q} - \boldsymbol{e}_n\|
\end{aligned}$$

holds for all $\boldsymbol{q} \in \mathcal{S}_\xi^{n+}$ with probability at least $1 - c_4 p^{-c_5 n\theta} - n^{-c_6} - ne^{-c_7\theta np}$. By combining all the bounds above, we obtain the desired result. ∎

**Proposition B.2 (Negative curvature on the gradient)** *Suppose $\theta \geqslant \frac{1}{n}$ and $\mu \leqslant \frac{c_0}{\sqrt{n}}$. For any index $i \in [n]$, when the sample*

$$p \geqslant C \max \left\{ n, \frac{\kappa^8}{\theta \mu^2 \sigma_{\min}^2} \log^4 n \right\} \xi^{-2} \theta^{-2} n^4 \log \left( \frac{\theta n}{\mu} \right),$$

*with probability at least $1 - n^{-c_1} - c_2 n p^{-c_3 n \theta}$ over the randomness of $\{x_i\}_{i=1}^p$, we have*

$$\left\langle \operatorname{grad} f(\boldsymbol{q}), \frac{1}{q_j} \boldsymbol{e}_j - \frac{1}{q_i} \boldsymbol{e}_i \right\rangle \geqslant c_4 \frac{\theta(1 - \theta)}{n} \frac{\xi}{1 + \xi}, \tag{21}$$

*holds for all $\boldsymbol{q} \in \mathcal{S}_\xi^{i+}$ and any $q_j$ such that $j \neq i$ and $q_j^2 \geqslant \frac{1}{3} q_i^2$. Here, $c_0$, $c_1$, $c_2$, $c_3$, $c_4$, and $C$ are positive numerical constants.*

**Proof** Without loss of generality, it is enough to consider the case $i = n$. For all $\boldsymbol{q} \in \mathcal{S}_\xi^{n+}$, we have

$$\left\langle \operatorname{grad} f(\boldsymbol{q}), \frac{1}{q_j} \boldsymbol{e}_j - \frac{1}{q_n} \boldsymbol{e}_n \right\rangle$$

$$= \left\langle \operatorname{grad} f(\boldsymbol{q}) - \operatorname{grad} \widetilde{f}(\boldsymbol{q}) + \operatorname{grad} \widetilde{f}(\boldsymbol{q}) - \operatorname{grad} \mathbb{E}\left[\widetilde{f}(\boldsymbol{q})\right] + \operatorname{grad} \mathbb{E}\left[\widetilde{f}(\boldsymbol{q})\right], \frac{1}{q_j} \boldsymbol{e}_j - \frac{1}{q_n} \boldsymbol{e}_n \right\rangle$$

$$\geqslant \left\langle \operatorname{grad} \mathbb{E}\left[\widetilde{f}(\boldsymbol{q})\right], \frac{1}{q_j} \boldsymbol{e}_j - \frac{1}{q_n} \boldsymbol{e}_n \right\rangle - \left| \left\langle \operatorname{grad} f(\boldsymbol{q}) - \operatorname{grad} \widetilde{f}(\boldsymbol{q}), \frac{1}{q_j} \boldsymbol{e}_j - \frac{1}{q_n} \boldsymbol{e}_n \right\rangle \right|$$

$$- \left| \left\langle \operatorname{grad} \widetilde{f}(\boldsymbol{q}) - \operatorname{grad} \mathbb{E}\left[\widetilde{f}(\boldsymbol{q})\right], \frac{1}{q_j} \boldsymbol{e}_j - \frac{1}{q_n} \boldsymbol{e}_n \right\rangle \right|.$$

From Proposition F.1, when $\theta \geqslant \frac{1}{n}$ and $\mu \leqslant \frac{c_0}{\sqrt{n}}$, we know that

$$\left\langle \operatorname{grad} \mathbb{E}\left[\widetilde{f}(\boldsymbol{q})\right], \frac{1}{q_j} \boldsymbol{e}_j - \frac{1}{q_n} \boldsymbol{e}_n \right\rangle \geqslant \frac{\theta(1 - \theta)}{4n} \frac{\xi}{1 + \xi}$$

holds for all $\boldsymbol{q} \in \mathcal{S}_\xi^{n+}$ and any $q_j$ such that $q_j^2 \geqslant \frac{1}{3} q_i^2$. On the other hand, by Corollary G.2, when $p \geqslant C_1 \theta^{-2} \xi^{-2} n^5 \log \left( \frac{\theta n}{\mu} \right)$, we have

$$\left| \left\langle \operatorname{grad} \widetilde{f}(\boldsymbol{q}) - \operatorname{grad} \mathbb{E}\left[\widetilde{f}(\boldsymbol{q})\right], \frac{1}{q_j} \boldsymbol{e}_j - \frac{1}{q_n} \boldsymbol{e}_n \right\rangle \right| \leqslant \left\| \operatorname{grad} \widetilde{f}(\boldsymbol{q}) - \operatorname{grad} \mathbb{E}\left[\widetilde{f}(\boldsymbol{q})\right] \right\| \cdot \left\| \frac{1}{q_j} \boldsymbol{e}_j - \frac{1}{q_n} \boldsymbol{e}_n \right\|$$

$$\leqslant \frac{\theta(1 - \theta)}{12n} \frac{\xi}{1 + \xi}$$

holds for all $\boldsymbol{q} \in \mathcal{S}_\xi^{n+}$ with probability at least $1 - np^{-c_2 \theta n} - n \exp\left(-c_3 n^2\right)$. For the last inequality, we used the fact that

$$\left\| \frac{1}{q_j} \boldsymbol{e}_j - \frac{1}{q_n} \boldsymbol{e}_n \right\| = \sqrt{\frac{1}{q_j^2} + \frac{1}{q_n^2}} \leqslant 2\sqrt{n}.$$

Moreover, from Proposition H.1, we know that when $p \geqslant C \frac{\kappa^8 n^4}{\mu^2 \theta^3 \sigma_{\min}^2 \xi^2} \log^4 n \log \left( \frac{\theta n}{\mu} \right)$

$$\left| \left\langle \operatorname{grad} f(\boldsymbol{q}) - \operatorname{grad} \widetilde{f}(\boldsymbol{q}), q_n \boldsymbol{q} - \boldsymbol{e}_n \right\rangle \right| \leqslant \left\| \operatorname{grad} f(\boldsymbol{q}) - \operatorname{grad} \widetilde{f}(\boldsymbol{q}) \right\| \cdot \left\| \frac{1}{q_j} \boldsymbol{e}_j - \frac{1}{q_n} \boldsymbol{e}_n \right\|$$

$$\leqslant \frac{\theta(1 - \theta)}{12n} \frac{\xi}{1 + \xi}$$

holds for all $\boldsymbol{q} \in \mathcal{S}_\xi^{n+}$ with probability at least $1 - c_4 p^{-c_5 n \theta} - n^{-c_6} - n e^{-c_7 \theta n p}$. By combining all the bounds above, we obtain the desired result. ∎

**Proposition B.3 (Bounded gradient)** *Suppose $\theta \geqslant \frac{1}{n}$ and $\mu \leqslant \frac{c_0}{\sqrt{n}}$. For any index $i \in [n]$, when the sample*

$$p \geqslant C \max\left\{ n, \frac{\kappa^8}{\theta\mu^2\sigma_{\min}^2} \log^4 n \right\} \theta^{-2} n \log\left(\frac{\theta n}{\mu}\right),$$

*with probability at least $1 - n^{-c_1} - c_2 n p^{-c_3 n\theta}$ over the randomness of $\{\boldsymbol{x}_i\}_{i=1}^p$, we have*

$$|\langle \operatorname{grad} f(\boldsymbol{q}), \boldsymbol{e}_i \rangle| \;\leqslant\; 2, \tag{22}$$

$$\|\operatorname{grad} f(\boldsymbol{q})\| \;\leqslant\; 2\sqrt{\theta n}. \tag{23}$$

*holds for all $\boldsymbol{q} \in \mathbb{S}^{n-1}$ and any index $i \in [n]$. Here, $c_0$, $c_1$, $c_2$, $c_3$ and $C$ are positive numerical constants.*

**Proof** For any index $i \in [n]$, we have

$$\sup_{q\in\mathbb{S}^{n-1}} |\langle \operatorname{grad} f(\boldsymbol{q}), \boldsymbol{e}_i \rangle| \;\leqslant\; \sup_{q\in\mathbb{S}^{n-1}} \left|\left\langle \operatorname{grad} \widetilde{f}(\boldsymbol{q}), \boldsymbol{e}_i \right\rangle\right| + \sup_{q\in\mathbb{S}^{n-1}} \left|\left\langle \operatorname{grad} f(\boldsymbol{q}) - \operatorname{grad} \widetilde{f}(\boldsymbol{q}), \boldsymbol{e}_i \right\rangle\right|$$

$$\leqslant\; \sup_{q\in\mathbb{S}^{n-1}} \left|\left\langle \operatorname{grad} \widetilde{f}(\boldsymbol{q}), \boldsymbol{e}_i \right\rangle\right| + \left\|\operatorname{grad} f(\boldsymbol{q}) - \operatorname{grad} \widetilde{f}(\boldsymbol{q})\right\|.$$

By Corollary G.3, when $p \geqslant C_1 n \log\left(\frac{\theta n}{\mu}\right)$, we have

$$\sup_{q\in\mathbb{S}^{n-1}} \left|\left\langle \operatorname{grad} \widetilde{f}(\boldsymbol{q}), \boldsymbol{e}_i \right\rangle\right| \;\leqslant\; \frac{3}{2}$$

holds for any index $i \in [n]$ with probability at least $1 - np^{-c_1\theta n} - n\exp(-c_2 p)$. On the other hand, Proposition H.1 implies that, when $p \geqslant C_2 \frac{\kappa^8 n}{\mu^2\theta\sigma_{\min}^2} \log^4 n \log\left(\frac{\theta n}{\mu}\right)$,, we have

$$\left\|\operatorname{grad} f(\boldsymbol{q}) - \operatorname{grad} \widetilde{f}(\boldsymbol{q})\right\| \;\leqslant\; \frac{1}{2},$$

holds with probability at least $1 - c_3 p^{-c_4 n\theta} - n^{-c_5} - n e^{-c_6\theta np}$. Combining the bounds above gives (22). The bound (23) can be proved in a similar fashion. ∎

# C    Convergence Analysis

In this section, we show linear convergence of *vanilla* gradient descent to target solutions. Firstly, for Huber loss, we prove that the gradient descent method converges to an approximate solution in polynomial steps. Second, we show linear convergence of subgradient method to the target solution, which solves phase-2 LP rounding problem.

Our analysis is based on based on the geometric properties of the optimization landscape showed in Appendix B. Namely, our following proofs are based on the results in Proposition B.1, Proposition B.2, and Proposition B.3 (i.e., the equations (19), (20), (22), and (23)) holding for the rest of this section.

## C.1    Proof of linear convergence for Algorithm 1

First, assuming the geometric properties in Appendix B hold, we show that starting from a random initialization, the gradient descent method optimizing

$$\min_{\boldsymbol{q}} \; f(\boldsymbol{q}) \;=\; \frac{1}{np} \sum_{i=1}^{p} H_\mu\left(\boldsymbol{C}_{\boldsymbol{x}_i} \boldsymbol{R} \boldsymbol{Q}^{-1} \boldsymbol{q}\right), \qquad \text{s.t.} \quad \boldsymbol{q} \in \mathbb{S}^{n-1} \tag{24}$$

recovers an approximate solution in polynomial steps.

491 **Theorem C.1 (Linear convergence of Algorithm 1)** *Given an initialization $q^{(0)} \sim \mathcal{U}(\mathbb{S}^{n-1})$ uni-*
492 *form random drawn from the sphere, choose a stepsize*

$$\tau = c \min \left\{ \frac{1}{n^{5/2}}, \frac{\mu}{n} \right\},$$

493 *then the vanilla gradient descent method for (5) produces a solution*

$$\left\| q^{(k)} - e_i \right\| \leqslant 2\mu$$

494 *for some $i \in [n]$, whenever*

$$k \geqslant K := \frac{C}{\theta} \max \left\{ n^4, \frac{n^{5/2}}{\mu} \right\} \log \left( \frac{1}{\mu} \right).$$

495 **Proof** [Proof of Theorem C.1]

496 **Initialization and iterate stays within the region.** First, from Lemma C.3, we know that when
497 $\xi = \frac{1}{5 \log n}$, with probability at least $1/2$, our random initialization $q^{(0)}$ falls into one of the sets
498 $\left\{ \mathcal{S}_\xi^{1+}, \mathcal{S}_\xi^{1-}, \ldots, \mathcal{S}_\xi^{n+}, \mathcal{S}_\xi^{n-} \right\}$. Without loss of generality, we assume that $q^{(0)} \in \mathcal{S}_\xi^{n+}$.

499 Once $q^{(0)}$ initialized within the region $\mathcal{S}_\xi^{n+}$, from Lemma C.4, whenever the stepsize $\tau \leqslant c_0/\sqrt{n}$,
500 we know that our gradient descent stays within the region $\mathcal{S}_\xi^{n+}$ when the stepsize $\tau \leqslant c_1/\sqrt{n}$ for
501 some $c_1 > 0$. Based on this, to complete the proof, we now proceed by proving the following results.

502 **Linear convergence until reaching $\|q - e_n\| \leqslant \mu$.** From Proposition B.1, there exists some
503 numerical constant $\gamma \in (\mu, 1)$, such that the regularity condition

$$\langle \operatorname{grad} f(q), q_n q - e_n \rangle \geqslant \underbrace{c_2 \theta(1-\theta) n^{-3/2}}_{\alpha_1} \cdot \|q - e_n\|, \quad \sqrt{1-q_n^2} \in \left[ \gamma, \sqrt{\frac{n-1}{n}} \right], \quad (25)$$

$$\langle \operatorname{grad} f(q), q_n q - e_n \rangle \geqslant \underbrace{c_2' \theta(1-\theta)}_{\alpha_2} \cdot \|q - e_n\|, \quad \sqrt{1-q_n^2} \in [\mu, \gamma], \quad (26)$$

504 holds w.h.p. for all $q \in \mathcal{S}_\xi^{n+}$. As $\alpha_2 \geqslant \alpha_1$, the regularity condition holds for all $q$ with $\alpha = \alpha_1$.
505 Select a stepsize $\tau$ such that $\tau \leqslant \gamma \frac{\alpha_1}{2\sqrt{2}\theta n}$. By Lemma C.5 and the regularity condition (25), we have

$$\left\| q^{(k)} - e_n \right\|^2 - \frac{\gamma^2}{2} \leqslant (1 - \tau\alpha_1)^k \left[ \left\| q^{(0)} - e_n \right\|^2 - \frac{\gamma^2}{2} \right] \leqslant 2 (1 - \tau\alpha_1)^k,$$

506 where the last inequality utilizes the fact that $\left\| q^{(0)} - e_n \right\|^2 \leqslant 2$. This further implies that

$$1 - q_n^2 \leqslant \left\| q^{(k)} - e_n \right\|^2 \leqslant \frac{\gamma^2}{2} + 2 (1 - \tau\alpha_1)^k \leqslant \gamma^2,$$

507 when

$$2 (1 - \tau\alpha_1)^k \leqslant \frac{\gamma^2}{2} \implies k \geqslant K_1 := \frac{\log \left( \gamma^2/4 \right)}{\log \left( 1 - \tau\alpha_1 \right)}.$$

508 This implies that $\sqrt{1-q_n^2} \leqslant \gamma$ for $\forall \ k \geqslant K_1$. Thus, from (26), we know that the regularity condition
509 holds with $\alpha = \alpha_2$. Choose stepsize $\tau \leqslant \frac{\mu\alpha_2}{2\sqrt{2}\theta n}$, apply Lemma C.5 again with $\alpha = \alpha_2$, for all $k \geqslant 1$,
510 we have

$$\left\| q^{(K_1+k)} - e_n \right\|^2 - \frac{\mu^2}{2} \leqslant (1 - \tau\alpha_2)^k \left( \left\| q^{(0)} - e_n \right\|^2 - \frac{\mu^2}{2} \right) \leqslant \left( \gamma^2 - \mu^2 \right) (1 - \tau\alpha_2)^k.$$

511 This further implies that

$$\left\| q^{(K_1+k)} - e_n \right\|^2 \leqslant \frac{\mu^2}{2} + \left( \gamma^2 - \frac{\mu^2}{2} \right) (1 - \tau\alpha_2)^k \leqslant \mu^2$$

whenever

$$\left(\gamma^2 - \frac{\mu^2}{2}\right)(1 - \tau\alpha_2)^k \leqslant \frac{\mu^2}{2} \quad \Longrightarrow \quad k \geqslant K_2 := \frac{\log\left(\mu^2/\left(2\gamma^2 - \mu^2\right)\right)}{\log\left(1 - \tau\alpha_2\right)}.$$

Therefore, combining the results above, by using the fact that $\alpha_1 = c_2\theta(1 - \theta)n^{-3/2}$ and $\alpha_2 = c_2'\theta(1 - \theta)$, we have $\left\|q^{(k)} - e_n\right\| \leqslant \mu$ whenever

$$\tau \leqslant \min\left\{\frac{\gamma\alpha_1}{2\sqrt{2\theta n}}, \frac{\mu\alpha_2}{2\sqrt{2\theta n}}\right\} = C\min\left\{\frac{1}{n^{5/2}}, \frac{\mu}{n}\right\}$$

and $k \geqslant K := K_1 + K_2$ with

$$K = \frac{\log\left(4/\gamma^2\right)}{\log\left((1 - \tau\alpha_1)^{-1}\right)} + \frac{\log\left(\left(2\gamma^2 - \mu^2\right)/\mu^2\right)}{\log\left((1 - \tau\alpha_2)^{-1}\right)}$$

$$\leqslant \frac{c_3}{\tau\alpha_1} + \frac{c_4}{\tau\alpha_2}\log\left(\frac{1}{\mu}\right) \leqslant \frac{c_5}{\theta}\max\left\{n^4, \frac{n^{5/2}}{\mu}\right\}\log\left(\frac{1}{\mu}\right),$$

where we used the fact that $\log^{-1}\left((1 - x)^{-1}\right) \leqslant 2/x$ for small $x$.

**No jump away from an approximate solution $e_n$.** Finally, we show that once our iterate reaches the region

$$\mathcal{S} := \left\{q \in \mathbb{S}^{n-1} \mid \|q - e_n\| \leqslant 2\mu\right\},$$

it will stay within the region $\mathcal{S}$, such that our final iterates will always stay close to an approximate solution $e_n$. Towards this end, suppose $q^{(k)} \in \mathcal{S}$. Therefore two possibilities: (i) $\mu \leqslant \left\|q^{(k)} - e_n\right\| \leqslant 2\mu$ (ii) $\left\|q^{(k)} - e_n\right\| \leqslant \mu$. If the case (i) holds, then our argument above implies that $\left\|q^{(k+1)} - e_n\right\| \leqslant \left\|q^{(k)} - e_n\right\| \leqslant 2\mu$. Otherwise $\left\|q^{(k)} - e_n\right\| \leqslant \mu$, for which we have

$$\left\|q^{(k+1)} - e_n\right\| \leqslant \left\|q^{(k)} - \tau\operatorname{grad} f(q) - e_n\right\|$$

$$\leqslant \left\|q^{(k)} - e_n\right\| + \tau\|\operatorname{grad} f(q)\| \leqslant \mu + 2\tau\sqrt{\theta n} \leqslant 2\mu,$$

where we used the fact that $\tau \leqslant \frac{\mu}{\sqrt{\theta n}}$. Thus, by induction, we have $q^{(k')} \in \mathcal{S}$ for all future iterates $k' = k + 1, k + 2, \cdots$. This completes the proof. ∎

**Lemma C.2** *For any $q \in \mathcal{S}_\xi^{n+}$, we have*

$$1 - q_n^2 \leqslant \|q - e_n\|^2 \leqslant 2\left(1 - q_n^2\right) \leqslant 2.$$

**Proof** We have

$$1 - q_n^2 \leqslant \|q - e_n\|^2 = \|q_{-n}\|^2 + (1 - q_n)^2\|e_n\|^2 = 2(1 - q_n) = 2\frac{1 - q_n^2}{1 + q_n^2} \leqslant 2(1 - q_n^2)$$

as desired. ∎

**Lemma C.3 (Random initialization falls into good region)** *Let $q^{(0)} \sim \mathcal{U}(\mathbb{S}^{n-1})$ be uniformly random generated from the unit sphere $\mathbb{S}^{n-1}$. When $\xi = \frac{1}{5\log n}$, then with probability at least $1/2$, $q^{(0)}$ belongs to one of the $2n$ sets $\left\{\mathcal{S}_\xi^{1+}, \mathcal{S}_\xi^{1-}, \ldots, \mathcal{S}_\xi^{n+}, \mathcal{S}_\xi^{n-}\right\}$. The set $q^{(0)}$ belongs to is uniformly at random.*

**Proof** We refer the readers to Lemma 3.9 of [23] and Theorem 1 of [22] for detailed proofs. ∎

**Lemma C.4 (Stay within the region $\mathcal{S}_\xi^{n+}$)** *Suppose $q^{(0)} \in \mathcal{S}_\xi^{n+}$ with $\xi \leqslant 1$. There exists some constant $c > 0$, such that when the stepsize satisfies $\tau \leqslant \frac{c}{\sqrt{n}}$, our Riemannian gradient iterate $q^{(k)} = \mathcal{P}_{\mathbb{S}^{n-1}}\left(q^{(k-1)} - \tau \cdot \operatorname{grad} f(q^{(k-1)})\right)$ satisfies $q^{(k)} \in \mathcal{S}_\xi^{n+}$ for all $k \geqslant 1$.*

**Proof** We prove this by induction. For any $k \geqslant 1$, suppose $\boldsymbol{q}^{(k)} \in \mathcal{S}_\xi^{n+}$. For convenience, let $\boldsymbol{g}^{(k)} = \operatorname{grad} f(\boldsymbol{q}^{(k)})$. Then, for any $j \neq k$, we have

$$
\left( \frac{q_n^{(k+1)}}{q_j^{(k+1)}} \right)^2 = \left( \frac{q_n^{(k)} - \tau g_n^{(k)}}{q_j^{(k)} - \tau g_j^{(k)}} \right)^2 .
$$

We proceed by considering the following two cases.

**Case (i):** $\left| q_n^{(k)} / q_j^{(k)} \right| \geqslant \sqrt{3}.$  In this case, we have

$$
\left( \frac{q_n^{(k+1)}}{q_j^{(k+1)}} \right)^2 = \left( \frac{q_n^{(k)} - \tau g_n^{(k)}}{q_j^{(k)} - \tau g_j^{(k)}} \right)^2 \geqslant \left( \frac{1 - \tau \cdot g_n^{(k)} / q_n^{(k)}}{q_j^{(k)} / q_n^{(k)} - \tau g_j^{(k)} / q_n^{(k)}} \right)^2 \geqslant \left( \frac{1 - 2\tau\sqrt{n}}{1/\sqrt{3} + 2\tau\sqrt{n}} \right)^2 \geqslant 2,
$$

where the second inequality utilizes (22) and the fact $q_n^{(k)} \geqslant \frac{1}{\sqrt{n}}$, and the last inequality follows when $\tau \leqslant \frac{\sqrt{3} - \sqrt{2}}{2(\sqrt{6} + \sqrt{3})} \frac{1}{\sqrt{n}}.$

**Case (ii):** $\left| q_n^{(k)} / q_j^{(k)} \right| \leqslant \sqrt{3}.$  Proposition B.1 and Proposition B.2 implies that

$$
\frac{g_j^{(k)}}{q_j^{(k)}} \geqslant 0, \quad \frac{g_j^{(k)}}{q_j^{(k)}} - \frac{g_n^{(k)}}{q_n^{(k)}} \geqslant 0. \tag{27}
$$

By noting that $\left| q_j^{(k)} \right| \geqslant \left| q_n^{(k)} \right| / \sqrt{3} \geqslant 1/\sqrt{3n}$ and $\left| g_j^{(k)} \right| \leqslant 2$, we have

$$
\tau \leqslant \frac{1}{2\sqrt{3n}} \leqslant \frac{q_j^{(k)}}{g_j^{(k)}} \quad \Longrightarrow \quad \tau \cdot \frac{g_j^{(k)}}{q_j^{(k)}} \leqslant 1. \tag{28}
$$

Thus, we have

$$
\begin{aligned}
\left( \frac{q_n^{(k+1)}}{q_j^{(k+1)}} \right)^2 &= \left( \frac{q_n^{(k)}}{q_j^{(k)}} \right)^2 \left( 1 + \tau \cdot \frac{g_j^{(k)} / q_j^{(k)} - g_n^{(k)} / q_n^{(k)}}{1 - \tau g_j^{(k)} / q_j^{(k)}} \right)^2 \\
&\geqslant \left( \frac{q_n^{(k)}}{q_j^{(k)}} \right)^2 \left( 1 + \tau \cdot \left( \frac{g_j^{(k)}}{q_j^{(k)}} - \frac{g_n^{(k)}}{q_n^{(k)}} \right) \right)^2 \geqslant \left( \frac{q_n^{(k)}}{q_j^{(k)}} \right)^2 \left( 1 + \tau \cdot \frac{\theta(1 - \theta)}{4n} \frac{\xi}{1 + \xi} \right)^2 .
\end{aligned}
$$

The first inequality follows from (27) and (28), and the second inequality directly follows from Proposition B.2. Therefore, when $\xi \leqslant 1$, this implies that $\boldsymbol{q}^{(k+1)} \in \mathcal{S}_\xi^{n+}$. By induction, this holds for all $k \geqslant 1$. ∎

In the following, we show that the iterates get closer to $\boldsymbol{e}_n$.

**Lemma C.5 (Iterate contraction)** *For any $\boldsymbol{q} \in \mathcal{S}_\xi^{n+}$, assuming the following regularity condition*

$$
\langle \operatorname{grad} f(\boldsymbol{q}), q_i \boldsymbol{q} - \boldsymbol{e}_n \rangle \geqslant \alpha \| \boldsymbol{q} - \boldsymbol{e}_n \| \tag{29}
$$

*holds for a parameter $\alpha > 0$. Then if $\boldsymbol{q}^{(k)} \in \mathcal{S}_\xi^{n+}$ and the stepsize $\tau \leqslant c \frac{\alpha}{\theta n}$, the iterate $\boldsymbol{q}^{(k+1)} = \mathcal{P}_{\mathbb{S}^{n-1}}(\boldsymbol{q} - \tau \cdot \operatorname{grad} f(\boldsymbol{q}))$ satisfies*

$$
\left\| \boldsymbol{q}^{(k+1)} - \boldsymbol{e}_n \right\|^2 - \left( \frac{2\tau\theta n}{\alpha} \right)^2 \leqslant (1 - \tau\alpha) \left[ \left\| \boldsymbol{q}^{(k)} - \boldsymbol{e}_n \right\|^2 - \left( \frac{2\tau\theta n}{\alpha} \right)^2 \right].
$$

**Proof** First, note that

$$
\begin{aligned}
\left\| \boldsymbol{q}^{(k+1)} - \boldsymbol{e}_n \right\|^2 &= \left\| \mathcal{P}_{\mathbb{S}^{n-1}} \left( \boldsymbol{q}^{(k)} - \tau \cdot \operatorname{grad} f(\boldsymbol{q}^{(k)}) \right) - \mathcal{P}_{\mathbb{S}^{n-1}}(\boldsymbol{e}_n) \right\|^2 \\
&\leqslant \left\| \boldsymbol{q}^{(k)} - \tau \cdot \operatorname{grad} f(\boldsymbol{q}^{(k)}) - \boldsymbol{e}_n \right\|^2 \\
&= \left\| \boldsymbol{q}^{(k)} - \boldsymbol{e}_n \right\|^2 - 2\tau \cdot \left\langle \operatorname{grad} f(\boldsymbol{q}^{(k)}), \boldsymbol{q}^{(k)} - \boldsymbol{e}_n \right\rangle + \tau^2 \left\| \operatorname{grad} f(\boldsymbol{q}^{(k)}) \right\|^2 \\
&\leqslant \left\| \boldsymbol{q}^{(k)} - \boldsymbol{e}_n \right\|^2 - 2\tau\alpha \left\| \boldsymbol{q}^{(k)} - \boldsymbol{e}_n \right\| + 4\tau^2 \theta n,
\end{aligned}
$$

where the first inequality utilizes the fact that $\mathcal{P}_{\mathbb{S}^{n-1}}(\cdot)$ is 1-Lipschitz continuous, and the last line follows from (29) and (23) in Proposition B.3. We now subtract both sides by $\left( \frac{2\tau\theta n}{\alpha} \right)^2$,

$$
\begin{aligned}
\left\| \boldsymbol{q}^{(k+1)} - \boldsymbol{e}_n \right\|^2 - \left( \frac{2\tau\theta n}{\alpha} \right)^2 &\leqslant \left\| \boldsymbol{q}^{(k)} - \boldsymbol{e}_n \right\|^2 - \left( \frac{2\tau\theta n}{\alpha} \right)^2 - 2\tau\alpha \left( \left\| \boldsymbol{q}^{(k)} - \boldsymbol{e}_n \right\| - \frac{2\tau\theta n}{\alpha} \right) \\
&= \left[ 1 - 2\tau\alpha \left( \left\| \boldsymbol{q}^{(k)} - \boldsymbol{e}_n \right\| + \frac{2\tau\theta n}{\alpha} \right)^{-1} \right] \left[ \left\| \boldsymbol{q}^{(k)} - \boldsymbol{e}_n \right\|^2 - \left( \frac{2\tau\theta n}{\alpha} \right)^2 \right] \\
&\leqslant (1 - \tau\alpha) \left[ \left\| \boldsymbol{q}^{(k)} - \boldsymbol{e}_n \right\|^2 - \left( \frac{2\tau\theta n}{\alpha} \right)^2 \right],
\end{aligned}
$$

where the last inequality follows because

$$
\left\| \boldsymbol{q}^{(k)} - \boldsymbol{e}_n \right\|^2 \leqslant 2, \quad \tau \leqslant \left( 1 - \frac{1}{\sqrt{2}} \right) \frac{\alpha}{\theta n},
$$

such that

$$
\left\| \boldsymbol{q} - \boldsymbol{e}_n \right\| + \frac{2\tau\theta n}{\alpha} \leqslant 2.
$$

This completes the proof. ∎

## C.2 Exact solution via LP rounding

To obtain exact solutions, we use the approximate solution $\boldsymbol{q}_\star$ from phase-1 gradient descent method as a warm start $\boldsymbol{r} = \boldsymbol{q}_\star$, and consider solving a *convex* phase-2 LP rounding problem,

$$
\min_{\boldsymbol{q}} \ \zeta(\boldsymbol{q}) := \frac{1}{np} \sum_{i=1}^{p} \left\| \boldsymbol{C}_{\boldsymbol{x}_i} \boldsymbol{R} \boldsymbol{Q}^{-1} \boldsymbol{q} \right\|_1, \quad \text{s.t.} \quad \langle \boldsymbol{r}, \boldsymbol{q} \rangle = 1.
$$

In the following, we show the function is sharp around the target solution, so that projected subgradient descent methods converge linearly to the truth.

### C.2.1 Sharpness of the objective function.

**Proposition C.6** *Suppose $\theta \in \left( \frac{1}{n}, \frac{1}{3} \right)$ and $\boldsymbol{r}$ satisfies*

$$
\frac{\| \boldsymbol{r}_{-n} \|}{r_n} \leqslant \frac{1}{20}. \tag{30}
$$

*Whenever $p \geqslant C \frac{\kappa^8}{\theta \sigma_{\min}^2(\boldsymbol{C_a})} \log^3 n$, with probability at least $1 - p^{-c_1 n\theta} - n^{-c_2}$, the function $\zeta(\boldsymbol{q})$ is sharp in a sense that*

$$
\zeta(\boldsymbol{q}) - \zeta \left( (\boldsymbol{R}\boldsymbol{Q}^{-1})^{-1} \frac{\boldsymbol{e}_n}{\widetilde{r}_n} \right) \geqslant \frac{1}{50} \sqrt{\frac{2}{\pi}} \theta \left\| \boldsymbol{q} - (\boldsymbol{R}\boldsymbol{Q}^{-1})^{-1} \frac{\boldsymbol{e}_n}{\widetilde{r}_n} \right\| \tag{31}
$$

*for any feasible $\boldsymbol{q}$ with $\langle \boldsymbol{r}, \boldsymbol{q} \rangle = 1$. Here, $\widetilde{\boldsymbol{r}} = (\boldsymbol{R}\boldsymbol{Q}^{-1})^{-\top} \boldsymbol{r}$.*

**Proof** Let us denote $\widetilde{q} = RQ^{-1}q$. Then we can rewrite our original problem as

$$\min_{\widetilde{q}} \; \widetilde{\zeta}(\widetilde{q}) \;=\; \frac{1}{np} \sum_{i=1}^{p} \|C_{x_i}\widetilde{q}\|_1 \quad \text{s.t.} \quad \langle \widetilde{r}, \widetilde{q}\rangle = 1,$$

which is reduced to the orthogonal problem in (32) of Lemma C.7. To utilize the result in Lemma C.7, we first prove that $\widetilde{r}$ satisfies (33) if $r$ satisfies (30). Towards that end, note that

$$\widetilde{r} \;=\; \left(RQ^{-1}\right)^{-\top} r \;=\; r + \left(\left(RQ^{-1}\right)^{-\top} - I\right) r.$$

By Lemma H.4, we know that, for any $\delta \in (0,1)$, whenever $p \geqslant C \frac{\kappa^8}{\theta \delta^2 \sigma_{\min}^2(C_a)} \log^3 n$,

$$\left\|\left(\left(RQ^{-1}\right)^{-\top} - I\right) r\right\| \;\leqslant\; \left\|\left(RQ^{-1}\right)^{-1} - I\right\| \|r\| \;\leqslant\; 2\delta \|r\|$$

holds with probability at least $1 - p^{-c_1 n\theta} - n^{-c_2}$. This further implies that

$$\widetilde{r}_n \;\geqslant\; r_n - 2\delta \|r\|, \quad \|\widetilde{r}_{-n}\| \;\leqslant\; \|r_{-n}\| + 2\delta \|r\|.$$

Therefore, by choose $\delta$ sufficiently small, we have

$$\frac{\|\widetilde{r}_{-n}\|}{\widetilde{r}_n} \;\leqslant\; \frac{\|r_{-n}\| + 2\delta \|r\|}{r_n - 2\delta \|r\|} \;=\; \frac{\|r_{-n}\|/r_n + 2\delta\sqrt{1 + (\|r_{-n}\|/r_n)^2}}{1 - 2\delta\sqrt{1 + (\|r_{-n}\|/r_n)^2}} \;\leqslant\; \frac{1}{10},$$

where the last inequality follows from (30). Therefore, by Lemma C.7, we obtain

$$\zeta(q) - \zeta\left(\left(RQ^{-1}\right)^{-1} \frac{e_n}{\widetilde{r}_n}\right) \;=\; \widetilde{\zeta}(q) - \widetilde{\zeta}\left(\frac{e_n}{\widetilde{r}_n}\right)$$

$$\geqslant\; \frac{1}{25}\sqrt{\frac{2}{\pi}}\theta\left\|\widetilde{q} - \frac{e_n}{\widetilde{r}_n}\right\|$$

$$=\; \frac{1}{25}\sqrt{\frac{2}{\pi}}\theta\left\|\left(RQ^{-1}\right)\cdot\left(q - \left(RQ^{-1}\right)^{-1}\frac{e_n}{\widetilde{r}_n}\right)\right\|$$

$$\geqslant\; \frac{1}{25}\sqrt{\frac{2}{\pi}}\theta \cdot \sigma_{\min}\left(RQ^{-1}\right)\cdot\left\|q - \left(RQ^{-1}\right)^{-1}\frac{e_n}{\widetilde{r}_n}\right\|$$

By Lemma H.4, we know that $\left\|\left(RQ^{-1}\right)^{-1}\right\| \leqslant 1 + 2\delta$, so that

$$\sigma_{\min}\left(RQ^{-1}\right) \;=\; \left\|\left(RQ^{-1}\right)^{-1}\right\|^{-1} \;\geqslant\; \frac{1}{1 + 2\delta}.$$

Thus, this further implies that

$$\zeta(q) - \zeta\left(\left(RQ^{-1}\right)^{-1}\frac{e_n}{\widetilde{r}_n}\right) \;\geqslant\; \frac{1}{25}\sqrt{\frac{2}{\pi}}\frac{\theta}{1 + 2\delta}\cdot\left\|q - \left(RQ^{-1}\right)^{-1}\frac{e_n}{\widetilde{r}_n}\right\|,$$

as desired. $\blacksquare$

**Lemma C.7 (Sharpness for the orthogonal case)** *Consider the following problem*

$$\min_{q} \widetilde{\zeta}(q) := \frac{1}{np} \sum_{i=1}^{p} \|C_{x_i}q\|_1 \quad s.t. \quad \langle r, q\rangle = 1, \tag{32}$$

*with $r \in \mathbb{S}^{n-1}$ satisfying*

$$\frac{\|r_{-n}\|}{r_n} \;\leqslant\; \frac{1}{10}, \quad r_n > 0. \tag{33}$$

*Whenever $p \geqslant \frac{C}{\theta^2} n \log\left(\frac{n}{\theta}\right)$, with probability at least $1 - c_1 np^{-6} - c_2 ne^{-c_3\theta^2 p}$, the function $\widetilde{\zeta}(q)$ is sharp in a sense that*

$$\widetilde{\zeta}(q) - \widetilde{\zeta}\left(\frac{e_n}{r_n}\right) \;\geqslant\; \frac{1}{25}\sqrt{\frac{2}{\pi}}\theta\left\|q - \frac{e_n}{r_n}\right\|$$

*for any feasible $q$ with $\langle r, q\rangle = 1$.*

**Proof** Observing that $\langle \boldsymbol{r}, \boldsymbol{q} \rangle = \boldsymbol{r}_{-n}^{\top} \boldsymbol{q}_{-n} + r_n q_n = 1$, we have

$$\|\boldsymbol{r}_{-n}\| \, \|\boldsymbol{q}_{-n}\| \; \geqslant \; \boldsymbol{r}_{-n}^{\top} \boldsymbol{q}_{-n} \; = \; r_n \left( \frac{1}{r_n} - q_n \right) \; \geqslant \; r_n \left( \frac{1}{r_n} - |q_n| \right).$$

This further implies that

$$\frac{1}{r_n} - |q_n| \; \leqslant \; \frac{\|\boldsymbol{r}_{-n}\|}{r_n} \, \|\boldsymbol{q}_{-n}\|. \tag{34}$$

Second, we have

$$\left\| \boldsymbol{q} - \frac{\boldsymbol{e}_n}{r_n} \right\| \; = \; \sqrt{ \left( \frac{1}{r_n} - q_n \right)^2 + \|\boldsymbol{q}_{-n}\|^2 } \; \leqslant \; \sqrt{ 1 + \left( \frac{\|\boldsymbol{r}_{-n}\|}{r_n} \right)^2 } \, \|\boldsymbol{q}_{-n}\|,$$

which implies that

$$\left( 1 + \left( \frac{\|\boldsymbol{r}_{-n}\|}{r_n} \right)^2 \right)^{-1/2} \left\| \boldsymbol{q} - \frac{\boldsymbol{e}_n}{r_n} \right\| \; \leqslant \; \|\boldsymbol{q}_{-n}\|. \tag{35}$$

We now proceed by considering the following two cases.

**Case i:** $|q_n| \geqslant \frac{1}{r_n}$. In this case, we have

$$\widetilde{\zeta}(\boldsymbol{q}) - \widetilde{\zeta}\left( \frac{\boldsymbol{e}_n}{r_n} \right) \; \geqslant \; \frac{1}{6} \sqrt{\frac{2}{\pi}} \theta \, \|\boldsymbol{q}_{-n}\| \; \geqslant \; \frac{1}{6} \sqrt{\frac{2}{\pi}} \theta \left( 1 + \left( \frac{\|\boldsymbol{r}_{-n}\|}{r_n} \right)^2 \right)^{-1/2} \left\| \boldsymbol{q} - \frac{\boldsymbol{e}_n}{r_n} \right\|$$

$$\geqslant \; \frac{5}{33} \sqrt{\frac{2}{\pi}} \theta \left\| \boldsymbol{q} - \frac{\boldsymbol{e}_n}{r_n} \right\|,$$

where the first inequality follows by (36), the second inequality follows by (35), and the last inequality follows because $\frac{\|\boldsymbol{r}_{-n}\|}{r_n} \leqslant \frac{1}{10}$.

**Case ii:** $|q_n| \leqslant \frac{1}{r_n}$. In this case, we have

$$\widetilde{\zeta}(\boldsymbol{q}) - \widetilde{\zeta}\left( \frac{\boldsymbol{e}_n}{r_n} \right) \; \geqslant \; \frac{1}{6} \sqrt{\frac{2}{\pi}} \theta \, \|\boldsymbol{q}_{-n}\| - \frac{5}{4} \sqrt{\frac{2}{\pi}} \theta \left( \frac{1}{r_n} - |q_n| \right)$$

$$\geqslant \; \theta \left( \frac{1}{6} \sqrt{\frac{2}{\pi}} - \frac{5}{4} \sqrt{\frac{2}{\pi}} \frac{\|\boldsymbol{r}_{-n}\|}{r_n} \right) \|\boldsymbol{q}_{-n}\|$$

$$\geqslant \; \theta \left( \frac{1}{6} \sqrt{\frac{2}{\pi}} - \frac{5}{4} \sqrt{\frac{2}{\pi}} \frac{\|\boldsymbol{r}_{-n}\|}{r_n} \right) \left( 1 + \left( \frac{\|\boldsymbol{r}_{-n}\|}{r_n} \right)^2 \right)^{-1/2} \left\| \boldsymbol{q} - \frac{\boldsymbol{e}_n}{r_n} \right\|$$

$$\geqslant \; \frac{\theta}{25} \sqrt{\frac{2}{\pi}} \left\| \boldsymbol{q} - \frac{\boldsymbol{e}_n}{r_n} \right\|,$$

where the first inequality follows by (36), the second inequality follows from (34), the third inequality follows from (35), and the last one follows because $\frac{\|\boldsymbol{r}_{-n}\|}{r_n} \leqslant \frac{1}{10}$.

Combining the results in both cases, we obtain the desired result. ∎

**Lemma C.8** *Suppose $\theta \in \left( \frac{1}{n}, \frac{1}{3} \right)$. Whenever $p \geqslant \frac{C}{\theta^2} n \log \left( \frac{n}{\theta} \right)$, we have*

$$\widetilde{\zeta}(\boldsymbol{q}) - \widetilde{\zeta}\left( \frac{\boldsymbol{e}_n}{r_n} \right) \; \geqslant \; \begin{cases} \frac{1}{6} \sqrt{\frac{2}{\pi}} \theta \, \|\boldsymbol{q}_{-n}\|, & \text{if } |q_n| - \frac{1}{r_n} \geqslant 0, \\ \frac{1}{6} \sqrt{\frac{2}{\pi}} \theta \, \|\overline{\boldsymbol{q}}\| - \frac{5}{4} \sqrt{\frac{2}{\pi}} \theta \left( \frac{1}{r_n} - |q_n| \right), & \text{if } |q_n| - \frac{1}{r_n} < 0, \end{cases} \tag{36}$$

*holds with probability at least $1 - c_1 n p^{-6} - c_2 n e^{-c_3 \theta^2 p}$.*

**Proof** For each $j \in [n]$, let us define an index set $\mathcal{I}_j := \{i \in [p] : (s_j [\breve{\boldsymbol{x}}_i])_n \neq 0\}$, and let us define events

$$\mathcal{E} := \bigcap_{j=0}^{n-1} \mathcal{E}_j, \quad \mathcal{E}_j := \left\{ |\mathcal{I}_i| \leqslant \frac{9}{8} \theta p \right\}, \quad (0 \leqslant j \leqslant n-1).$$

By Hoeffding's inequality and a union bound, we know that

$$\mathbb{P}\left(\mathcal{E}^c\right) \leqslant \sum_{j=0}^{n-1} \mathbb{P}\left(\mathcal{E}_j^c\right) \leqslant n \exp\left(-p\theta^2/2\right).$$

Based on this, we have

$$\tilde{\zeta}(\boldsymbol{q}) - \tilde{\zeta}\left(\frac{\boldsymbol{e}_n}{r_n}\right)$$

$$= \frac{1}{np} \sum_{i=1}^{p} \|\boldsymbol{C}_{\boldsymbol{x}_i} \boldsymbol{q}\|_1 - \frac{1}{np} \frac{1}{r_n} \sum_{i=1}^{p} \|\boldsymbol{x}_i\|_1$$

$$= \frac{1}{np} \sum_{i=1}^{p} \sum_{j=0}^{n-1} |\langle s_j [\breve{\boldsymbol{x}}_i], \boldsymbol{q} \rangle| - \frac{1}{np} \frac{1}{r_n} \sum_{i=1}^{p} \|\boldsymbol{x}_i\|_1$$

$$\geqslant \frac{1}{np} \left( |q_n| - \frac{1}{r_n} \right) \sum_{i=1}^{p} \|\boldsymbol{x}_i\|_1 + \frac{1}{np} \sum_{j=0}^{n-1} \left( \sum_{i \in \mathcal{I}_j^c} |\langle (s_j [\breve{\boldsymbol{x}}_i])_{-n}, \boldsymbol{q}_{-n} \rangle| - \sum_{i \in \mathcal{I}_j} |\langle (s_j [\breve{\boldsymbol{x}}_i])_{-n}, \boldsymbol{q}_{-n} \rangle| \right)$$

$$= \frac{1}{np} \left( |q_n| - \frac{1}{r_n} \right) \sum_{i=1}^{p} \|\boldsymbol{x}_i\|_1 + \frac{1}{np} \sum_{j=0}^{n-1} \left( \left\| \boldsymbol{q}_{-n}^\top \boldsymbol{M}_{\mathcal{I}_j^c}^j \right\|_1 - \left\| \boldsymbol{q}_{-n}^\top \boldsymbol{M}_{\mathcal{I}_j}^j \right\|_1 \right),$$

where we denote $\boldsymbol{M}^j = [(s_j [\breve{\boldsymbol{x}}_1])_{-n} \quad (s_j [\breve{\boldsymbol{x}}_2])_{-n} \quad \cdots \quad (s_j [\breve{\boldsymbol{x}}_p])_{-n}]$, and $\boldsymbol{M}_{\mathcal{I}}^j$ denote a submatrix of $\boldsymbol{M}^j$ with columns indexed by $\mathcal{I}$. Conditioned on the event $\mathcal{E}$, by Lemma D.5 and a union bound, whenever $p \geqslant \frac{C}{\theta^2} n \log\left(\frac{n}{\theta}\right)$, we have

$$\left\| \boldsymbol{q}_{-n}^\top \boldsymbol{M}_{\mathcal{I}_j^c}^j \right\|_1 - \left\| \boldsymbol{q}_{-n}^\top \boldsymbol{M}_{\mathcal{I}_j}^j \right\|_1 \geqslant \frac{p}{6} \sqrt{\frac{2}{\pi}} \theta \|\boldsymbol{q}_{-n}\|, \ \forall \ \boldsymbol{q}_{-n} \in \mathbb{R}^{n-1}, \ (0 \leqslant j \leqslant n-1)$$

with probability at least $1 - cnp^{-6}$. On the other hand, by Gaussian concentration inequality, we have

$$\mathbb{P}\left( \frac{1}{np} \sum_{i=1}^{p} \|\boldsymbol{x}_i\|_1 \geqslant \frac{5}{4} \sqrt{\frac{2}{\pi}} \theta \right) \leqslant \exp\left( -\frac{\theta^2 p}{64\pi} \right).$$

Therefore, combining all the results above, we have

$$\tilde{\zeta}(\boldsymbol{q}) - \tilde{\zeta}\left(\frac{\boldsymbol{e}_n}{r_n}\right) \geqslant \begin{cases} \frac{1}{6} \sqrt{\frac{2}{\pi}} \theta \|\boldsymbol{q}_{-n}\|, & \text{if } |q_n| - \frac{1}{r_n} \geqslant 0, \\ \frac{1}{6} \sqrt{\frac{2}{\pi}} \theta \|\overline{\boldsymbol{q}}\| - \frac{5}{4} \sqrt{\frac{2}{\pi}} \theta \left( \frac{1}{r_n} - |q_n| \right), & \text{if } |q_n| - \frac{1}{r_n} < 0, \end{cases}$$

as desired. ∎

## C.3 Linear convergence for projection subgradient descent in Algorithm 3

Now based on the sharpness condition, we are ready to show that the projected subgradient descent method

$$\boldsymbol{q}^{(k+1)} = \boldsymbol{q}^{(k)} - \tau^{(k)} \mathcal{P}_{\boldsymbol{r}^\perp} \boldsymbol{g}^{(k)}, \quad \boldsymbol{g}^{(k)} = \sum_{i=1}^{p} \left( \boldsymbol{R}\boldsymbol{Q}^{-1} \right)^\top \boldsymbol{C}_{\boldsymbol{x}_i}^\top \operatorname{sign}\left( \boldsymbol{C}_{\boldsymbol{x}_i} \boldsymbol{R}\boldsymbol{Q}^{-1} \boldsymbol{q}^{(k)} \right).$$

on $\zeta(\boldsymbol{q})$ converges linearly to the target solution up to a scaling factor. For convenience, let us first define the distance between the iterate and the target solution

$$d^{(k)} := \left\| \boldsymbol{s}^{(k)} \right\|, \quad \boldsymbol{s}^{(k)} := \boldsymbol{q}^{(k)} - \left( \boldsymbol{R}\boldsymbol{Q}^{-1} \right)^{-1} \frac{\boldsymbol{e}_n}{\tilde{r}_n},$$

and several parameters

$$\alpha := \frac{1}{50}\sqrt{\frac{2}{\pi}}\theta, \quad \beta := 36\log(np).$$

We show the following result.

**Proposition C.9** *Suppose $\theta \in \left(\frac{1}{n}, \frac{1}{3}\right)$ and $\boldsymbol{r}$ satisfies*

$$\frac{\|\boldsymbol{r}_{-n}\|}{r_n} \leqslant \frac{1}{20}, \quad r_n > 0, \quad \|\boldsymbol{r}\| = 1. \tag{37}$$

*Let $\boldsymbol{q}^{(k)}$ be the sequence generated by the projected subgradient method (cf. Algorithm 3) with initialization $\boldsymbol{q}^{(0)} = \boldsymbol{r}$ and geometrically decreasing step size*

$$\tau^{(k)} = \eta^k \tau^{(0)}, \quad \tau^{(0)} = \frac{16}{25}\frac{\alpha}{\beta^2}, \quad \sqrt{1 - \frac{\alpha^2}{2\beta^2}} \leqslant \eta < 1 \tag{38}$$

*Whenever $p \geqslant C\frac{\kappa^8}{\theta\sigma_{\min}^2(\boldsymbol{C_a})}\log^3 n$, with probability at least $1 - p^{-c_1 n\theta} - n^{-c_2}$, the sequence $\left\{\boldsymbol{q}^{(k)}\right\}_{k\geqslant 0}$ satisfies*

$$\left\|\boldsymbol{q}^{(k)} - \left(\boldsymbol{RQ}^{-1}\right)^{-1}\frac{\boldsymbol{e}_n}{\widetilde{r}_n}\right\| \leqslant \frac{2}{5}\eta^k, \tag{39}$$

*for all iteration $k = 0, 1, 2, \cdots$.*

**Proof** Given the initialization $\boldsymbol{q}^{(0)} = \boldsymbol{r}$, we have

$$d^{(0)} = \left\|\boldsymbol{r} - \left(\boldsymbol{RQ}^{-1}\right)^{-1}\frac{\boldsymbol{e}_n}{\widetilde{r}_n}\right\| \leqslant \left\|\left(\boldsymbol{RQ}^{-1}\right)^{-1}\right\|\left\|\widetilde{\boldsymbol{r}} - \frac{\boldsymbol{e}_n}{\widetilde{r}_n}\right\|$$

$$\leqslant \frac{10}{9}\cdot\left(\|\widetilde{\boldsymbol{r}}_{-n}\|^2 + \left(\widetilde{r}_n - \frac{1}{\widetilde{r}_n}\right)^2\right)^{1/2},$$

where the last inequality we used Lemma H.4. From the argument in Proposition C.6, we know that (37) implies $\|\widetilde{\boldsymbol{r}}_{-n}\|/\widetilde{r}_n \leqslant 1/10$. By the fact that $\|\widetilde{\boldsymbol{r}}\| \leqslant 10/9$, we have

$$\|\widetilde{\boldsymbol{r}}_{-n}\| \leqslant \frac{1}{9}, \quad \left|\widetilde{r}_n - \frac{1}{\widetilde{r}_n}\right| \leqslant \left|\frac{8}{9} - \frac{9}{8}\right|^2 \leqslant \frac{1}{4} \implies d^{(0)} \leqslant \frac{2}{5}. \tag{40}$$

On the other hand, notice that

$$\left(d^{(k+1)}\right)^2 = \left\|\boldsymbol{q}^{(k)} - \tau^{(k)}\mathcal{P}_{\boldsymbol{r}^\perp}\boldsymbol{g}^{(k)} - \left(\boldsymbol{RQ}^{-1}\right)^{-1}\frac{\boldsymbol{e}_n}{\widetilde{r}_n}\right\|^2$$

$$= \left(d^{(k)}\right)^2 - 2\tau^{(k)}\left\langle\boldsymbol{s}^{(k)}, \mathcal{P}_{\boldsymbol{r}^\perp}\boldsymbol{g}^{(k)}\right\rangle + \left(\tau^{(k)}\right)^2\left\|\mathcal{P}_{\boldsymbol{r}^\perp}\boldsymbol{g}^{(k)}\right\|^2$$

By Lemma C.10, we know that when $p \geqslant C\frac{\kappa^8}{\theta\sigma_{\min}^2(\boldsymbol{C_a})}\log^3 n$, for any $k = 1, 2, \cdots$,

$$\left\|\mathcal{P}_{\boldsymbol{r}^\perp}\boldsymbol{g}^{(k)}\right\|^2 \leqslant 36\log(np) = \beta$$

holds with probability at least $1 - p^{-c_1 n\theta} - n^{-c_2}$. On the other hand, by the sharpness property of the function in Proposition C.6, for any $k = 1, 2, \cdots$,

$$\left\langle\boldsymbol{s}^{(k)}, \mathcal{P}_{\boldsymbol{r}^\perp}\boldsymbol{g}^{(k)}\right\rangle = \left\langle\boldsymbol{s}^{(k)}, \boldsymbol{g}^{(k)}\right\rangle \geqslant \zeta\left(\boldsymbol{q}^{(k)}\right) - \zeta\left(\left(\boldsymbol{RQ}^{-1}\right)^{-1}\frac{\boldsymbol{e}_n}{\widetilde{r}_n}\right)$$

$$\geqslant \frac{1}{50}\sqrt{\frac{2}{\pi}}\theta\left\|\boldsymbol{q}^{(k)} - \left(\boldsymbol{RQ}^{-1}\right)^{-1}\frac{\boldsymbol{e}_n}{\widetilde{r}_n}\right\| = \alpha \cdot d^{(k)},$$

where the first equality follows from the fact that $\langle \boldsymbol{r}, \boldsymbol{s}^{(k)}\rangle = 0$ so that $\mathcal{P}_{\boldsymbol{r}^\perp} \boldsymbol{s}^{(k)} = \boldsymbol{s}^{(k)}$, the first inequality follows from the fact that $\zeta(\boldsymbol{q})$ is convex, and the second inequality utilizes the sharpness of the function in Proposition C.6 given the condition (37). Thus, we have

$$\left(d^{(k+1)}\right)^2 \;\leqslant\; \left(d^{(k)}\right)^2 - 2\alpha \cdot \tau^{(k)} \cdot d^{(k)} + \beta^2 \cdot \left(\tau^{(k)}\right)^2.$$

Now we proceed to prove (39) by induction. It is clear that (39) holds for $\boldsymbol{q}^{(0)}$. Suppose $\boldsymbol{q}^{(k)}$ satisfies (39), i.e., $d^{(k)} \leqslant \eta^k d^{(0)}$ for some $k \geqslant 1$. The quadratic term of $d^{(k)}$ on the right hand side of the inequality above will obtain its maximum at $\frac{2}{5}\eta^k$ due to the definition of $\tau^{(0)}$ and $d^{(0)} \leqslant \frac{2}{5}$ as shown in (40). This, together with $\tau^{(k)} = \eta\tau^{(k-1)}$, it gives

$$
\begin{aligned}
\left(d^{(k+1)}\right)^2 &\;\leqslant\; \frac{4}{25}\eta^{2k} - \frac{4}{5}\alpha \cdot \eta^{2k}\tau^{(0)} + \beta^2 \cdot \eta^{2k}\left(\tau^{(0)}\right)^2 \\
&\;=\; \frac{4}{25}\eta^{2k} \cdot \left[1 - 5\alpha\tau^{(0)} + \frac{25}{4}\beta^2 \left(\tau^{(0)}\right)^2\right] \;\leqslant\; \eta^{2k+2} \cdot \left(d^{(0)}\right)^2
\end{aligned}
$$

where the last inequality follows from (38), where

$$1 - 5\alpha\tau^{(0)} + \frac{25}{4}\beta^2\left(\tau^{(0)}\right)^2 \;\leqslant\; 1 - \alpha\tau^{(0)} \;\leqslant\; 1 - \frac{\alpha^2}{2\beta^2} \;\leqslant\; \eta^2 < 1.$$

This completes the proof. ∎

**Lemma C.10** *Suppose $\theta \in \left(\frac{1}{n}, \frac{1}{3}\right)$. Whenever $p \geqslant C\frac{\kappa^8}{\theta\sigma_{\min}^2(\boldsymbol{C_a})}\log^3 n$, we have*

$$\rho \;:=\; \sup_{\boldsymbol{q}:\boldsymbol{q}^\top\boldsymbol{r}=1} \frac{1}{np}\left\|\mathcal{P}_{\boldsymbol{r}^\perp}\sum_{i=1}^p \left(\boldsymbol{R}\boldsymbol{Q}^{-1}\right)^\top \boldsymbol{C}_{\boldsymbol{x}_i}^\top \operatorname{sign}\left(\boldsymbol{C}_{\boldsymbol{x}_i}\boldsymbol{R}\boldsymbol{Q}^{-1}\boldsymbol{q}\right)\right\| \;\leqslant\; 6\sqrt{\log(np)} \qquad (41)$$

*holds with probability at least $1 - p^{-c_1 n\theta} - n^{-c_2}$.*

**Proof** We have

$$\rho \;\leqslant\; \frac{1}{np}\left\|\boldsymbol{R}\boldsymbol{Q}^{-1}\right\|\sum_{i=1}^p \left(\left\|\boldsymbol{C}_{\boldsymbol{x}_i}\right\| \sup_{\boldsymbol{q}:\boldsymbol{q}^\top\boldsymbol{r}=1}\left\|\operatorname{sign}\left(\boldsymbol{C}_{\boldsymbol{x}_i}\boldsymbol{R}\boldsymbol{Q}^{-1}\boldsymbol{q}\right)\right\|\right).$$

Since the $\operatorname{sign}(\cdot)$ function is bounded by 1, we have

$$\rho \;\leqslant\; \frac{1}{np}\left\|\boldsymbol{R}\boldsymbol{Q}^{-1}\right\| \cdot \left(\sum_{i=1}^p \left\|\boldsymbol{F}\boldsymbol{x}_i\right\|_\infty\right) \cdot \sqrt{n},$$

where we used the fact that $\left\|\boldsymbol{C}_{\boldsymbol{x}_i}\right\| = \left\|\boldsymbol{F}\boldsymbol{x}_i\right\|_\infty$. As $\boldsymbol{x}_i \sim_{i.i.d.} \mathcal{BG}(\theta)$, let $\boldsymbol{x}_i = \boldsymbol{b}_i \odot \boldsymbol{g}_i$ with $\boldsymbol{b}_i \sim \mathcal{B}(\theta)$ and $\boldsymbol{g}_i \sim \mathcal{N}(\boldsymbol{0}, \boldsymbol{I})$. Then we have

$$\left\|\boldsymbol{C}_{\boldsymbol{x}_i}\right\| = \left\|\boldsymbol{F}\boldsymbol{x}_i\right\|_\infty = \max_{1\leqslant j\leqslant n}\left|(\boldsymbol{f}_j \odot \boldsymbol{b}_i)^* \boldsymbol{g}_i\right|.$$

By Gaussian concentration inequality in Lemma D.4 and a union bound, we have

$$\mathbb{P}\left(\max_{1\leqslant i\leqslant p}\left\|\boldsymbol{F}\boldsymbol{x}_i\right\| \geqslant t\right) \;\leqslant\; (np) \cdot \exp\left(-\frac{t^2}{2n}\right).$$

Choose $t = 4\sqrt{n\log(np)}$, then we have

$$\max_{1\leqslant i\leqslant p}\left\|\boldsymbol{F}\boldsymbol{x}_i\right\| \;\leqslant\; 4\sqrt{n\log(np)},$$

with probability at least $1 - (np)^{-7}$. On the other hand, by Lemma H.4, we know that whenever $p \geqslant C\frac{\kappa^8}{\theta\sigma_{\min}^2(\boldsymbol{C_a})}\log^3 n$, we have

$$\left\|\boldsymbol{R}\boldsymbol{Q}^{-1}\right\| \;\leqslant\; \frac{3}{2},$$

holds with probability at least $1 - p^{-c_1 n\theta} - n^{-c_2}$. Combining all the results above, we obtain

$$\rho \;\leqslant\; \frac{1}{np} \cdot \frac{3}{2} \cdot \left(4p\sqrt{n\log(np)}\right) \cdot \sqrt{n} \;=\; 6\sqrt{\log(np)},$$

as desired. ∎

 # D  Basics

651 **Lemma D.1 (Moments of the Gaussian Random Variable)** *If $X \sim \mathcal{N}\left(0, \sigma_X^2\right)$, then it holds for*
650 *all integer $m \geqslant 1$ that*

$$\mathbb{E}\left[|X|^m\right] = \sigma_X^m \left(m-1\right)!! \left[\sqrt{\frac{2}{\pi}} \mathbb{1}_{m=2k+1} + \mathbb{1}_{m=2k}\right] \leqslant \sigma_X^m \left(m-1\right)!!, \ k = \lfloor m/2 \rfloor.$$

651 **Lemma D.2 (sub-Gaussian Random Variables)** *Let $X$ be a centered $\sigma^2$ sub-Gaussian random*
652 *variable, such that*

$$\mathbb{P}\left(|X| \geqslant t\right) \leqslant 2 \exp\left(-\frac{t^2}{2\sigma^2}\right),$$

653 *then for any integer $p \geqslant 1$, we have*

$$\mathbb{E}\left[|X|^p\right] \leqslant \left(2\sigma^2\right)^{p/2} p\Gamma(p/2).$$

654 *In particular, we have*

$$\|X\|_{L^p} = \left(\mathbb{E}\left[|X|^p\right]\right)^{1/p} \leqslant \sigma e^{1/e} \sqrt{p}, \quad p \geqslant 2,$$

655 *and $\mathbb{E}\left[|X|\right] \leqslant \sigma\sqrt{2\pi}$.*

656 **Lemma D.3 (Moment-Control Bernstein's Inequality for Random Variables [42])** *Let*
657 *$X_1, \cdots, X_N$ be i.i.d. real-valued random variables. Suppose that there exist some positive*
658 *numbers $R$ and $\sigma_X^2$ such that*

$$\mathbb{E}\left[|X_k|^m\right] \leqslant \frac{m!}{2} \sigma_X^2 R^{m-2}, \ \text{for all integers } m \geqslant 2.$$

659 *Let $S \doteq \frac{1}{N} \sum_{k=1}^N X_k$, then for all $t > 0$, it holds that*

$$\mathbb{P}\left[|S - \mathbb{E}\left[S\right]| \geqslant t\right] \leqslant 2 \exp\left(-\frac{Nt^2}{2\sigma_X^2 + 2Rt}\right).$$

660 **Lemma D.4 (Gaussian Concentration Inequality)** *Let $\boldsymbol{g} \in \mathbb{R}^n$ be a standard Gaussian random*
661 *variable $\boldsymbol{g} \sim \mathcal{N}(\boldsymbol{0}, \boldsymbol{I})$, and let $f : \mathbb{R}^n \mapsto \mathbb{R}$ denote an L-Lipschitz function. Then for all $t > 0$,*

$$\mathbb{P}\left(|f(\boldsymbol{g}) - \mathbb{E}\left[f(\boldsymbol{g})\right]| \geqslant t\right) \leqslant 2 \exp\left(-\frac{t^2}{2L^2}\right).$$

662 **Lemma D.5 (Lemma VII.1, [34])** *Let $\boldsymbol{M} \in \mathbb{R}^{n_1 \times n_2}$ with $\boldsymbol{M} \sim \mathcal{BG}(\theta)$ and $\theta \in (0, 1/3)$. For a*
663 *given set $\mathcal{I} \subseteq [n_2]$ with $|\mathcal{I}| \leqslant \frac{9}{8}\theta n_2$, whenever $n_2 \geqslant \frac{C}{\theta^2} n_1 \log\left(\frac{n_1}{\theta}\right)$, it holds*

$$\left\|\boldsymbol{v}^\top \boldsymbol{M}_{\mathcal{I}^c}\right\|_1 - \left\|\boldsymbol{v}^\top \boldsymbol{M}_{\mathcal{I}}\right\|_1 \geqslant \frac{n_2}{6}\sqrt{\frac{2}{\pi}}\theta \|\boldsymbol{v}\|$$

664 *for all $\boldsymbol{v} \in \mathbb{R}^{n_1}$, with probability at least $1 - cn_2^{-6}$.*

665 **Lemma D.6 (Derivates of $h_\mu(z)$)** *The first two derivatives of $h_\mu(z)$ are*

$$\nabla h_\mu(z) = \begin{cases} \text{sign}(z) & |z| \geqslant \mu \\ z/\mu & |z| < \mu \end{cases}, \quad \nabla^2 h_\mu(z) = \begin{cases} 0 & |z| > \mu \\ 1/\mu & |z| < \mu \end{cases}. \tag{42}$$

666 *Whenever necessary, we define $\nabla^2 h_\mu(\mu) = 0$, and write the "second derivative" as $\nabla^2 \overline{h}_\mu(\mu)$*
667 *instead. Moreover for all $z, z'$,*

$$\left|\nabla h_\mu(z) - \nabla h_\mu(z')\right| \leqslant \frac{1}{\mu}|z - z'|. \tag{43}$$

**Lemma D.7** *Let $X \sim \mathcal{N}(0, \sigma_x^2)$ and $Y \sim \mathcal{N}(0, \sigma_y^2)$ and $Z \sim \mathcal{N}\left(0, \sigma_z^2\right)$ be independent random variables. Then we have*

$$\mathbb{E}\left[X \mathbb{1}_{X+Y \geqslant \mu}\right] = \frac{\sigma_x^2}{\sqrt{2\pi}\sqrt{\sigma_x^2 + \sigma_y^2}} \exp\left(-\frac{\mu^2}{2(\sigma_x^2 + \sigma_y^2)}\right), \tag{44}$$

$$\mathbb{E}\left[XY \mathbb{1}_{|X+Y| \leqslant \mu}\right] = -\sqrt{\frac{2}{\pi}} \frac{\mu \sigma_x^2 \sigma_y^2}{\left(\sigma_x^2 + \sigma_y^2\right)^{3/2}} \exp\left(-\frac{\mu^2}{2\left(\sigma_x^2 + \sigma_y^2\right)}\right), \tag{45}$$

$$\mathbb{E}\left[|X| \mathbb{1}_{|X| > \mu}\right] = \sqrt{\frac{2}{\pi}} \sigma_x \exp\left(-\frac{\mu^2}{2\sigma_x^2}\right), \tag{46}$$

$$\mathbb{E}\left[XY \mathbb{1}_{|X+Y+Z| < \mu}\right] = -\sqrt{\frac{2}{\pi}} \mu \exp\left(-\frac{\mu^2}{2\left(\sigma_x^2 + \sigma_y^2 + \sigma_z^2\right)}\right) \frac{\sigma_x^2 \sigma_y^2}{\left(\sigma_x^2 + \sigma_y^2 + \sigma_z^2\right)^{3/2}}, \tag{47}$$

$$\mathbb{E}\left[X^2 \mathbb{1}_{|X| < \mu}\right] = -\sqrt{\frac{2}{\pi}} \sigma_x \mu \exp\left(-\frac{\mu^2}{2\sigma_x^2}\right) + \sigma_x^2 \mathbb{P}\left[|X| < \mu\right], \tag{48}$$

$$\mathbb{E}\left[X^2 \mathbb{1}_{|X+Y| < \mu}\right] = -\sqrt{\frac{2}{\pi}} \mu \frac{\sigma_x^4}{\left(\sigma_x^2 + \sigma_y^2\right)^{3/2}} \exp\left(-\frac{\mu^2}{2\left(\sigma_x^2 + \sigma_y^2\right)}\right) + \sigma_x^2 \mathbb{P}\left[|X+Y| < \mu\right]. \tag{49}$$

**Proof** *Direct calculations.* ∎

**Lemma D.8 (Calculus for Function of Matrices, Chapter X of [43])** *Let $\mathcal{S}^{n \times n}$ be the set of symmetric matrices of size $n \times n$. We define a map $f : \mathcal{S}^{n \times n} \mapsto \mathcal{S}^{n \times n}$ as*

$$f(\boldsymbol{A}) = \boldsymbol{U} f(\boldsymbol{\Lambda}) \boldsymbol{U}^*,$$

*where $\boldsymbol{A} \in \mathcal{S}^{n \times n}$ has the eigen-decomposition $\boldsymbol{A} = \boldsymbol{U} \boldsymbol{\Lambda} \boldsymbol{U}^*$. The map $f$ is called (Fréchet) differentiable at $\boldsymbol{A}$ if there exists a linear transformation on $\mathcal{S}^{n \times n}$ such that for all $\boldsymbol{\Delta}$*

$$\|f(\boldsymbol{A} + \boldsymbol{\Delta}) - f(\boldsymbol{A}) - \mathrm{D}f(\boldsymbol{A})[\boldsymbol{\Delta}]\| = o\left(\|\boldsymbol{\Delta}\|\right).$$

*The linear operator $\mathrm{D}f(\boldsymbol{A})$ is called the derivative of $f$ at $\boldsymbol{A}$, and $\mathrm{D}f(\boldsymbol{A})[\boldsymbol{\Delta}]$ is the directional derivative of $f$ along $\boldsymbol{\Delta}$. If $f$ is differentiable at $\boldsymbol{A}$, then*

$$\mathrm{D}f(\boldsymbol{A})[\boldsymbol{\Delta}] = \frac{d}{dt} f(\boldsymbol{A} + t\boldsymbol{\Delta}) \bigg|_{t=0}.$$

*We denote the operator norm of the derivative $\mathrm{D}f(\boldsymbol{A})$ as*

$$\|\mathrm{D}f(\boldsymbol{A})\| \doteq \sup_{\|\boldsymbol{\Delta}\|=1} \|\mathrm{D}f(\boldsymbol{A})[\boldsymbol{\Delta}]\|.$$

**Lemma D.9 (Mean Value Theorem for Function of Matrices)** *Let $f$ be a differentiable map from a convex subset $\mathcal{U}$ of a Banach space $\mathcal{X}$ into the Banach space $\mathcal{Y}$. Let $\boldsymbol{A}, \boldsymbol{B} \in \mathcal{U}$, and let $\mathcal{L}$ be the line segment joining them. Then*

$$\|f(\boldsymbol{B}) - f(\boldsymbol{A})\| \leqslant \|\boldsymbol{B} - \boldsymbol{A}\| \sup_{\boldsymbol{U} \in \mathcal{L}} \|\mathrm{D}f(\boldsymbol{U})\|.$$

**Lemma D.10 (Theorem VII.2.3 of [43])** *Let $\boldsymbol{A}$ and $\boldsymbol{B}$ be operators whose spectra are contained in the open right half-plane and open left half-plane, respectively. Then the solution of the equation $\boldsymbol{A}\boldsymbol{X} - \boldsymbol{X}\boldsymbol{B} = \boldsymbol{Y}$ can be expressed as*

$$\boldsymbol{X} = \int_0^\infty e^{-t\boldsymbol{A}} \boldsymbol{Y} e^{t\boldsymbol{B}} dt$$

**Lemma D.11** *Let $f(\boldsymbol{A}) = \boldsymbol{A}^{-1/2}$, defined the set of all $n \times n$ positive definite matrices $\mathcal{S}_+^{n \times n}$, then we have*

$$\|\mathrm{D}f(\boldsymbol{A})\| \leqslant \frac{1}{\sigma_{\min}^2(\boldsymbol{A})},$$

*where $\sigma_{\min}(\boldsymbol{A})$ is the smallest singular value of $\boldsymbol{A}$.*

687 **Proof** To bound the operator norm $\|\mathrm{D}f(\boldsymbol{A})\|$, we introduce an auxiliary function

$$g(\boldsymbol{A}) \;=\; \boldsymbol{A}^{-2}, \qquad f(\boldsymbol{A}) = g^{-1}(\boldsymbol{A}),$$

688 such that $f$ and $g$ are the inverse function to each other. Whenever $g \circ f(\boldsymbol{A}) \neq 0$ (which is true for
689 our case $\boldsymbol{A} > \boldsymbol{0}$), this gives

$$\mathrm{D}f(\boldsymbol{A}) \;=\; [\mathrm{D}(g \circ f)(\boldsymbol{A})]^{-1} \;=\; \left[\mathrm{D}g(\boldsymbol{A}^{-1/2})\right]^{-1}. \tag{50}$$

690 This suggests that we can estimate $\mathrm{D}f(\boldsymbol{A})$ via estimating $\mathrm{D}g(\boldsymbol{A})$ of its inverse function $g$. Let

$$g \;=\; h \circ w(\boldsymbol{A}), \quad h(\boldsymbol{A}) \;=\; \boldsymbol{A}^{-1}, \quad w(\boldsymbol{A}) = \boldsymbol{A}^2,$$

691 such that their directional derivatives have simple form

$$\mathrm{D}h(\boldsymbol{A})[\boldsymbol{\Delta}] \;=\; -\boldsymbol{A}^{-1}\boldsymbol{\Delta}\boldsymbol{A}^{-1}, \quad \mathrm{D}w(\boldsymbol{A})[\boldsymbol{\Delta}] \;=\; \boldsymbol{\Delta}\boldsymbol{A} + \boldsymbol{A}\boldsymbol{\Delta}.$$

692 By using chain rule, simple calculation gives

$$\begin{aligned}
\mathrm{D}g(\boldsymbol{A})[\boldsymbol{\Delta}] &\;=\; \mathrm{D}h(w(\boldsymbol{A}))\left[\mathrm{D}w(\boldsymbol{A})[\boldsymbol{\Delta}]\right], \\
&\;=\; -\left(\boldsymbol{A}^{-2}\boldsymbol{\Delta}\boldsymbol{A}^{-1} + \boldsymbol{A}^{-1}\boldsymbol{\Delta}\boldsymbol{A}^{-2}\right).
\end{aligned}$$

693 Now by (50), the directional derivative

$$\boldsymbol{Z} \;\dot{=}\; \mathrm{D}f(\boldsymbol{A})[\boldsymbol{\Delta}]$$

694 satisfies

$$\boldsymbol{A}\boldsymbol{Z}\boldsymbol{A}^{1/2} + \boldsymbol{A}^{1/2}\boldsymbol{Z}\boldsymbol{A} \;=\; -\boldsymbol{\Delta}.$$

695 Since $\boldsymbol{A} > \boldsymbol{0}$, we write the eigen decomposition as $\boldsymbol{A} = \boldsymbol{U}\boldsymbol{\Lambda}\boldsymbol{U}^*$, with $\boldsymbol{U}$ orthogonal and $\boldsymbol{\Lambda} > 0$
696 diagonal. Let $\widetilde{\boldsymbol{Z}} = \boldsymbol{U}^*\boldsymbol{Z}\boldsymbol{U}$ and $\widetilde{\boldsymbol{\Delta}} = \boldsymbol{U}^*\boldsymbol{\Delta}\boldsymbol{U}$, then the equation above gives

$$\boldsymbol{\Lambda}^{1/2}\widetilde{\boldsymbol{Z}} - \widetilde{\boldsymbol{Z}}\left(-\boldsymbol{\Lambda}^{1/2}\right) = -\boldsymbol{\Lambda}^{-1/2}\widetilde{\boldsymbol{\Delta}}\boldsymbol{\Lambda}^{-1/2},$$

697 which is the Sylvester equation []. Since $\boldsymbol{\Lambda}^{1/2}$ and $-\boldsymbol{\Lambda}^{1/2}$ do not have common eigenvalues, Lemma
698 D.10 gives

$$\mathrm{D}f(\boldsymbol{A})[\boldsymbol{\Delta}] \;=\; \boldsymbol{U}\left[\int_0^\infty e^{-\boldsymbol{\Lambda}^{1/2}\tau}\left(-\boldsymbol{\Lambda}^{-1/2}\widetilde{\boldsymbol{\Delta}}\boldsymbol{\Lambda}^{-1/2}\right)e^{-\boldsymbol{\Lambda}^{1/2}\tau}d\tau\right]\boldsymbol{U}^*.$$

699 Thus, by Lemma D.8 we know that

$$\begin{aligned}
\|\mathrm{D}f(\boldsymbol{A})\| &\;=\; \sup_{\|\boldsymbol{\Delta}\|=1} \|\mathrm{D}f(\boldsymbol{A})[\boldsymbol{\Delta}]\| \\
&\;\leqslant\; \int_0^\infty \left\|e^{-\boldsymbol{\Lambda}^{1/2}\tau}\left(-\boldsymbol{\Lambda}^{-1/2}\widetilde{\boldsymbol{\Delta}}\boldsymbol{\Lambda}^{-1/2}\right)e^{-\boldsymbol{\Lambda}^{1/2}\tau}\right\| d\tau \\
&\;\leqslant\; \left\|\boldsymbol{\Lambda}^{-1/2}\widetilde{\boldsymbol{\Delta}}\boldsymbol{\Lambda}^{-1/2}\right\| \int_0^\infty e^{-\sigma_{\min}\tau}d\tau \;\leqslant\; \frac{1}{\sigma_{\min}^2(\boldsymbol{A})}.
\end{aligned}$$

700 ∎

701 **Lemma D.12 (Matrix Perturbation Bound)** *Suppose $\boldsymbol{A} > \boldsymbol{0}$. Then for any symmetric perturba-*
702 *tion matrix $\boldsymbol{\Delta}$ with $\|\boldsymbol{\Delta}\| \leqslant \frac{1}{2}\sigma_{\min}(\boldsymbol{A})$, it holds that*

$$\left\|(\boldsymbol{A} + \boldsymbol{\Delta})^{-1/2} - \boldsymbol{A}^{-1/2}\right\| \;\leqslant\; \frac{4\|\boldsymbol{\Delta}\|}{\sigma_{\min}^2(\boldsymbol{A})},$$

703 *where $\sigma_{\min}(\boldsymbol{A})$ denotes the minimum singular value of $\boldsymbol{A}$.*

704 **Proof** Let us denote $f(\boldsymbol{A}) = \boldsymbol{A}^{-1/2}$. Given a symmetric perturbation matrix $\boldsymbol{\Delta}$, by mean value
705 theorem, we have

$$\begin{aligned}
\left\|(\boldsymbol{A} + \boldsymbol{\Delta})^{-1/2} - \boldsymbol{A}^{-1/2}\right\| &\;=\; \left\|\int_0^1 \mathrm{D}f(\boldsymbol{A} + t\boldsymbol{\Delta})[\boldsymbol{\Delta}]dt\right\| \\
&\;\leqslant\; \left(\sup_{t\in[0,1]} \|\mathrm{D}f(\boldsymbol{A} + t\boldsymbol{\Delta})\|\right) \cdot \|\boldsymbol{\Delta}\|.
\end{aligned}$$

Thus, by Lemma D.11 and by using the fact that $\|\boldsymbol{\Delta}\| \leqslant \frac{1}{2}\sigma_{\min}(\boldsymbol{A})$, we have

$$\left\|(\boldsymbol{A} + \boldsymbol{\Delta})^{-1/2} - \boldsymbol{A}^{-1/2}\right\| \;\leqslant\; \left(\sup_{t \in [0,1]} \frac{1}{\sigma_{min}^2(\boldsymbol{A} + t\boldsymbol{\Delta})}\right) \|\boldsymbol{\Delta}\| \;\leqslant\; \frac{4\,\|\boldsymbol{\Delta}\|}{\sigma_{\min}^2(\boldsymbol{A})}.$$

∎

# E   Regularity Condition in Population

**Proposition E.1** *For every $i \in [n]$, define a set*

$$\boldsymbol{q} \in \mathcal{S}_\xi^{i+} \;\doteq\; \left\{\boldsymbol{q} \in \mathbb{R}^n \;\mid\; q_i > 0, \frac{q_i}{\|\boldsymbol{q}_{-i}\|_\infty} \geqslant \sqrt{1 + \xi}\right\}.$$

*Whenever $\theta \in \left(\frac{1}{n}, c_0\right)$ and $\mu \leqslant c_1 \min\left\{\theta, \frac{1}{\sqrt{n}}\right\}$, we have*

$$\left\langle \mathbb{E}\left[\operatorname{grad} \widetilde{f}(\boldsymbol{q})\right], q_i\boldsymbol{q} - \boldsymbol{e}_i\right\rangle \;\geqslant\; c_2\theta(1 - \theta)q_i\|\boldsymbol{q}_{-i}\|, \quad \sqrt{1 - q_i^2} \in [\mu, c_3] \tag{51}$$

$$\left\langle \mathbb{E}\left[\operatorname{grad} \widetilde{f}(\boldsymbol{q})\right], q_i\boldsymbol{q} - \boldsymbol{e}_i\right\rangle \;\geqslant\; c_2\theta(1 - \theta)q_i n^{-1}\|\boldsymbol{q}_{-i}\|, \quad \sqrt{1 - q_i^2} \in \left[c_3, \sqrt{\frac{n-1}{n}}\right], \tag{52}$$

*hold for any $\boldsymbol{q} \in \mathcal{S}_\xi^{i+}$ and each $i \in [n]$.*

**Remarks.**  For proving this result, we first introduce some basic notations. We use $\mathcal{I}$ to denote the generic support set of $\boldsymbol{q} \in \mathbb{S}^{n-1}$ of i.i.d. $\mathcal{B}(\theta)$ law. Since the landscape is symmetric for each $i \in [n]$, without loss of generality, it is enough to consider the case when $i = n$. We reparameterize $\boldsymbol{q} \in \mathbb{S}^{n-1}$ by

$$\boldsymbol{q}(\boldsymbol{w}): \; \boldsymbol{w} \mapsto \begin{bmatrix} \boldsymbol{w} \\ \sqrt{1 - \|\boldsymbol{w}\|^2} \end{bmatrix}, \tag{53}$$

where $\boldsymbol{w} \in \mathbb{R}^{n-1}$ with $\|\boldsymbol{w}\| \leqslant \sqrt{\frac{n-1}{n}}$. We write

$$\boldsymbol{q}_{\mathcal{I}} = \begin{bmatrix} \boldsymbol{w}_{\mathcal{J}} \\ q_n \mathbb{1}_{n \in \mathcal{I}} \end{bmatrix},$$

where we use $\mathcal{J}$ to denote the support set of $\boldsymbol{w}$ of i.i.d. $\mathcal{B}(\theta)$ law.

**Proof**  We denote

$$g(\boldsymbol{w}) = h_\mu\left(\boldsymbol{w}^\top \boldsymbol{x}_{-n} + x_n\sqrt{1 - \|\boldsymbol{w}\|^2}\right) \tag{54}$$

Note that if $\boldsymbol{e}_n$ is a local minimizer of $\mathbb{E}\left[\widetilde{f}(\boldsymbol{q})\right]$, then $\mathbb{E}\left[g(\boldsymbol{w})\right]$ has a corresponding local minimum at $\boldsymbol{0}$. Since $g(\cdot)$ satisfies chain rule when computing its gradient, we have

$$\langle\mathbb{E}\left[\nabla g(\boldsymbol{w})\right], \boldsymbol{w} - \boldsymbol{0}\rangle = \left\langle \begin{bmatrix} \boldsymbol{I}_{n-1} & \dfrac{-\boldsymbol{w}}{\sqrt{1 - \|\boldsymbol{w}\|^2}} \end{bmatrix} \nabla\mathbb{E}\left[\widetilde{f}(\boldsymbol{q})\right], \boldsymbol{w}\right\rangle$$

$$= \left\langle \mathbb{E}\left[\nabla\widetilde{f}(\boldsymbol{q})\right], \boldsymbol{q} - \frac{1}{q_n}\boldsymbol{e}_n\right\rangle = \frac{1}{q_n}\left\langle \mathbb{E}\left[\operatorname{grad}\widetilde{f}(\boldsymbol{q})\right], q_n\boldsymbol{q} - \boldsymbol{e}_n\right\rangle,$$

which gives

$$\left\langle \mathbb{E}\left[\operatorname{grad}\widetilde{f}(\boldsymbol{q})\right], q_n\boldsymbol{q} - \boldsymbol{e}_n\right\rangle = q_n \langle\mathbb{E}\left[\nabla g(\boldsymbol{w})\right], \boldsymbol{w}\rangle. \tag{55}$$

Thus, the above relationship implies that we can work on the "unconstrained" function $g(\boldsymbol{w})$ and establish the following: for any $\boldsymbol{q}(w) \in \mathcal{S}_\xi^{n+}$ with $\xi > 0$, or equivalently,

$$\|\boldsymbol{w}\|^2 + (1 + \xi)\|\boldsymbol{w}\|_\infty^2 \leqslant 1,$$

724 the following holds

$$\langle \nabla \mathbb{E}\left[g(\boldsymbol{w})\right], \boldsymbol{w} - \boldsymbol{0} \rangle \gtrsim \|\boldsymbol{w}\|.$$

725 When $\|\boldsymbol{w}\| \in [c_0\mu, c_1]$, Lemma E.4 implies that

$$\boldsymbol{w}^\top \nabla \mathbb{E}\left[g(\boldsymbol{w})\right] \geqslant c_2\theta(1-\theta)\|\boldsymbol{w}\|.$$

726 By Lemma E.5, we know that when $c_1 \leqslant \|\boldsymbol{w}\| \leqslant \sqrt{\frac{n-1}{n}}$,

$$\boldsymbol{w}^\top \nabla^2 \mathbb{E}\left[g(\boldsymbol{w})\right] \boldsymbol{w} \;\leqslant\; -c_3\theta(1-\theta)\|\boldsymbol{w}\|^2,$$

727 which implies concavity of $g(\boldsymbol{w})$ along the $\boldsymbol{w}$ direction. Let us denote $\boldsymbol{v} = \boldsymbol{w}/\|\boldsymbol{w}\|$, then the
728 directional concavity implies that

$$t\boldsymbol{v}^\top \nabla \mathbb{E}\left[g(t\boldsymbol{v})\right] \;\geqslant\; (t'\boldsymbol{v})^\top \nabla \mathbb{E}\left[g(t'\boldsymbol{v})\right] + c_4\theta(1-\theta)\left(t' - t\right),$$

729 for any $t, t' \in \left[c_1, \sqrt{\frac{n-1}{n}}\right]$. Choose $t' = \frac{\|\boldsymbol{w}\|}{\sqrt{\|\boldsymbol{w}\|^2 + \|\boldsymbol{w}\|_\infty^2}}$ and $t = \|\boldsymbol{w}\|$, by Lemma E.3, we know that

$$\boldsymbol{w}^\top \nabla \mathbb{E}\left[g(\boldsymbol{w})\right] \;\geqslant\; c_4\theta(1-\theta)\|\boldsymbol{w}\|\left(\frac{1}{\sqrt{\|\boldsymbol{w}\|^2 + \|\boldsymbol{w}\|_\infty^2}} - 1\right).$$

730 The function

$$h_{\boldsymbol{v}}(t) \;\doteq\; \frac{\|t\boldsymbol{v}\|}{\sqrt{\|t\boldsymbol{v}\|^2 + \|t\boldsymbol{v}\|_\infty^2}} - \|t\boldsymbol{v}\| = \frac{1}{\sqrt{1 + \|\boldsymbol{v}\|_\infty^2}} - t$$

731 is obviously monotonically decreasing w.r.t. $t$. Since $\boldsymbol{q} \in \mathcal{S}_\xi^{n+}$, we have

$$\|t\boldsymbol{v}\|^2 + (1+\xi)\|t\boldsymbol{v}\|_\infty^2 \;\leqslant\; 1 \implies t \leqslant \frac{1}{\sqrt{1 + (1+\xi)\|\boldsymbol{v}\|_\infty^2}}.$$

732 Therefore, we can uniformly lower bound $h_{\boldsymbol{v}}(t)$ by

$$h_{\boldsymbol{v}}(t) \;\geqslant\; \frac{1}{\sqrt{1 + \|\boldsymbol{v}\|_\infty^2}} - \frac{1}{\sqrt{1 + (1+\xi)\|\boldsymbol{v}\|_\infty^2}} \;\geqslant\; \xi\|\boldsymbol{v}\|_\infty^2 \geqslant \xi n^{-1}$$

733 Therefore, we have

$$\boldsymbol{w}^\top \nabla \mathbb{E}\left[g(\boldsymbol{w})\right] \;\geqslant\; c_4\xi\theta(1-\theta)n^{-1}\|\boldsymbol{w}\|,$$

734 when $\|\boldsymbol{w}\| \in \left[c_1, \sqrt{\frac{n-1}{n}}\right]$. Combining the bounds above, we obtain the desired results. ∎

735 **Lemma E.2** *Suppose* $\boldsymbol{g} \in \mathcal{N}(\boldsymbol{0}, \boldsymbol{I}_n)$, *we have*

$$\boldsymbol{w}^\top \nabla \mathbb{E}\left[g(\boldsymbol{w})\right] = \frac{1}{\mu}\mathbb{E}_{\mathcal{I}}\left[\left(\|\boldsymbol{q}_{\mathcal{I}}\|^2 - \mathbb{1}_{n\in\mathcal{I}}\right)\mathbb{P}\left(|\boldsymbol{q}_{\mathcal{I}}^\top \boldsymbol{g}| \leqslant \mu\right)\right]. \tag{56}$$

736 **Proof** In particular, exchange of gradient and expectation operator can again be justified. By simple
737 calculation, we obtain that

$$\nabla g(\boldsymbol{w}) \;=\; h'_\mu\left(\boldsymbol{q}^\top \boldsymbol{x}\right)\left(\boldsymbol{x}_{-n} - \frac{x_n}{q_n}\boldsymbol{w}\right) \;=\; \begin{cases} \frac{\boldsymbol{q}^\top \boldsymbol{x}}{\mu}\left(\boldsymbol{x}_{-n} - \frac{x_n}{q_n}\boldsymbol{w}\right), & |\boldsymbol{q}^\top \boldsymbol{x}| \leqslant \mu \\ \mathrm{sign}\left(\boldsymbol{q}^\top \boldsymbol{x}\right)\left(\boldsymbol{x}_{-n} - \frac{x_n}{q_n}\boldsymbol{w}\right), & |\boldsymbol{q}^\top \boldsymbol{x}| > \mu. \end{cases} \tag{57}$$

738 Thus, we obtain

$$\boldsymbol{w}^\top \nabla \mathbb{E}\left[g(\boldsymbol{w})\right]$$

$$= \mathbb{E}\left[\mathrm{sign}\left(\boldsymbol{q}^\top \boldsymbol{x}\right)\left(\boldsymbol{w}^\top \boldsymbol{x}_{-n} - \frac{x_n}{q_n}\|\boldsymbol{w}\|^2\right)\mathbb{1}_{|\boldsymbol{q}^\top \boldsymbol{x}| \geqslant \mu}\right] + \mathbb{E}\left[\frac{\boldsymbol{q}^\top \boldsymbol{x}}{\mu}\left(\boldsymbol{w}^\top \boldsymbol{x}_{-n} - \frac{x_n}{q_n}\|\boldsymbol{w}\|^2\right)\mathbb{1}_{|\boldsymbol{q}^\top \boldsymbol{x}| \leqslant \mu}\right]$$

$$= \mathbb{E}\left[\mathrm{sign}\left(\boldsymbol{q}^\top \boldsymbol{x}\right)\left(\boldsymbol{q}^\top \boldsymbol{x} - \frac{x_n}{q_n}\right)\mathbb{1}_{|\boldsymbol{q}^\top \boldsymbol{x}| \geqslant \mu}\right] + \frac{1}{\mu}\mathbb{E}\left[\left(\boldsymbol{q}^\top \boldsymbol{x}\right)\left(\boldsymbol{q}^\top \boldsymbol{x} - \frac{x_n}{q_n}\right)\mathbb{1}_{|\boldsymbol{q}^\top \boldsymbol{x}| \leqslant \mu}\right],$$

where we used the fact that

$$\boldsymbol{w}^\top \boldsymbol{x}_{-n} - \frac{x_n}{q_n} \|\boldsymbol{w}\|^2 = \boldsymbol{w}^\top \boldsymbol{x}_{-n} + q_n x_n - x_n \frac{\|\boldsymbol{w}\|^2 + q_n^2}{q_n} = \boldsymbol{q}^\top \boldsymbol{x} - \frac{x_n}{q_n}.$$

Let $Z = X + Y$, with

$$X = \boldsymbol{w}^\top \boldsymbol{x}_{-n} \sim \mathcal{N}(\boldsymbol{0}, \|\boldsymbol{w}_{\mathcal{J}}\|^2), \ Y = q_n x_n \sim \mathcal{N}(0, q_n^2 \mathbb{1}_{n \in \mathcal{I}}), \ Z \sim \mathcal{N}(\boldsymbol{0}, \|\boldsymbol{q}_{\mathcal{I}}\|^2). \qquad (58)$$

This gives

$$
\begin{aligned}
\boldsymbol{w}^\top \nabla \mathbb{E}\left[g(\boldsymbol{w})\right] \ &= \ \mathbb{E}\left[\left|\boldsymbol{q}^\top \boldsymbol{x}\right| \mathbb{1}_{|\boldsymbol{q}^\top \boldsymbol{x}| \geqslant \mu}\right] - \frac{1}{q_n} \mathbb{E}\left[\operatorname{sign}\left(\boldsymbol{q}^\top \boldsymbol{x}\right) x_n \mathbb{1}_{|\boldsymbol{q}^\top \boldsymbol{x}| \geqslant \mu}\right] \\
&\quad + \frac{1}{\mu} \mathbb{E}\left[\left(\boldsymbol{q}^\top \boldsymbol{x}\right)^2 \mathbb{1}_{|\boldsymbol{q}^\top \boldsymbol{x}| \leqslant \mu}\right] - \frac{1}{q_n \mu} \mathbb{E}\left[x_n \left(\boldsymbol{w}^\top \boldsymbol{x}_{-n}\right) \mathbb{1}_{|\boldsymbol{q}^\top \boldsymbol{x}| \leqslant \mu}\right] - \frac{1}{\mu} \mathbb{E}\left[x_n^2 \mathbb{1}_{|\boldsymbol{q}^\top \boldsymbol{x}| \leqslant \mu}\right] \\
&= \ \mathbb{E}\left[|Z| \mathbb{1}_{|Z| \geqslant \mu}\right] - \frac{1}{q_n^2} \mathbb{E}\left[\operatorname{sign}\left(X + Y\right) Y \mathbb{1}_{|X+Y| \geqslant \mu}\right] + \frac{1}{\mu} \mathbb{E}\left[Z^2 \mathbb{1}_{|Z| \leqslant \mu}\right] \\
&\quad - \frac{1}{\mu q_n^2} \mathbb{E}\left[XY \mathbb{1}_{|X+Y| \leqslant \mu}\right] - \frac{1}{\mu q_n^2} \mathbb{E}\left[Y^2 \mathbb{1}_{|X+Y| \leqslant \mu}\right].
\end{aligned}
$$

Now by Lemma D.7, we have

$$
\begin{aligned}
\mathbb{E}\left[|Z| \mathbb{1}_{|Z| \geqslant \mu}\right] \ &= \ \sqrt{\frac{2}{\pi}} \mathbb{E}_{\mathcal{I}}\left[\|\boldsymbol{q}_{\mathcal{I}}\| \exp\left(-\frac{\mu^2}{2 \|\boldsymbol{q}_{\mathcal{I}}\|^2}\right)\right] \\
\mathbb{E}\left[\operatorname{sign}\left(X + Y\right) Y \mathbb{1}_{|X+Y| \geqslant \mu}\right] \ &= \ q_n^2 \sqrt{\frac{2}{\pi}} \mathbb{E}\left[\frac{\mathbb{1}_{n \in \mathcal{I}}}{\|\boldsymbol{q}_{\mathcal{I}}\|} \exp\left(-\frac{\mu^2}{2 \|\boldsymbol{q}_{\mathcal{I}}\|^2}\right)\right] \\
\mathbb{E}\left[Z^2 \mathbb{1}_{|Z| \leqslant \mu}\right] \ &= \ -\mu \sqrt{\frac{2}{\pi}} \mathbb{E}_{\mathcal{I}}\left[\|\boldsymbol{q}_{\mathcal{I}}\| \exp\left(-\frac{\mu^2}{2 \|\boldsymbol{q}_{\mathcal{I}}\|^2}\right)\right] + \mathbb{E}_{\mathcal{I}}\left[\|\boldsymbol{q}_{\mathcal{I}}\|^2 \mathbb{P}\left(|\boldsymbol{q}_{\mathcal{I}}^\top \boldsymbol{g}| \leqslant \mu\right)\right] \\
\mathbb{E}\left[XY \mathbb{1}_{|X+Y| \leqslant \mu}\right] \ &= \ -\mu q_n^2 \sqrt{\frac{2}{\pi}} \mathbb{E}_{\mathcal{I}}\left[\frac{\mathbb{1}_{n \in \mathcal{I}} \|\boldsymbol{w}_{\mathcal{J}}\|^2}{\|\boldsymbol{q}_{\mathcal{I}}\|^3} \exp\left(-\frac{\mu^2}{2 \|\boldsymbol{q}_{\mathcal{I}}\|^2}\right)\right] \\
\mathbb{E}\left[Y^2 \mathbb{1}_{|X+Y| \leqslant \mu}\right] \ &= \ -\mu q_n^4 \sqrt{\frac{2}{\pi}} \mathbb{E}_{\mathcal{I}}\left[\frac{\mathbb{1}_{n \in \mathcal{I}}}{\|\boldsymbol{q}_{\mathcal{I}}\|^3} \exp\left(-\frac{\mu^2}{2 \|\boldsymbol{q}_{\mathcal{I}}\|^2}\right)\right] + q_n^2 \mathbb{E}_{\mathcal{I}}\left[\mathbb{1}_{n \in \mathcal{I}} \mathbb{P}\left(|\boldsymbol{q}_{\mathcal{I}}^\top \boldsymbol{g}| \leqslant \mu\right)\right]
\end{aligned}
$$

Putting the above calculations together and simplify, we obtain the desired result in (56).

■

**Lemma E.3** *When for any $\boldsymbol{w} \in \mathbb{R}^{n-1}$ satisfies $\|\boldsymbol{w}\|^2 + \|\boldsymbol{w}\|_\infty^2 \leqslant 1$, we have*

$$\boldsymbol{w}^\top \nabla \mathbb{E}\left[g(\boldsymbol{w})\right] \ \geqslant \ 0.$$

**Proof** From Lemma E.2, we know that

$$
\mu \cdot \boldsymbol{w}^\top \nabla \mathbb{E}\left[g(\boldsymbol{w})\right]
$$

$$
= \mathbb{E}_{\mathcal{I}}\left[\left(\|\boldsymbol{q}_{\mathcal{I}}\|^2 - \mathbb{1}_{n \in \mathcal{I}}\right) \mathbb{P}\left(\left|\boldsymbol{q}_{\mathcal{I}}^\top \boldsymbol{g}\right| \leqslant \mu\right)\right]
$$

$$
= \mathbb{E}_{\mathcal{J}}\left[(1-\theta)\|\boldsymbol{w}_{\mathcal{J}}\|^2 \mathbb{P}\left(\left|\boldsymbol{g}_{-n}^\top \boldsymbol{w}_{\mathcal{J}}\right| \leqslant \mu\right) - \theta \|\boldsymbol{w}_{\mathcal{J}^c}\|^2 \mathbb{P}\left(\left|\boldsymbol{g}_{-n}^\top \boldsymbol{w}_{\mathcal{J}} + q_n g_n\right| \leqslant \mu\right)\right]
$$

$$
= \mathbb{E}_{\mathcal{J}}\left[\int_{-\mu}^{\mu}\left(\frac{1-\theta}{\sqrt{2\pi}}\frac{\|\boldsymbol{w}_{\mathcal{J}}\|^2}{\|\boldsymbol{w}_{\mathcal{J}}\|}\exp\left(-\frac{t^2}{2\|\boldsymbol{w}_{\mathcal{J}}\|^2}\right) - \frac{\theta}{\sqrt{2\pi}}\frac{\|\boldsymbol{w}_{\mathcal{J}^c}\|^2}{\sqrt{1 - \|\boldsymbol{w}_{\mathcal{J}^c}\|^2}}\exp\left(\frac{-t^2}{2 - 2\|\boldsymbol{w}_{\mathcal{J}^c}\|^2}\right)\right)dt\right]
$$

$$
= \frac{1-\theta}{\sqrt{2\pi}}\sum_{i=1}^{n-1}\int_{-\mu}^{\mu}\mathbb{E}_{\mathcal{J}}\left[\frac{w_i^2 \mathbb{1}_{i \in \mathcal{J}}}{\sqrt{w_i^2 \mathbb{1}_{i \in \mathcal{J}} + \|\boldsymbol{w}_{\mathcal{J}\setminus\{i\}}\|^2}}\exp\left(-\frac{t^2}{2w_i^2 \mathbb{1}_{i \in \mathcal{J}} + 2\|\boldsymbol{w}_{\mathcal{J}\setminus\{i\}}\|^2}\right)\right]dt
$$

$$
- \frac{\theta}{\sqrt{2\pi}}\sum_{i=1}^{n-1}\int_{-\mu}^{\mu}\mathbb{E}_{\mathcal{J}}\left[\frac{w_i^2 \mathbb{1}_{i \notin \mathcal{J}}}{\sqrt{1 - w_i^2 \mathbb{1}_{i \notin \mathcal{J}} - \|\boldsymbol{w}_{\mathcal{J}^c\setminus\{i\}}\|^2}}\exp\left(-\frac{t^2}{2 - 2w_i^2 \mathbb{1}_{i \notin \mathcal{J}} - 2\|\boldsymbol{w}_{\mathcal{J}^c\setminus\{i\}}\|^2}\right)\right]dt
$$

$$
= \frac{(1-\theta)\theta}{\sqrt{2\pi}}\sum_{i=1}^{n-1}\int_{-\mu}^{\mu}\mathbb{E}_{\mathcal{J}}\left[\frac{w_i^2}{\sqrt{w_i^2 + \|\boldsymbol{w}_{\mathcal{J}\setminus\{i\}}\|^2}}\exp\left(-\frac{t^2}{2w_i^2 + 2\|\boldsymbol{w}_{\mathcal{J}\setminus\{i\}}\|^2}\right)\right]dt
$$

$$
- \frac{(1-\theta)\theta}{\sqrt{2\pi}}\sum_{i=1}^{n-1}\int_{-\mu}^{\mu}\mathbb{E}_{\mathcal{J}}\left[\frac{w_i^2}{\sqrt{1 - \|\boldsymbol{w}\|^2 + \|\boldsymbol{w}_{\mathcal{J}\setminus\{i\}}\|^2}}\exp\left(-\frac{t^2}{2 - 2\|\boldsymbol{w}\|^2 + 2\|\boldsymbol{w}_{\mathcal{J}\setminus\{i\}}\|^2}\right)\right]dt
$$

$$
= (1-\theta)\theta\sum_{i=1}^{n-1}w_i^2 \mathbb{E}_{\mathcal{J}}\left[\mathbb{P}\left(|Z_{i1}| \leqslant \mu\right) - \mathbb{P}\left(|Z_{i2}| \leqslant \mu\right)\right], \tag{59}
$$

where

$$
Z_{i1} \sim \mathcal{N}\left(0, w_i^2 + \|\boldsymbol{w}_{\mathcal{J}\setminus\{i\}}\|^2\right), \quad Z_{i2} \sim \mathcal{N}\left(0, 1 - \|\boldsymbol{w}\|^2 + \|\boldsymbol{w}_{\mathcal{J}\setminus\{i\}}\|^2\right). \tag{60}
$$

Since we have $1 - \|\boldsymbol{w}\|^2 \geqslant \|\boldsymbol{w}\|_\infty^2 \geqslant w_i^2$, the variance of $Z_i^2$ is larger than that of $Z_i^1$. Therefore, we have $\mathbb{P}\left(|Z_{i1}| \leqslant \mu\right) \geqslant \mathbb{P}\left(|Z_{i2}| \leqslant \mu\right)$ for each $i = 1, \cdots, n-1$. Hence, we obtain

$$
\boldsymbol{w}^\top \nabla \mathbb{E}\left[g(\boldsymbol{w})\right] = \frac{1}{\mu}\theta(1-\theta)\sum_{i=1}^{n-1}w_i^2 \mathbb{E}_{\mathcal{J}}\left[\mathbb{P}\left(|Z_{i1}| \leqslant \mu\right) - \mathbb{P}\left(|Z_{i2}| \leqslant \mu\right)\right] \geqslant 0.
$$

∎

**Lemma E.4** *For any $\boldsymbol{w}$ with $c_0\mu \leqslant \|\boldsymbol{w}\| \leqslant c_1$, we have*

$$
\boldsymbol{w}^\top \nabla \mathbb{E}\left[g(\boldsymbol{w})\right] \geqslant c\theta(1-\theta)\|\boldsymbol{w}\|
$$

**Proof** Recall from (59), we have

$$
\boldsymbol{w}^\top \nabla \mathbb{E}\left[g(\boldsymbol{w})\right] = \frac{1}{\mu}(1-\theta)\theta\sum_{i=1}^{n-1}w_i^2 \mathbb{E}_{\mathcal{J}}\left[\mathbb{P}\left(|Z_{i1}| \leqslant \mu\right) - \mathbb{P}\left(|Z_{i2}| \leqslant \mu\right)\right],
$$

where $Z_{i1}$ and $Z_{i2}$ are defined the same as (60). Let us denote

$$
Z_1 \sim \mathcal{N}\left(0, \|\boldsymbol{w}\|^2\right), \quad Z_2 \sim \mathcal{N}\left(0, 1 - \|\boldsymbol{w}\|^2\right).
$$

Since we have $\|\boldsymbol{w}\|^2 \geqslant w_i^2 + \|\boldsymbol{w}_{\mathcal{J}\setminus\{i\}}\|^2$, the variance of $Z_1$ is larger than that of $Z_{i1}$. Therefore, we have $\mathbb{P}\left(|Z_{i1}| \leqslant \mu\right) \geqslant \mathbb{P}\left(|Z_1| \leqslant \mu\right)$ for each $i = 1, \cdots, n-1$. By a similar argument, we have

$\mathbb{P}\left(|Z_{i2}| \leqslant \mu\right) \leqslant \mathbb{P}\left(|Z_2| \leqslant \mu\right)$ for each $i = 1, \cdots, n-1$. Thus, we obtain

$$
\begin{aligned}
&\mathbb{P}\left(|Z_{i1}| \leqslant \mu\right) - \mathbb{P}\left(|Z_{i2}| \leqslant \mu\right) \\
&\geqslant \mathbb{P}\left(|Z_1| \leqslant \mu\right) - \mathbb{P}\left(|Z_2| \leqslant \mu\right) \\
&= \sqrt{\frac{2}{\pi}} \frac{1}{\|\boldsymbol{w}\|} \int_0^\mu \exp\left(-\frac{t^2}{2\|\boldsymbol{w}\|^2}\right) dt - \sqrt{\frac{2}{\pi}} \frac{1}{\sqrt{1-\|\boldsymbol{w}\|^2}} \int_0^\mu \exp\left(-\frac{t^2}{2-2\|\boldsymbol{w}\|^2}\right) dt \\
&\geqslant \sqrt{\frac{2}{\pi}} \left[ \frac{1}{\|\boldsymbol{w}\|} \int_0^\mu \left(1 - \frac{t^2}{2\|\boldsymbol{w}\|^2}\right) dt - \frac{\mu}{\sqrt{1-\|\boldsymbol{w}\|^2}} \right] \\
&= \sqrt{\frac{2}{\pi}} \left[ \frac{1}{\|\boldsymbol{w}\|} \left(\mu - \frac{1}{6}\frac{\mu^3}{\|\boldsymbol{w}\|^2}\right) - \frac{\mu}{\sqrt{1-\|\boldsymbol{w}\|^2}} \right] \\
&\geqslant \mu\sqrt{\frac{2}{\pi}} \left(\frac{1}{\|\boldsymbol{w}\|} - 2\frac{1}{\sqrt{1-\|\boldsymbol{w}\|^2}}\right) \geqslant \frac{\mu}{2\sqrt{2\pi}}\frac{1}{\|\boldsymbol{w}\|}
\end{aligned}
\tag{61}
$$

where we used the fact that $\mu/\sqrt{3} \leqslant \|\boldsymbol{w}\| \leqslant 1/\sqrt{17}$ for the last two inequalities. Plugging (61) back into (59) gives

$$
\begin{aligned}
\boldsymbol{w}^\top \nabla \mathbb{E}\left[g(\boldsymbol{w})\right] &= \frac{1}{\mu}(1-\theta)\theta \sum_{i=1}^{n-1} w_i^2 \mathbb{E}_{\mathcal{J}}\left[\mathbb{P}\left(|Z_{i1}| \leqslant \mu\right) - \mathbb{P}\left(|Z_{i2}| \leqslant \mu\right)\right] \\
&\geqslant \frac{(1-\theta)\theta}{2\sqrt{2\pi}\|\boldsymbol{w}\|} \sum_{i=1}^{n-1} w_i^2 = \frac{1}{2\sqrt{2\pi}}(1-\theta)\theta\|\boldsymbol{w}\|,
\end{aligned}
$$

as desired. ∎

**Lemma E.5** *When $\mu \leqslant c_0 \min\left\{\frac{1}{\sqrt{n}}, \theta\right\}$ and $\theta \in \left(\frac{1}{n}, c_1\right)$, we have*

$$
\boldsymbol{w}^\top \nabla^2 \mathbb{E}\left[g(\boldsymbol{w})\right] \boldsymbol{w} \leqslant -c_2\theta(1-\theta)\|\boldsymbol{w}\|^2
$$

*for all $\boldsymbol{w}$ with $c_3 \leqslant \|\boldsymbol{w}\| \leqslant \sqrt{\frac{n-1}{n}}$. Here, $c_0$, $c_1$, $c_2$, and $c_3$ are some numerical constants.*

**Proof** Since the expectation and derivative are exchangeable, we have

$$
\boldsymbol{w}^\top \nabla^2 \mathbb{E}\left[g(\boldsymbol{w})\right] \boldsymbol{w} = \boldsymbol{w}^\top \mathbb{E}\left[\nabla^2 g(\boldsymbol{w})\right] \boldsymbol{w}.
$$

From (57), we obtain

$$
\boldsymbol{w}^\top \nabla^2 g(\boldsymbol{w})\boldsymbol{w} = \begin{cases} \frac{1}{\mu}\left[\left(\boldsymbol{q}^\top \boldsymbol{x}\right)^2 - \frac{x_n}{q_n}\left(\boldsymbol{q}^\top \boldsymbol{x}\right) - \frac{x_n}{q_n^3}\left(\boldsymbol{x}_{-n}^\top \boldsymbol{w}\right)\right], & |\boldsymbol{q}^\top \boldsymbol{x}| \leqslant \mu \\ -\frac{x_n}{q_n^3}\|\boldsymbol{w}\|^2 \operatorname{sign}\left(\boldsymbol{q}^\top \boldsymbol{x}\right), & |\boldsymbol{q}^\top \boldsymbol{x}| \geqslant \mu. \end{cases}
$$

Thus, we have

$$
\begin{aligned}
\mathbb{E}\left[\boldsymbol{w}^\top \nabla^2 g(\boldsymbol{w})\boldsymbol{w} \mathbb{1}_{|\boldsymbol{q}^\top \boldsymbol{x}|\geqslant\mu}\right] &= -\frac{\|\boldsymbol{w}\|^2}{q_n^4} \mathbb{E}\left[q_n x_n \operatorname{sign}\left(\boldsymbol{q}^\top \boldsymbol{x}\right) \mathbb{1}_{|\boldsymbol{q}^\top \boldsymbol{x}|\geqslant\mu}\right] \\
&= -\sqrt{\frac{2}{\pi}}\frac{\|\boldsymbol{w}\|^2}{q_n^2} \mathbb{E}_{\mathcal{I}}\left[\frac{\mathbb{1}_{n\in\mathcal{I}}}{\|\boldsymbol{q}_{\mathcal{I}}\|} \exp\left(-\frac{\mu^2}{2\|\boldsymbol{q}_{\mathcal{I}}\|^2}\right)\right]
\end{aligned}
$$

and

$$
\begin{aligned}
&\mathbb{E}\left[\boldsymbol{w}^\top \nabla^2 g(\boldsymbol{w})\boldsymbol{w} \mathbb{1}_{|\boldsymbol{q}^\top \boldsymbol{x}|\leqslant\mu}\right] \\
&= \frac{1}{\mu}\mathbb{E}\left[\left(\boldsymbol{q}^\top \boldsymbol{x}\right)^2 \mathbb{1}_{|\boldsymbol{q}^\top \boldsymbol{x}|\leqslant\mu}\right] - \frac{1}{\mu}\mathbb{E}\left[\frac{x_n}{q_n}\left(\boldsymbol{q}^\top \boldsymbol{x}\right) \mathbb{1}_{|\boldsymbol{q}^\top \boldsymbol{x}|\leqslant\mu}\right] - \frac{1}{\mu}\mathbb{E}\left[\frac{x_n}{q_n^3}\left(\boldsymbol{x}_{-n}^\top \boldsymbol{w}\right) \mathbb{1}_{|\boldsymbol{q}^\top \boldsymbol{x}|\leqslant\mu}\right] \\
&= \frac{1}{\mu}\mathbb{E}\left[Z^2 \mathbb{1}_{|Z|\leqslant\mu}\right] - \frac{1}{\mu q_n^2}\mathbb{E}\left[Y^2 \mathbb{1}_{|X+Y|\leqslant\mu}\right] - \frac{1}{\mu}\left(\frac{1}{q_n^2} + \frac{1}{q_n^4}\right)\mathbb{E}\left[XY \mathbb{1}_{|X+Y|\leqslant\mu}\right],
\end{aligned}
$$

where $X$, $Y$ and $Z = X + Y$ are defined the same as (58). Similar to Lemma E.2, by using Lemma D.7, we obtain

$$\mathbb{E}\left[\boldsymbol{w}^\top \nabla^2 g(\boldsymbol{w})\boldsymbol{w}\mathbb{1}_{|\boldsymbol{q}^\top \boldsymbol{x}|\leqslant \mu}\right]$$

$$= -\sqrt{\frac{2}{\pi}}\mathbb{E}_{\mathcal{I}}\left[\|\boldsymbol{q}_{\mathcal{I}}\|\exp\left(-\frac{\mu^2}{2\|\boldsymbol{q}_{\mathcal{I}}\|^2}\right)\right] + \frac{1}{\mu}\mathbb{E}\left[\left(\|\boldsymbol{q}_{\mathcal{I}}\|^2 - \mathbb{1}_{n\in\mathcal{I}}\right)\mathbb{P}\left(|\boldsymbol{q}_{\mathcal{I}}^\top \boldsymbol{g}|\leqslant \mu\right)\right]$$

$$+ \sqrt{\frac{2}{\pi}}\mathbb{E}_{\mathcal{I}}\left[\frac{q_n^2\,\mathbb{1}_{n\in\mathcal{I}}}{\|\boldsymbol{q}_{\mathcal{I}}\|^3}\exp\left(-\frac{\mu^2}{2\|\boldsymbol{q}_{\mathcal{I}}\|^2}\right)\right] + \sqrt{\frac{2}{\pi}}\left(1 + \frac{1}{q_n^2}\right)\mathbb{E}_{\mathcal{I}}\left[\frac{\|\boldsymbol{w}_{\mathcal{J}}\|^2\,\mathbb{1}_{n\in\mathcal{I}}}{\|\boldsymbol{q}_{\mathcal{I}}\|^3}\exp\left(-\frac{\mu^2}{2\|\boldsymbol{q}_{\mathcal{I}}\|^2}\right)\right].$$

Combining the results above and using integral by parts, we obtain

$$\boldsymbol{w}^\top \nabla^2 \mathbb{E}\left[g(\boldsymbol{w})\right]\boldsymbol{w}$$

$$= -\sqrt{\frac{2}{\pi}}\mathbb{E}_{\mathcal{I}}\left[\frac{\mathbb{1}_{n\in\mathcal{I}}}{\|\boldsymbol{q}_{\mathcal{I}}\|^3}\exp\left(-\frac{\mu^2}{2\|\boldsymbol{q}_{\mathcal{I}}\|^2}\right)\right] + 2\sqrt{\frac{2}{\pi}}\mathbb{E}_{\mathcal{I}}\left[\frac{\mathbb{1}_{n\in\mathcal{I}}}{\|\boldsymbol{q}_{\mathcal{I}}\|}\exp\left(-\frac{\mu^2}{2\|\boldsymbol{q}_{\mathcal{I}}\|^2}\right)\right]$$

$$- \sqrt{\frac{2}{\pi}}\mathbb{E}_{\mathcal{I}}\left[\|\boldsymbol{q}_{\mathcal{I}}\|\exp\left(-\frac{\mu^2}{2\|\boldsymbol{q}_{\mathcal{I}}\|^2}\right)\right] + \frac{1}{\mu}\mathbb{E}\left[\left(\|\boldsymbol{q}_{\mathcal{I}}\|^2 - \mathbb{1}_{n\in\mathcal{I}}\right)\mathbb{P}\left(|\boldsymbol{q}_{\mathcal{I}}^\top \boldsymbol{g}|\leqslant \mu\right)\right]$$

$$= -\sqrt{\frac{2}{\pi}}\mathbb{E}_{\mathcal{I}}\left[\frac{\|\boldsymbol{w}_{\mathcal{J}^c}\|^2\,\mathbb{1}_{n\in\mathcal{I}}}{\|\boldsymbol{q}_{\mathcal{I}}\|^3}\exp\left(-\frac{\mu^2}{2\|\boldsymbol{q}_{\mathcal{I}}\|^2}\right)\right]$$

$$+ \sqrt{\frac{2}{\pi}}\mathbb{E}_{\mathcal{I}}\left[\frac{\mathbb{1}_{n\in\mathcal{I}}}{\|\boldsymbol{q}_{\mathcal{I}}\|}\left(\exp\left(-\frac{\mu^2}{2\|\boldsymbol{q}_{\mathcal{I}}\|^2}\right) - \frac{\|\boldsymbol{q}_{\mathcal{I}}\|}{\mu}\int_0^{\mu/\|\boldsymbol{q}_{\mathcal{I}}\|}\exp\left(-t^2/2\right)dt\right)\right]$$

$$- \sqrt{\frac{2}{\pi}}\mathbb{E}_{\mathcal{I}}\left[\|\boldsymbol{q}_{\mathcal{I}}\|\left(\exp\left(-\frac{\mu^2}{2\|\boldsymbol{q}_{\mathcal{I}}\|^2}\right) - \frac{\|\boldsymbol{q}_{\mathcal{I}}\|}{\mu}\int_0^{\mu/\|\boldsymbol{q}_{\mathcal{I}}\|}\exp\left(-t^2/2\right)dt\right)\right]$$

$$= -\sqrt{\frac{2}{\pi}}\mathbb{E}_{\mathcal{I}}\left[\|\boldsymbol{w}_{\mathcal{J}^c}\|^2\,\frac{\mathbb{1}_{n\in\mathcal{I}}}{\|\boldsymbol{q}_{\mathcal{I}}\|^3}\exp\left(-\frac{\mu^2}{2\|\boldsymbol{q}_{\mathcal{I}}\|^2}\right)\right] - \frac{1}{\mu}\sqrt{\frac{2}{\pi}}\mathbb{E}_{\mathcal{I}}\left[\mathbb{1}_{n\in\mathcal{I}}\int_0^{\mu/\|\boldsymbol{q}_{\mathcal{I}}\|}t^2\exp\left(-t^2/2\right)dt\right]$$

$$+ \frac{1}{\mu}\sqrt{\frac{2}{\pi}}\mathbb{E}_{\mathcal{I}}\left[\|\boldsymbol{q}_{\mathcal{I}}\|^2\int_0^{\mu/\|\boldsymbol{q}_{\mathcal{I}}\|}t^2\exp\left(-t^2/2\right)dt\right]$$

$$\leqslant -\sqrt{\frac{2}{\pi}}\mathbb{E}_{\mathcal{I}}\left[\|\boldsymbol{w}_{\mathcal{J}^c}\|^2\,\frac{\mathbb{1}_{n\in\mathcal{I}}}{\|\boldsymbol{q}_{\mathcal{I}}\|^3}\exp\left(-\frac{\mu^2}{2\|\boldsymbol{q}_{\mathcal{I}}\|^2}\right)\right] + \frac{1}{\mu}\sqrt{\frac{2}{\pi}}\int_0^{\mu}t^2\mathbb{E}_{\mathcal{I}}\left[\frac{1}{\|\boldsymbol{q}_{\mathcal{I}}\|}\exp\left(-\frac{t^2}{2\|\boldsymbol{q}_{\mathcal{I}}\|^2}\right)\right]dt.$$

First, when $\sqrt{\frac{n-1}{n}}\geqslant \|\boldsymbol{w}\|\geqslant c_0$, we have

$$\mathbb{E}_{\mathcal{I}}\left[\|\boldsymbol{w}_{\mathcal{J}^c}\|^2\,\frac{\mathbb{1}_{n\in\mathcal{I}}}{\|\boldsymbol{q}_{\mathcal{I}}\|^3}\exp\left(-\frac{\mu^2}{2\|\boldsymbol{q}_{\mathcal{I}}\|^2}\right)\right]$$

$$= \theta\mathbb{E}_{\mathcal{J}}\left[\|\boldsymbol{w}_{\mathcal{J}^c}\|^2\,\frac{1}{\left(q_n^2 + \|\boldsymbol{w}_{\mathcal{J}}\|^2\right)^{3/2}}\exp\left(-\frac{\mu^2}{2\left(q_n^2 + \|\boldsymbol{w}_{\mathcal{J}}\|^2\right)}\right)\right]$$

$$\geqslant \theta\mathbb{E}_{\mathcal{J}}\left[\|\boldsymbol{w}_{\mathcal{J}^c}\|^2\exp\left(-\frac{\mu^2}{2q_n^2 + 2\|\boldsymbol{w}_{\mathcal{J}}\|^2}\right)\right]$$

$$\geqslant \theta\mathbb{E}_{\mathcal{J}}\left[\|\boldsymbol{w}_{\mathcal{J}^c}\|^2\exp\left(-\frac{\mu^2}{2q_n^2}\right)\right] \geqslant c_1\theta(1-\theta)\|\boldsymbol{w}\|^2.$$

Second, notice that the function

$$h(x) = x^{-1}\exp\left(-\frac{t^2}{2x^2}\right), \quad x\in[0,1]$$

reaches the maximum when $x = t$. Thus, we have

$$\frac{1}{\mu}\sqrt{\frac{2}{\pi}}\int_0^\mu t^2 \mathbb{E}_\mathcal{I}\left[\frac{1}{\|\boldsymbol{q}_\mathcal{I}\|}\exp\left(-\frac{t^2}{2\|\boldsymbol{q}_\mathcal{I}\|^2}\right)\right]dt \leqslant \frac{1}{\mu}\sqrt{\frac{2}{\pi}}\int_0^\mu t\exp\left(-\frac{1}{2}\right)dt \leqslant \frac{1}{\sqrt{2\pi}}e^{-1/2}\mu.$$

Therefore, when $\mu \leqslant \frac{1}{n} \leqslant \theta$, we have

$$\boldsymbol{w}^\top \nabla^2 \mathbb{E}\left[g(\boldsymbol{w})\right]\boldsymbol{w} \leqslant -c_2\theta(1-\theta)\|\boldsymbol{w}\|^2$$

for any $\sqrt{\frac{n-1}{n}} \geqslant \|\boldsymbol{w}\| \geqslant c_0$.

∎

# F  Negative Curvature on Gradient in Population

**Proposition F.1** *Suppose $\theta \geqslant \frac{1}{n}$. Given any index $i \in [n]$, when $\mu \leqslant \frac{1}{\sqrt{3}n}$, we have*

$$\left\langle \operatorname{grad}\mathbb{E}\left[\tilde{f}(\boldsymbol{q})\right], \frac{1}{q_j}\boldsymbol{e}_j - \frac{1}{q_i}\boldsymbol{e}_i \right\rangle \geqslant \frac{\theta(1-\theta)}{4n}\frac{\xi}{1+\xi},$$

*holds for all $\boldsymbol{q} \in \mathcal{S}_\xi^{i+}$ and any $q_j$ such that $j \neq i$ and $q_j^2 \geqslant \frac{1}{3}q_i^2$*

**Proof** Without loss of generality, let us consider the case $i = n$. For any $j \neq n$, we have

$$\left\langle \operatorname{grad}\mathbb{E}\left[\tilde{f}(\boldsymbol{q})\right], \frac{1}{q_j}\boldsymbol{e}_j - \frac{1}{q_n}\boldsymbol{e}_n \right\rangle$$

$$= \left(\frac{1}{q_j}\boldsymbol{e}_j - \frac{1}{q_n}\boldsymbol{e}_n\right)^\top \mathcal{P}_{\boldsymbol{q}^\perp}\mathbb{E}\left[\boldsymbol{x}\cdot h_\mu'(\boldsymbol{x}^\top\boldsymbol{q})\right]$$

$$= \left(\frac{1}{q_j}\boldsymbol{e}_j - \frac{1}{q_n}\boldsymbol{e}_n\right)^\top \mathbb{E}\left[\boldsymbol{x}\cdot h_\mu'(\boldsymbol{x}^\top\boldsymbol{q})\right].$$

Let

$$Z = Z_1 + Z_2, \quad Z_1 = q_i x_i \sim \mathcal{N}(0, (b_i q_i)^2), \quad Z_2 = \boldsymbol{q}_{-i}^\top \boldsymbol{x}_{-i} \sim \mathcal{N}(0, \|\boldsymbol{q}_{-i}\odot\boldsymbol{b}_{-i}\|^2).$$

Notice that for every $i \in [n]$, we have

$$\frac{1}{q_i}\boldsymbol{e}_i^\top\mathbb{E}\left[\boldsymbol{x}\cdot h_\mu'(\boldsymbol{x}^\top\boldsymbol{q})\right]$$

$$= \frac{1}{q_i^2}\frac{1}{\mu}\mathbb{E}\left[Z_1^2\mathbb{1}_{|Z_1+Z_2|\leqslant\mu}\right] + \frac{1}{q_i^2}\frac{1}{\mu}\mathbb{E}\left[Z_1 Z_2\mathbb{1}_{|Z_1+Z_2|\leqslant\mu}\right] + \frac{1}{q_i^2}\mathbb{E}\left[Z_1\operatorname{sign}(Z_1+Z_2)\mathbb{1}_{|Z_1+Z_2|\geqslant\mu}\right].$$

By Lemma D.7, we have

$$\mathbb{E}\left[Z_1^2\mathbb{1}_{|Z_1+Z_2|\leqslant\mu}\right] = -\sqrt{\frac{2}{\pi}}\mu\mathbb{E}_\mathcal{I}\left[\frac{q_i^4\mathbb{1}_{i\in\mathcal{I}}}{\|\boldsymbol{q}_\mathcal{I}\|^3}\exp\left(-\frac{\mu^2}{2\|\boldsymbol{q}_\mathcal{I}\|^2}\right)\right]$$

$$+ \mathbb{E}\left[q_i^2\mathbb{1}_{i\in\mathcal{I}}\mathbb{P}\left(|Z|\leqslant\mu\right)\right],$$

$$\mathbb{E}\left[Z_1 Z_2\mathbb{1}_{|Z_1+Z_2|\leqslant\mu}\right] = -\sqrt{\frac{2}{\pi}}\mu\mathbb{E}_\mathcal{I}\left[\frac{q_i^2\mathbb{1}_{i\in\mathcal{I}}\|(\boldsymbol{q}_{-i})_\mathcal{J}\|^2}{\|\boldsymbol{q}_\mathcal{I}\|^3}\exp\left(-\frac{\mu^2}{2\|\boldsymbol{q}_\mathcal{I}\|^2}\right)\right]$$

$$\mathbb{E}\left[Z_1\operatorname{sign}(Z_1+Z_2)\mathbb{1}_{|Z_1+Z_2|\geqslant\mu}\right] = \sqrt{\frac{2}{\pi}}\mathbb{E}_\mathcal{I}\left[\frac{q_i^2\mathbb{1}_{i\in\mathcal{I}}}{\|\boldsymbol{q}_\mathcal{I}\|}\exp\left(-\frac{\mu^2}{2\|\boldsymbol{q}_\mathcal{I}\|^2}\right)\right].$$

Combining the results above, we obtain

$$\frac{1}{q_i}\boldsymbol{e}_i^\top\mathbb{E}\left[\boldsymbol{x}\cdot h_\mu'(\boldsymbol{x}^\top\boldsymbol{q})\right] = \frac{1}{\mu}\mathbb{E}\left[\mathbb{1}_{i\in\mathcal{I}}\mathbb{P}\left(|Z|\leqslant\mu\right)\right].$$

Therefore, we have

$$
\left\langle \operatorname{grad} \mathbb{E}\left[\widetilde{f}(\boldsymbol{q})\right], \frac{1}{q_j}\boldsymbol{e}_j - \frac{1}{q_n}\boldsymbol{e}_n \right\rangle
$$

$$
= \frac{1}{\mu}\left(\mathbb{E}\left[\mathbb{1}_{j\in\mathcal{I}}\mathbb{P}\left(|Z|\leqslant\mu\right)\right] - \mathbb{E}\left[\mathbb{1}_{n\in\mathcal{I}}\mathbb{P}\left(|Z|\leqslant\mu\right)\right]\right)
$$

$$
= \frac{\theta}{\mu}\sqrt{\frac{2}{\pi}}\mathbb{E}_{\mathcal{I}}\left[\frac{1}{\sqrt{q_j^2 + \left\|\boldsymbol{q}_{\mathcal{I}\backslash j}\right\|^2}}\int_0^\mu \exp\left(-\frac{t^2}{q_j^2 + \left\|\boldsymbol{q}_{\mathcal{I}\backslash j}\right\|^2}\right)dt\right]
$$

$$
- \frac{\theta}{\mu}\sqrt{\frac{2}{\pi}}\mathbb{E}_{\mathcal{I}}\left[\frac{1}{\sqrt{q_n^2 + \left\|\boldsymbol{q}_{\mathcal{I}\backslash n}\right\|^2}}\int_0^\mu \exp\left(-\frac{t^2}{q_n^2 + \left\|\boldsymbol{q}_{\mathcal{I}\backslash n}\right\|^2}\right)dt\right]
$$

$$
= \frac{\theta(1-\theta)}{\mu}\sqrt{\frac{2}{\pi}}\mathbb{E}_{\mathcal{I}}\left[\frac{1}{\sqrt{q_j^2 + \left\|\boldsymbol{q}_{\mathcal{I}\backslash\{j,n\}}\right\|^2}}\int_0^\mu \exp\left(-\frac{t^2}{q_j^2 + \left\|\boldsymbol{q}_{\mathcal{I}\backslash\{j,n\}}\right\|^2}\right)dt\right]
$$

$$
- \frac{\theta(1-\theta)}{\mu}\sqrt{\frac{2}{\pi}}\mathbb{E}_{\mathcal{I}}\left[\frac{1}{\sqrt{q_n^2 + \left\|\boldsymbol{q}_{\mathcal{I}\backslash\{j,n\}}\right\|^2}}\int_0^\mu \exp\left(-\frac{t^2}{q_n^2 + \left\|\boldsymbol{q}_{\mathcal{I}\backslash\{j,n\}}\right\|^2}\right)dt\right]
$$

$$
= \frac{\theta(1-\theta)}{\mu}\mathbb{E}_{\mathcal{I}}\left[\operatorname{erf}\left(\frac{\mu}{\sqrt{q_i^2 + \left\|\boldsymbol{q}_{\mathcal{I}\backslash\{j,n\}}\right\|^2}}\right) - \operatorname{erf}\left(\frac{\mu}{\sqrt{q_n^2 + \left\|\boldsymbol{q}_{\mathcal{I}\backslash\{j,n\}}\right\|^2}}\right)\right]
$$

where $\operatorname{erf}(x)$ is the Gaussian error function

$$
\operatorname{erf}(x) = \frac{1}{\sqrt{2\pi}}\int_{-x}^x \exp\left(-t^2/2\right)dt = \sqrt{\frac{2}{2\pi}}\int_0^x \exp\left(-t^2/2\right)dt, \quad x\geqslant 0.
$$

When $\mu\leqslant\frac{1}{\sqrt{3n}}$ such that $\frac{\mu}{\sqrt{q_n^2 + \left\|\boldsymbol{q}_{\mathcal{I}\backslash\{j,n\}}\right\|^2}}\leqslant 1$ for $\boldsymbol{q}\in\mathcal{S}_\xi^{n+}$, by Taylor approximation we have

$$
\operatorname{erf}\left(\frac{\mu}{\sqrt{q_i^2 + \left\|\boldsymbol{q}_{\mathcal{I}\backslash\{j,n\}}\right\|^2}}\right) - \operatorname{erf}\left(\frac{\mu}{\sqrt{q_n^2 + \left\|\boldsymbol{q}_{\mathcal{I}\backslash\{j,n\}}\right\|^2}}\right)
$$

$$
\geqslant \frac{\mu}{2}\left[\frac{1}{\sqrt{q_i^2 + \left\|\boldsymbol{q}_{\mathcal{I}\backslash\{j,n\}}\right\|^2}} - \frac{1}{\sqrt{q_n^2 + \left\|\boldsymbol{q}_{\mathcal{I}\backslash\{j,n\}}\right\|^2}}\right] = \frac{\mu}{4}\int_{q_i^2}^{q_n^2}\frac{1}{\left(t^2 + \left\|\boldsymbol{q}_{\mathcal{I}\backslash\{j,n\}}\right\|^2\right)^{3/2}}dt.
$$

Therefore, we have

$$
\left\langle \operatorname{grad} \mathbb{E}\left[\widetilde{f}(\boldsymbol{q})\right], \frac{1}{q_j}\boldsymbol{e}_j - \frac{1}{q_n}\boldsymbol{e}_n \right\rangle
$$

$$
\geqslant \frac{\theta(1-\theta)}{4}\int_{q_i^2}^{q_n^2}\frac{1}{\left(t^2 + \left\|\boldsymbol{q}_{\mathcal{I}\backslash\{j,n\}}\right\|^2\right)^{3/2}}dt
$$

$$
\geqslant \frac{\theta(1-\theta)}{4}\left(q_n^2 - \left\|\boldsymbol{q}_{-n}\right\|_\infty^2\right) \geqslant \frac{\theta(1-\theta)}{4}\frac{\xi}{1+\xi}q_n^2 \geqslant \frac{\theta(1-\theta)}{4n}\frac{\xi}{1+\xi}.
$$

This gives the desired result. ∎

# G  Gradient Concentration

In this section, we uniformly bound the deviation between the empirical process $\operatorname{grad}\widetilde{f}(\boldsymbol{q})$ and its mean $\mathbb{E}\left[\operatorname{grad}\widetilde{f}(\boldsymbol{q})\right]$ over the sphere. Namely, we show the following

 **Proposition G.1** *For every $i \in [n]$ and any $\delta \in (0, 1)$, when*

$$p \geqslant C\delta^{-2}n \log\left(\frac{\theta n}{\mu\delta}\right), \tag{62}$$

792 *we have*

$$\sup_{\boldsymbol{q} \in \mathbb{S}^{n-1}} \left| \left\langle \operatorname{grad} \widetilde{f}(\boldsymbol{q}) - \mathbb{E}\left[\operatorname{grad} \widetilde{f}(\boldsymbol{q})\right], \boldsymbol{e}_i \right\rangle \right| \leqslant \delta$$

793 *holds with probability at least $1 - np^{-c_1\theta n} - n \exp\left(-c_2 p\delta^2\right)$, for any $\boldsymbol{e}_i$. Here, $c_1$, $c_2$, and $C$ are*
794 *some universal positive numerical constants.*

795 **Remarks.** Here, our bound is loose by roughly a factor of $n$ because of the looseness in handling
796 the probabilistic dependency due to the convolution measurement. We believe this bound can be
797 improved by an order of $\mathcal{O}(n)$ using more advanced probability tools, such as decoupling and
798 chaining [44–46].

799 **Proof** First, note that

$$\widetilde{f}(\boldsymbol{q}) \;=\; \frac{1}{np} \sum_{i=1}^{p} H_\mu\left(\boldsymbol{C}_{\boldsymbol{x}_i}\boldsymbol{q}\right), \quad \operatorname{grad} \widetilde{f}(\boldsymbol{q}) \;=\; \frac{1}{np}\mathcal{P}_{\boldsymbol{q}^\perp} \sum_{i=1}^{p} \boldsymbol{C}_{\boldsymbol{x}_i}^\top h_\mu'\left(\boldsymbol{C}_{\boldsymbol{x}_i}\boldsymbol{q}\right). \tag{63}$$

800 Thus, we have

$$\left\langle \operatorname{grad} \widetilde{f}(\boldsymbol{q}) - \mathbb{E}\left[\operatorname{grad} \widetilde{f}(\boldsymbol{q})\right], \boldsymbol{e}_n \right\rangle$$
$$= \frac{1}{np} \sum_{i=1}^{p} \sum_{j=0}^{n-1} \left[ \left\langle \mathcal{P}_{\boldsymbol{q}^\perp} s_j\left[\breve{\boldsymbol{x}}_i\right], \boldsymbol{e}_n \right\rangle h_\mu'\left(s_j\left[\breve{\boldsymbol{x}}_i\right]^\top \boldsymbol{q}\right) - \mathbb{E}\left[\left(\boldsymbol{e}_n^\top \mathcal{P}_{\boldsymbol{q}^\perp}\boldsymbol{x}\right) h_\mu'\left(\boldsymbol{x}^\top\boldsymbol{q}\right)\right] \right].$$

801 This is a summation of dependent random variables, which is very difficult to show measurement
802 concentration in general. We alleviate this difficulty by only considering a partial summation of
803 independent random variables, namely,

$$\mathcal{L}(\boldsymbol{q}) \;=\; \frac{1}{p}\frac{1}{\|\boldsymbol{P}_{\boldsymbol{q}^\perp}\boldsymbol{e}_n\|} \sum_{i=1}^{p} \left[ \left\langle \mathcal{P}_{\boldsymbol{q}^\perp}\boldsymbol{x}_i, \boldsymbol{e}_n \right\rangle h_\mu'\left(\boldsymbol{x}_i^\top\boldsymbol{q}\right) - \mathbb{E}\left[\left(\boldsymbol{e}_n^\top \mathcal{P}_{\boldsymbol{q}^\perp}\boldsymbol{x}\right) h_\mu'\left(\boldsymbol{x}^\top\boldsymbol{q}\right)\right] \right],$$

804 where $\boldsymbol{x}_i \sim_{i.i.d.} \mathcal{BG}(\theta)$. Note that the bound of $\mathcal{L}(\boldsymbol{q})$ automatically gives an upper bound of
805 $\left\langle \operatorname{grad} \widetilde{f}(\boldsymbol{q}) - \mathbb{E}\left[\operatorname{grad} \widetilde{f}(\boldsymbol{q})\right], \boldsymbol{e}_n \right\rangle$ in distribution. To uniformly control $\mathcal{L}(\boldsymbol{q})$ over the sphere, we
806 first consider controlling $\mathcal{L}(\boldsymbol{q})$ for a fixed $\boldsymbol{q} \in \mathbb{S}^{n-1}$. For each $\ell = 1, 2, \cdots$, we have the moments

$$\mathbb{E}\left[\left|\left\langle \mathcal{P}_{\boldsymbol{q}^\perp}\boldsymbol{x}_i, \boldsymbol{e}_n \right\rangle h_\mu'\left(\boldsymbol{x}_i^\top\boldsymbol{q}\right)\right|^\ell\right] \;\leqslant\; \mathbb{E}\left[\left|\boldsymbol{e}_n^\top \mathcal{P}_{\boldsymbol{q}^\perp}\boldsymbol{x}_i\right|^\ell\right] \;=\; \mathbb{E}\left[|Z_i|^\ell\right],$$

807 where conditioned on the Bernoulli distribution, we have $Z_i \sim \mathcal{N}\left(0, \left\|\left(\mathcal{P}_{\boldsymbol{q}^\perp}\boldsymbol{e}_n\right)_{\mathcal{J}}\right\|^2\right)$. By Lemma
808 D.1, we have

$$\mathbb{E}\left[\left|\left\langle \mathcal{P}_{\boldsymbol{q}^\perp}\boldsymbol{x}_i, \boldsymbol{e}_n \right\rangle h_\mu'\left(\boldsymbol{x}_i^\top\boldsymbol{q}\right)\right|^\ell\right] \;\leqslant\; \mathbb{E}_{\mathcal{J}}\left[(\ell-1)!!\left\|\left(\mathcal{P}_{\boldsymbol{q}^\perp}\boldsymbol{e}_n\right)_{\mathcal{J}}\right\|^\ell\right] \leqslant \frac{\ell!}{2}\left\|\boldsymbol{P}_{\boldsymbol{q}^\perp}\boldsymbol{e}_n\right\|^\ell,$$

809 where we used the fact that $\left|h_\mu'(z)\right| \leqslant 1$ for any $z$. Thus, we are controlling the concentration of
810 summation of sub-Gaussian r.v., for which we have

$$\mathbb{P}\left(|\mathcal{L}(\boldsymbol{q})| \geqslant t\right) \;\leqslant\; \exp\left(-C\frac{pt^2}{2}\right).$$

811 Next, we turn this point-wise concentration into a uniform bound for all $\boldsymbol{q} \in \mathbb{S}^{n-1}$ via a standard
812 covering argument. Let $\mathcal{N}(\varepsilon)$ be an $\varepsilon$-net of the sphere, whose cardinality can be controlled by

$$|\mathcal{N}(\varepsilon)| \;\leqslant\; \left(\frac{3}{\varepsilon}\right)^{n-1}.$$

Thus, we have

$$\mathbb{P}\left(\sup_{\boldsymbol{q}\in\mathcal{N}(\varepsilon)}|\mathcal{L}(\boldsymbol{q})|\geqslant t\right)\leqslant\left(\frac{3}{\varepsilon}\right)^{n-1}\exp\left(-\frac{pt^2}{2+2t}\right).$$

For any point $\boldsymbol{q}\in\mathbb{S}^{n-1}$, it can written as $\boldsymbol{q}=\boldsymbol{q}'+\boldsymbol{e}$, where $\boldsymbol{q}'\in\mathcal{N}(\varepsilon)$ and $\|\boldsymbol{e}\|\leqslant\varepsilon$. Now we control the all points over the sphere through the Lipschitz property of $\mathcal{L}$.

$$\sup_{\boldsymbol{q}\in\mathbb{S}^{n-1}}|\mathcal{L}(\boldsymbol{q})|$$
$$=\sup_{\boldsymbol{q}'\in\mathcal{N}(\varepsilon),\|\boldsymbol{e}\|\leqslant\varepsilon}\left|\mathcal{L}(\boldsymbol{q}'+\boldsymbol{e})\right|$$
$$\leqslant\sup_{\boldsymbol{q}'\in\mathcal{N}(\varepsilon)}\left|\mathcal{L}(\boldsymbol{q}')\right|+\underbrace{\sup_{\boldsymbol{q}'\in\mathcal{N}(\varepsilon),\|\boldsymbol{e}\|\leqslant\varepsilon}\left|\mathbb{E}\left[\left(\boldsymbol{e}_n^{\top}\mathcal{P}_{(\boldsymbol{q}'+\boldsymbol{e})^{\perp}}\boldsymbol{x}-\boldsymbol{e}_n^{\top}\mathcal{P}_{(\boldsymbol{q}')^{\perp}}\boldsymbol{x}\right)h'_{\mu}\left(\boldsymbol{x}^{\top}\boldsymbol{q}'\right)\right]\right|}_{\mathcal{L}_1}$$
$$+\underbrace{\sup_{\boldsymbol{q}'\in\mathcal{N}(\varepsilon),\|\boldsymbol{e}\|\leqslant\varepsilon}\left|\mathbb{E}\left[\left(\boldsymbol{e}_n^{\top}\mathcal{P}_{(\boldsymbol{q}'+\boldsymbol{e})^{\perp}}\boldsymbol{x}\right)\left(h'_{\mu}\left(\boldsymbol{x}^{\top}(\boldsymbol{q}'+\boldsymbol{e})\right)-h'_{\mu}\left(\boldsymbol{x}^{\top}\boldsymbol{q}'\right)\right)\right]\right|}_{\mathcal{L}_2}$$
$$+\underbrace{\sup_{\boldsymbol{q}'\in\mathcal{N}(\varepsilon),\|\boldsymbol{e}\|\leqslant\varepsilon}\left|\frac{1}{p}\sum_{i=1}^{p}\left[\boldsymbol{e}_n^{\top}\mathcal{P}_{(\boldsymbol{q}'+\boldsymbol{e})^{\perp}}\boldsymbol{x}_i-\boldsymbol{e}_n^{\top}\mathcal{P}_{(\boldsymbol{q}')^{\perp}}\boldsymbol{x}_i\right]h'_{\mu}(\boldsymbol{x}_i^{\top}\boldsymbol{q}')\right|}_{\mathcal{L}_3}$$
$$+\underbrace{\sup_{\boldsymbol{q}'\in\mathcal{N}(\varepsilon),\|\boldsymbol{e}\|\leqslant\varepsilon}\left|\frac{1}{p}\sum_{i=1}^{p}\left(\boldsymbol{e}_n^{\top}\mathcal{P}_{(\boldsymbol{q}'+\boldsymbol{e})^{\perp}}\boldsymbol{x}_i\right)\left[h'_{\mu}\left(\boldsymbol{x}_i^{\top}(\boldsymbol{q}'+\boldsymbol{e})\right)-h'_{\mu}\left(\boldsymbol{x}_i^{\top}\boldsymbol{q}'\right)\right]\right|}_{\mathcal{L}_4}.$$

By Lipschitz continuity and the fact that $h'_{\mu}(z)\leqslant 1$ for any $z$, we obtain

$$\mathcal{L}_1\leqslant\sup_{\boldsymbol{q}'\in\mathcal{N}(\varepsilon),\|\boldsymbol{e}\|\leqslant\varepsilon}\sqrt{\theta}\left\|\left(\mathcal{P}_{(\boldsymbol{q}'+\boldsymbol{e})^{\perp}}-\mathcal{P}_{(\boldsymbol{q}')^{\perp}}\right)\boldsymbol{e}_n\right\|\leqslant 3\sqrt{\theta}\varepsilon$$

$$\mathcal{L}_2\leqslant\sup_{\boldsymbol{q}'\in\mathcal{N}(\varepsilon),\|\boldsymbol{e}\|\leqslant\varepsilon}\frac{1}{\mu}\mathbb{E}\left[\|\boldsymbol{x}\|\|\boldsymbol{x}^{\top}\boldsymbol{e}\|\right]\leqslant\frac{\theta n}{\mu}\varepsilon.$$

For each $\boldsymbol{x}_i$, we know that $\boldsymbol{x}_i=\boldsymbol{g}_i\odot\boldsymbol{b}_i$ with $\boldsymbol{g}_i\sim\mathcal{N}(\boldsymbol{0},\boldsymbol{I})$ and $\boldsymbol{b}_i\sim_{i.i.d.}\mathcal{B}(\theta)$. By Gaussian concentration inequality, we know that for each $\boldsymbol{x}_i$,

$$\mathbb{P}\left(\|\boldsymbol{x}_i\|-\sqrt{\theta n}\geqslant t\right)\leqslant\mathbb{P}\left(\|\boldsymbol{x}_i\|-\mathbb{E}\left[\|\boldsymbol{x}_i\|\right]\geqslant t\right)\leqslant\exp\left(-\frac{t^2}{2\|\boldsymbol{b}_i\|_{\infty}}\right)\leqslant\exp\left(-\frac{t^2}{2}\right).$$

Therefore, by a union bound, we have

$$\max_{1\leqslant i\leqslant p}\|\boldsymbol{x}_i\|\leqslant 5\sqrt{\theta n\log p}$$

holds with probability at least $1-p^{-8\theta n}$. Therefore, w.h.p we have

$$\mathcal{L}_3\leqslant\left(\max_{1\leqslant i\leqslant p}\|\boldsymbol{x}_i\|\right)\sup_{\boldsymbol{q}'\in\mathcal{N}(\varepsilon),\|\boldsymbol{e}\|\leqslant\varepsilon}\left\|\mathcal{P}_{(\boldsymbol{q}'+\boldsymbol{e})^{\perp}}-\mathcal{P}_{(\boldsymbol{q}')^{\perp}}\right\|\leqslant 15\sqrt{\theta n\log p}\varepsilon,$$

$$\mathcal{L}_4\leqslant\frac{1}{\mu}\left(\max_{1\leqslant i\leqslant p}\|\boldsymbol{x}_i\|^2\right)\sup_{\boldsymbol{q}'\in\mathcal{N}(\varepsilon),\|\boldsymbol{e}\|\leqslant\varepsilon}\|\boldsymbol{e}\|\leqslant 25\frac{\theta n\log p}{\mu}\varepsilon.$$

Combining the bounds above, choose $\varepsilon=\frac{\mu t}{c\theta n\log p}$, we have

$$\sup_{\boldsymbol{q}\in\mathbb{S}^{n-1}}|\mathcal{L}(\boldsymbol{q})|\leqslant\sup_{\boldsymbol{q}'\in\mathcal{N}(\varepsilon)}\left|\mathcal{L}(\boldsymbol{q}')\right|+c\frac{\theta n\log p}{\mu}\varepsilon\leqslant 2t$$

holds with probability at least

$$1-p^{-8\theta n}-\exp\left(-C\frac{pt^2}{2}+c'n\log\left(\frac{\theta n}{\mu t}\right)\right).$$

823    Thus, applying a union bound, we obtain the desired result holding for every $i \in [n]$.     ∎

824    Similarly, we also show the following result.

825    **Corollary G.2** *For any $\delta \in (0, 1)$, when*

$$p \geqslant C\delta^{-2}n^2 \log\left(\frac{\theta n}{\mu\delta}\right), \tag{64}$$

826    *we have*

$$\sup_{\boldsymbol{q} \in \mathbb{S}^{n-1}} \left\| \mathrm{grad}\, \widetilde{f}(\boldsymbol{q}) - \mathbb{E}\left[ \mathrm{grad}\, \widetilde{f}(\boldsymbol{q}) \right] \right\| \ \leqslant \ \delta,$$

$$\sup_{\boldsymbol{q} \in \mathbb{S}^{n-1}} \left\| \nabla \widetilde{f}(\boldsymbol{q}) - \mathbb{E}\left[ \nabla \widetilde{f}(\boldsymbol{q}) \right] \right\| \ \leqslant \ \delta,$$

827    *hold with probability at least $1 - p^{-c_1\theta n} - n\exp\left(-c_2 p\delta^2\right)$. Here, $c_1$, $c_2$, and $C$ are some universal*
828    *positive numerical constants.*

829    **Proof** From Proposition G.1, we know that when $p \geqslant C_0 \varepsilon^{-2} n \log\left(\frac{\theta n}{\mu\varepsilon}\right)$,

$$\sup_{q \in \mathbb{S}^{n-1}} \left\| \mathrm{grad}\, \widetilde{f}(\boldsymbol{q}) - \mathbb{E}\left[ \mathrm{grad}\, \widetilde{f}(\boldsymbol{q}) \right] \right\|^2$$

$$\leqslant \sum_{i=1}^{n} \sup_{q \in \mathbb{S}^{n-1}} \left| \left\langle \mathrm{grad}\, \widetilde{f}(\boldsymbol{q}) - \mathbb{E}\left[ \mathrm{grad}\, \widetilde{f}(\boldsymbol{q}) \right], \boldsymbol{e}_i \right\rangle \right|^2 \ \leqslant \ n\varepsilon^2.$$

830    holds with probability at least $1 - p^{-c_1\theta n} - n\exp\left(-c_2 p\delta^2\right)$. Therefore, by letting $\delta = \sqrt{n}\varepsilon$, w.h.p.
831    we have

$$\sup_{q \in \mathbb{S}^{n-1}} \left\| \mathrm{grad}\, \widetilde{f}(\boldsymbol{q}) - \mathbb{E}\left[ \mathrm{grad}\, \widetilde{f}(\boldsymbol{q}) \right] \right\| \leqslant \delta,$$

832    whenever $p \geqslant C\delta^{-2}n^2 \log\left(\frac{\theta n}{\mu\delta}\right)$. By a similar argument, we can also provide the same bound for
833    $\sup_{\boldsymbol{q} \in \mathcal{S}^{n-1}} \left\| \nabla \widetilde{f}(\boldsymbol{q}) - \mathbb{E}\left[ \nabla \widetilde{f}(\boldsymbol{q}) \right] \right\|$.     ∎

834    **Corollary G.3** *For each $i \in [n]$ and any $\delta \in (0, 1)$, when $p \geqslant C\delta^{-2}n \log\left(\frac{\theta n}{\mu\delta}\right)$, we have*

$$\sup_{q \in \mathbb{S}^{n-1}} \left| \left\langle \mathrm{grad}\, \widetilde{f}(\boldsymbol{q}), \boldsymbol{e}_i \right\rangle \right| \ \leqslant \ 1 + \delta,$$

835    *hold with probability at least $1 - np^{-c_1\theta n} - n\exp\left(-c_2 p\delta^2\right)$. Here, $c_1$, $c_2$, and $C$ are some universal*
836    *positive numerical constants.*

837    **Proof** For any $\boldsymbol{q} \in \mathbb{S}^{n-1}$ and every $i \in [n]$, we have

$$\mathbb{E}\left[ \left| \left\langle \mathrm{grad}\, \widetilde{f}(\boldsymbol{q}), \boldsymbol{e}_i \right\rangle \right| \right] \ = \ \mathbb{E}\left[ \left| \left( \boldsymbol{e}_i^\top \mathcal{P}_{\boldsymbol{q}^\perp} \boldsymbol{x} \right) \cdot h_\mu'(\boldsymbol{x}^\top \boldsymbol{q}) \right| \right] \leqslant \mathbb{E}\left[ \left\| \boldsymbol{e}_i^\top \mathcal{P}_{\boldsymbol{q}^\perp} \boldsymbol{x} \right\| \right] \leqslant 1.$$

838    Thus, we have

$$\sup_{q \in \mathbb{S}^{n-1}} \left| \left\langle \mathrm{grad}\, \widetilde{f}(\boldsymbol{q}) - \mathbb{E}\left[ \mathrm{grad}\, \widetilde{f}(\boldsymbol{q}) \right], \boldsymbol{e}_i \right\rangle \right|$$

$$\geqslant \sup_{q \in \mathbb{S}^{n-1}} \left( \left| \left\langle \mathrm{grad}\, \widetilde{f}(\boldsymbol{q}), \boldsymbol{e}_i \right\rangle \right| - \mathbb{E}\left[ \left| \left\langle \mathrm{grad}\, \widetilde{f}(\boldsymbol{q}), \boldsymbol{e}_i \right\rangle \right| \right] \right)$$

$$\geqslant \sup_{q \in \mathbb{S}^{n-1}} \left| \left\langle \mathrm{grad}\, \widetilde{f}(\boldsymbol{q}), \boldsymbol{e}_i \right\rangle \right| - \sup_{q \in \mathbb{S}^{n-1}} \mathbb{E}\left[ \left| \left\langle \mathrm{grad}\, \widetilde{f}(\boldsymbol{q}), \boldsymbol{e}_i \right\rangle \right| \right].$$

839    Therefore, by using the result in Proposition G.1, we obtain the desired result.     ∎

**Corollary G.4** *For any $\delta \in (0, 1)$, when $p$ satisfies (64), we have*

$$\sup_{q \in \mathbb{S}^{n-1}} \left\| \operatorname{grad} \widetilde{f}(\boldsymbol{q}) \right\| \leqslant \sqrt{\theta n} + \delta,$$

*hold with probability at least $1 - p^{-c_1 \theta n} - n \exp\left(-c_2 p \delta^2\right)$. Here, $c_1, c_2,$ and $C$ are some universal positive numerical constants.*

**Proof** For any $\boldsymbol{q} \in \mathbb{S}^{n-1}$, we have

$$\mathbb{E}\left[\left\| \operatorname{grad} \widetilde{f}(\boldsymbol{q}) \right\|\right] = \mathbb{E}\left[\left\| \mathcal{P}_{\boldsymbol{q}^\perp} \boldsymbol{x} h'_\mu(\boldsymbol{x}^\top \boldsymbol{q}) \right\|\right] \leqslant \mathbb{E}\left[\|\boldsymbol{x}\|\right] \leqslant \sqrt{\theta n}.$$

Note that

$$\sup_{q \in \mathbb{S}^{n-1}} \left\| \operatorname{grad} \widetilde{f}(\boldsymbol{q}) - \mathbb{E}\left[\operatorname{grad} \widetilde{f}(\boldsymbol{q})\right] \right\| \geqslant \sup_{q \in \mathbb{S}^{n-1}} \left( \left\| \operatorname{grad} \widetilde{f}(\boldsymbol{q}) \right\| - \mathbb{E}\left[\left\| \operatorname{grad} \widetilde{f}(\boldsymbol{q}) \right\|\right] \right)$$

$$\geqslant \sup_{q \in \mathbb{S}^{n-1}} \left\| \operatorname{grad} \widetilde{f}(\boldsymbol{q}) \right\| - \sup_{q \in \mathbb{S}^{n-1}} \mathbb{E}\left[\left\| \operatorname{grad} \widetilde{f}(\boldsymbol{q}) \right\|\right].$$

Thus, by using the result in Corollary G.2, we obtain the desired result. ∎

# H  Preconditioning

In this section, given the Riemannian gradient of $\widetilde{f}(\boldsymbol{q})$ and its preconditioned variant

$$\operatorname{grad} \widetilde{f}(\boldsymbol{q}) = \frac{1}{np} \mathcal{P}_{\boldsymbol{q}^\perp} \sum_{i=1}^{p} \boldsymbol{C}_{\boldsymbol{x}_i}^\top h'_\mu\left(\boldsymbol{C}_{\boldsymbol{x}_i} \boldsymbol{q}\right),$$

$$\operatorname{grad} f(\boldsymbol{q}) = \frac{1}{np} \mathcal{P}_{\boldsymbol{q}^\perp} \sum_{i=1}^{p} \left(\boldsymbol{R}\boldsymbol{Q}^{-1}\right)^\top \boldsymbol{C}_{\boldsymbol{x}_i}^\top h'_\mu\left(\boldsymbol{C}_{\boldsymbol{x}_i}\left(\boldsymbol{R}\boldsymbol{Q}^{-1}\right)\boldsymbol{q}\right),$$

we prove the following result.

**Proposition H.1** *Suppose $\theta \geqslant \frac{1}{n}$. For any $\delta \in (0, 1)$, whenever*

$$p \geqslant C \frac{\kappa^8 n}{\mu^2 \theta \delta^2 \sigma_{\min}^2} \log^4 n \log\left(\frac{\theta n}{\mu}\right),$$

*we have*

$$\sup_{\boldsymbol{q} \in \mathbb{S}^{n-1}} \left\| \operatorname{grad} \widetilde{f}(\boldsymbol{q}) - \operatorname{grad} f(\boldsymbol{q}) \right\| \leqslant \delta$$

*holds with probability at least $1 - c_1 p^{-c_2 n \theta} - n^{-c_3} - n e^{-c_4 \theta n p}$. Here, $\kappa$ and $\sigma_{\min}$ denote the condition number and minimum singular value of $\boldsymbol{C_a}$, and $c_1, c_2, c_3, c_4$ and $C$ are some positive numerical constants.*

**Proof** Notice that

$$\boldsymbol{R} = \boldsymbol{C_a} \left(\frac{1}{\theta np} \sum_{i=1}^{p} \boldsymbol{C}_{\boldsymbol{y}_i}^\top \boldsymbol{C}_{\boldsymbol{y}_i}\right)^{-1/2}, \quad \boldsymbol{Q} = \boldsymbol{C_a}\left(\boldsymbol{C_a}^\top \boldsymbol{C_a}\right)^{-1/2}$$

so that

$$\boldsymbol{R}\boldsymbol{Q}^{-1} = \boldsymbol{C_a}\left(\frac{1}{\theta np} \sum_{i=1}^{p} \boldsymbol{C}_{\boldsymbol{y}_i}^\top \boldsymbol{C}_{\boldsymbol{y}_i}\right)^{-1/2} \left(\boldsymbol{C_a}^\top \boldsymbol{C_a}\right)^{1/2} \boldsymbol{C_a}^{-1}.$$

Thus, we have

$$\sup_{\boldsymbol{q}\in\mathbb{S}^{n-1}}\left\|\operatorname{grad}\widetilde{f}(\boldsymbol{q})-\operatorname{grad}f(\boldsymbol{q})\right\|$$

$$\leqslant\frac{1}{np}\left\|\mathcal{P}_{\boldsymbol{q}^{\perp}}\left(\boldsymbol{I}-\left(\boldsymbol{RQ}^{-1}\right)\right)^{\top}\sum_{i=1}^{p}\boldsymbol{C}_{\boldsymbol{x}_i}^{\top}h_{\mu}'\left(\boldsymbol{C}_{\boldsymbol{x}_i}\boldsymbol{q}\right)\right\|$$

$$+\frac{1}{np}\left\|\mathcal{P}_{\boldsymbol{q}^{\perp}}\left(\boldsymbol{RQ}^{-1}\right)^{\top}\sum_{i=1}^{p}\boldsymbol{C}_{\boldsymbol{x}_i}^{\top}\left[h_{\mu}'\left(\boldsymbol{C}_{\boldsymbol{x}_i}\boldsymbol{q}\right)-h_{\mu}'\left(\boldsymbol{C}_{\boldsymbol{x}_i}\left(\boldsymbol{RQ}^{-1}\right)\boldsymbol{q}\right)\right]\right\|$$

$$\leqslant\left\|\boldsymbol{I}-\boldsymbol{RQ}^{-1}\right\|\left\|\nabla\widetilde{f}(\boldsymbol{q})\right\|+\left\|\boldsymbol{RQ}^{-1}\right\|\left\|\frac{1}{np}\sum_{i=1}^{p}\boldsymbol{C}_{\boldsymbol{x}_i}^{\top}\left[h_{\mu}'\left(\boldsymbol{C}_{\boldsymbol{x}_i}\boldsymbol{q}\right)-h_{\mu}'\left(\boldsymbol{C}_{\boldsymbol{x}_i}\left(\boldsymbol{RQ}^{-1}\right)\boldsymbol{q}\right)\right]\right\|$$

$$\leqslant\left\|\boldsymbol{I}-\boldsymbol{RQ}^{-1}\right\|\left\|\nabla\widetilde{f}(\boldsymbol{q})\right\|+\frac{1}{\mu\sqrt{n}}\left\|\boldsymbol{RQ}^{-1}\right\|\left(\max_{1\leqslant i\leqslant p}\|\boldsymbol{x}_i\|\,\|\boldsymbol{Fx}_i\|_{\infty}\right)\left\|\boldsymbol{I}-\boldsymbol{RQ}^{-1}\right\|. \qquad (65)$$

Here, by Lemma H.4, for any given $\varepsilon\in(0,1)$, when $p\geqslant C\frac{\kappa^8}{\theta\varepsilon^2\sigma_{\min}^2(\boldsymbol{C_a})}\log^3 n$, we have

$$\left\|\boldsymbol{RQ}^{-1}-\boldsymbol{I}\right\|\leqslant\varepsilon,\quad\left\|\boldsymbol{RQ}^{-1}\right\|\leqslant 1+\varepsilon, \qquad (66)$$

holding with probability at least $1-p^{-c_1 n\theta}-n^{-c_2}$. On the other hand, by Gaussian concentration inequality and a union bound, we have

$$\max_{1\leqslant i\leqslant p}\|\boldsymbol{x}_i\|\leqslant 4\sqrt{n\log p},\quad\max_{1\leqslant i\leqslant p}\|\boldsymbol{Fx}_i\|_{\infty}\leqslant 4\sqrt{n\log p}, \qquad (67)$$

hold with probability at least $1-p^{-c_3 n}$. By Corollary G.4, when $p\geqslant C_2\theta^{-1}n\log\left(\frac{\theta n}{\mu}\right)$, we have

$$\sup_{q\in\mathbb{S}^{n-1}}\left\|\operatorname{grad}\widetilde{f}(\boldsymbol{q})\right\|\leqslant 2\sqrt{\theta n} \qquad (68)$$

holds with probability at least $1-p^{-c_4\theta n}-ne^{-c_5\theta np}$. Plugging the bounds in (66) and (67) into (65), we obtain

$$\sup_{\boldsymbol{q}\in\mathbb{S}^{n-1}}\left\|\operatorname{grad}\widetilde{f}(\boldsymbol{q})-\operatorname{grad}f(\boldsymbol{q})\right\|\leqslant\varepsilon\left[2\sqrt{\theta n}+\frac{16\sqrt{n}\log p}{\mu}\cdot(1+\varepsilon)\right].$$

By a change of variable, we obtain the desired result. ∎

**Lemma H.2** *When $\theta\geqslant 1/n$,*

$$\left\|\frac{1}{\theta np}\sum_{i=1}^{p}\boldsymbol{C}_{\boldsymbol{x}_i}^{\top}\boldsymbol{C}_{\boldsymbol{x}_i}-\boldsymbol{I}\right\|\leqslant t \qquad (69)$$

*holds with probability at least $1-p^{-c_1 n\theta}-n\exp\left(-c_2\min\left\{\frac{pt^2}{\theta\log p},\frac{pt}{\sqrt{\theta\log p}}\right\}\right)$ for some numerical constants $c_1,c_2>0$.*

**Proof**

Notice that

$$\boldsymbol{C}_{\boldsymbol{x}_i}^{\top}\boldsymbol{C}_{\boldsymbol{x}_i}=\boldsymbol{F}^*\operatorname{diag}\left(|\boldsymbol{Fx}_i|^{\odot 2}\right)\boldsymbol{F}.$$

Then

$$\left\|\frac{1}{\theta np}\sum_{i=1}^{p}\boldsymbol{C}_{\boldsymbol{x}_i}^{\top}\boldsymbol{C}_{\boldsymbol{x}_i}-\boldsymbol{I}\right\|=\left\|\boldsymbol{F}^*\left(\operatorname{diag}\left(\frac{1}{\theta np}\sum_{i=1}^{p}|\boldsymbol{Fx}_i|^{\odot 2}\right)-\boldsymbol{F}^{-1}(\boldsymbol{F}^*)^{-1}\right)\boldsymbol{F}\right\|$$

$$=\left\|\frac{1}{\theta np}\sum_{i=1}^{p}|\boldsymbol{Fx}_i|^{\odot 2}-\boldsymbol{1}\right\|_{\infty}. \qquad (70)$$

Let $\boldsymbol{x}_i = \boldsymbol{b}_i \odot \boldsymbol{g}_i$ with $\boldsymbol{b}_i \sim_{i.i.d.} \mathcal{B}(\theta)$ and $\boldsymbol{g}_i \sim \mathcal{N}(\boldsymbol{0}, \boldsymbol{I})$, and let us define events

$$\mathcal{E}_{i,j} \doteq \left\{ \|\boldsymbol{b}_i \odot \boldsymbol{f}_j\|^2 \leqslant 5n\sqrt{\theta \log p} \right\}, \quad 1 \leqslant i \leqslant p, \ 1 \leqslant j \leqslant n.$$

We use $\mathcal{E}_j = \bigcap_{i=1}^{p} \mathcal{E}_{i,j}$. For each individual $i$ and $j$, by the Hoeffding's inequality, we have

$$\mathbb{P}\left(\mathcal{E}_{i,j}^c\right) \leqslant \exp\left(-8n\theta \log p\right)$$

For each $j = 1, \cdots, n$, by conditional probability and union bound, we have

$$
\mathbb{P}\left( \left| \frac{1}{\theta n p} \sum_{i=1}^{p} |\boldsymbol{f}_j^* \boldsymbol{x}_i|^2 - 1 \right| \geqslant t \right) \leqslant \mathbb{P}\left( \bigcup_{i=1}^{p} \mathcal{E}_{i,j}^c \right) + \mathbb{P}\left( \left| \frac{1}{\theta n p} \sum_{i=1}^{p} |\boldsymbol{f}_j^* \boldsymbol{x}_i|^2 - 1 \right| \geqslant t \mid \mathcal{E}_j \right)
$$

$$
\leqslant \sum_{i=1}^{p} \mathbb{P}\left(\mathcal{E}_{i,j}^c\right) + \mathbb{P}\left( \left| \frac{1}{\theta n p} \sum_{i=1}^{p} |\boldsymbol{f}_j^* \boldsymbol{x}_i|^2 - 1 \right| \geqslant t \mid \mathcal{E}_j \right)
$$

$$
\leqslant p e^{-8n\theta \log p} + \mathbb{P}\left( \left| \frac{1}{\theta n p} \sum_{i=1}^{p} |\boldsymbol{f}_j^* \boldsymbol{x}_i|^2 - 1 \right| \geqslant t \mid \mathcal{E}_j \right). \quad (71)
$$

For the second term, since $\boldsymbol{x}_i \sim \mathcal{BG}(\theta)$, we have

$$\boldsymbol{f}_j^* \boldsymbol{x}_i = \sum_{k=1}^{n} f_{ji} b_{ik} g_{ik} \sim \mathcal{N}\left(0, \|\boldsymbol{b}_i \odot \boldsymbol{f}_j\|^2\right)$$

for all $\ell \geqslant 1$, by Lemma D.1, we have

$$
\mathbb{E}\left[ (\theta n)^{-\ell} |\boldsymbol{f}_j^* \boldsymbol{x}_i|^{2\ell} \mid \mathcal{E}_{i,j} \right] = \frac{(2\ell - 1)!!}{(\theta n)^{\ell}} \mathbb{E}\left[ \|\boldsymbol{b} \odot \boldsymbol{f}\|^{2\ell} \mid \mathcal{E}_{i,j} \right]
$$

$$
\leqslant \frac{\ell!}{2} 10^{\ell} \theta^{-\ell/2} \log^{\ell/2} p.
$$

Thus, by Bernstein inequality in Lemma D.3, we have

$$
\mathbb{P}\left( \left| \frac{1}{\theta n p} \sum_{i=1}^{p} |\boldsymbol{f}_j^* \boldsymbol{x}_i|^2 - 1 \right| \geqslant t \mid \mathcal{E}_j \right) \leqslant \exp\left( -\frac{p t^2}{200 \log p + 20\sqrt{\log p} t} \right)
$$

$$
\leqslant \exp\left( -\min\left\{ \frac{p t^2}{400 \theta \log p}, \frac{p t}{40\sqrt{\theta \log p}} \right\} \right). \quad (72)
$$

Plugging (72) into (71), we obtain

$$
\left| \frac{1}{\theta n p} \sum_{i=1}^{p} |\boldsymbol{f}_j^* \boldsymbol{x}_i|^2 - 1 \right| \leqslant t
$$

holds with high probability for each $j = 1, \cdots, n$. We apply a union bound to control the $\ell_\infty$-norm in (70), and hence get the desired result. ∎

**Lemma H.3** *For any $\varepsilon \in (0, 1)$, when $p \geqslant C \theta^{-1} \varepsilon^{-2} \log^3 n$, we have*

$$
\left\| \frac{1}{\theta n p} \sum_{i=1}^{p} \boldsymbol{C}_{\boldsymbol{y}_i}^\top \boldsymbol{C}_{\boldsymbol{y}_i} \right\| \leqslant (1 + \varepsilon) \|\boldsymbol{C}_{\boldsymbol{a}}\|^2
$$

$$
\left\| \left( \frac{1}{\theta n p} \sum_{i=1}^{p} \boldsymbol{C}_{\boldsymbol{y}_i}^\top \boldsymbol{C}_{\boldsymbol{y}_i} \right)^{-1/2} - \left(\boldsymbol{C}_{\boldsymbol{a}}^\top \boldsymbol{C}_{\boldsymbol{a}}\right)^{-1/2} \right\| \leqslant \frac{4\kappa^2 \varepsilon}{\sigma_{\min}^2(\boldsymbol{C}_{\boldsymbol{a}})}
$$

*holds with probability at least $1 - p^{-c_1 n\theta} - n^{-c_2}$. Here, $\kappa$ is the condition number of $\boldsymbol{C}_{\boldsymbol{a}}$, and $\sigma_{\min}(\boldsymbol{C}_{\boldsymbol{a}})$ is the smallest singular value of $\boldsymbol{C}_{\boldsymbol{a}}$.*

**Proof** For any $\varepsilon \in (0,1)$, from Lemma H.2, when $p \geqslant C\theta^{-1}\varepsilon^{-2}\log^3 n$ we know that the event

$$\mathcal{E}(\varepsilon) \doteq \left\{ \left\| \frac{1}{\theta np} \sum_{i=1}^{p} \boldsymbol{C}_{\boldsymbol{x}_i}^{\top} \boldsymbol{C}_{\boldsymbol{x}_i} - \boldsymbol{I} \right\| \leqslant \varepsilon \right\}$$

holds with probability at least $1 - p^{-c_1 n\theta} - n^{-c_2}$. Conditioned on the event $\mathcal{E}(\varepsilon)$, let us denote

$$\boldsymbol{A} = \boldsymbol{C}_{\boldsymbol{a}}^{\top} \boldsymbol{C}_{\boldsymbol{a}} > \boldsymbol{0},$$

and let $\sigma_{\max}(\boldsymbol{A})$, $\sigma_{\min}(\boldsymbol{A})$ be the largest and smallest singular values of $\boldsymbol{A}$, respectively. Then we observe,

$$\frac{1}{\theta np} \sum_{i=1}^{p} \boldsymbol{C}_{\boldsymbol{y}_i}^{\top} \boldsymbol{C}_{\boldsymbol{y}_i} = \boldsymbol{C}_{\boldsymbol{a}}^{\top} \boldsymbol{C}_{\boldsymbol{a}} + \underbrace{\boldsymbol{C}_{\boldsymbol{a}}^{\top} \left[ \frac{1}{\theta np} \sum_{i=1}^{p} \boldsymbol{C}_{\boldsymbol{x}_i}^{\top} \boldsymbol{C}_{\boldsymbol{x}_i} - \boldsymbol{I} \right] \boldsymbol{C}_{\boldsymbol{a}}}_{\boldsymbol{\Delta}},$$

$$= \boldsymbol{A} + \boldsymbol{\Delta}, \qquad \|\boldsymbol{\Delta}\| \leqslant \varepsilon \cdot \sigma_{\max}(\boldsymbol{A}).$$

Therefore, we have

$$\left\| \frac{1}{\theta np} \sum_{i=1}^{p} \boldsymbol{C}_{\boldsymbol{y}_i}^{\top} \boldsymbol{C}_{\boldsymbol{y}_i} \right\| \leqslant \|\boldsymbol{A}\| + \|\boldsymbol{\Delta}\| \leqslant (1+\varepsilon) \|\boldsymbol{C}_{\boldsymbol{a}}\|^2.$$

By Lemma D.12, whenever

$$\|\boldsymbol{\Delta}\| \leqslant \frac{1}{2}\sigma_{\min}(\boldsymbol{A}) \implies \varepsilon \leqslant \frac{1}{2}\frac{\sigma_{\min}(\boldsymbol{A})}{\sigma_{\max}(\boldsymbol{A})} = \frac{1}{2\kappa^2},$$

we know that

$$\left\| \left( \frac{1}{\theta np} \sum_{i=1}^{p} \boldsymbol{C}_{\boldsymbol{y}_i}^{\top} \boldsymbol{C}_{\boldsymbol{y}_i} \right)^{-1/2} - \left( \boldsymbol{C}_{\boldsymbol{a}}^{\top} \boldsymbol{C}_{\boldsymbol{a}} \right)^{-1/2} \right\| = \left\| (\boldsymbol{A} + \boldsymbol{\Delta})^{-1/2} - \boldsymbol{A}^{-1/2} \right\|$$

$$\leqslant \frac{4\|\boldsymbol{\Delta}\|}{\sigma_{\min}^2(\boldsymbol{A})} \leqslant \frac{4\varepsilon\sigma_{\max}(\boldsymbol{A})}{\sigma_{\min}^2(\boldsymbol{A})} = \frac{4\kappa^2\varepsilon}{\sigma_{\min}^2(\boldsymbol{C}_{\boldsymbol{a}})}.$$

∎

**Lemma H.4** *Let $\theta \in (1/n, 1/3)$, and given a $\delta \in (0,1)$. Whenever*

$$p \geqslant C\frac{\kappa^8}{\theta\delta^2\sigma_{\min}^2(\boldsymbol{C}_{\boldsymbol{a}})}\log^3 n,$$

*we have*

$$\|\boldsymbol{R}\boldsymbol{Q}^{-1} - \boldsymbol{I}\| \leqslant \delta, \qquad \|\boldsymbol{R}\boldsymbol{Q}^{-1}\| \leqslant 1+\delta,$$
$$\left\| (\boldsymbol{R}\boldsymbol{Q}^{-1})^{-1} - \boldsymbol{I} \right\| \leqslant 2\delta, \qquad \left\| (\boldsymbol{R}\boldsymbol{Q}^{-1})^{-1} \right\| \leqslant 1+2\delta$$

*hold with probability at least $1 - p^{-c_1 n\theta} - n^{-c_2}$.*

**Proof** First, by Lemma H.3, for a given $\varepsilon \in (0,1)$, when $p \geqslant C_1\theta^{-1}\varepsilon^{-2}\log^3 n$, we have

$$\|\boldsymbol{R}\boldsymbol{Q}^{-1} - \boldsymbol{I}\| = \left\| \boldsymbol{I} - \boldsymbol{C}_{\boldsymbol{a}} \left( \frac{1}{\sqrt{\theta np}} \sum_{i=1}^{p} \boldsymbol{C}_{\boldsymbol{y}_i}^{\top} \boldsymbol{C}_{\boldsymbol{y}_i} \right)^{-1/2} (\boldsymbol{C}_{\boldsymbol{a}}^{\top}\boldsymbol{C}_{\boldsymbol{a}})^{1/2} \boldsymbol{C}_{\boldsymbol{a}}^{-1} \right\|$$

$$\leqslant \kappa \cdot \|\boldsymbol{C}_{\boldsymbol{a}}\| \cdot \left\| \left( \frac{1}{\theta np} \sum_{i=1}^{p} \boldsymbol{C}_{\boldsymbol{y}_i}^{\top} \boldsymbol{C}_{\boldsymbol{y}_i} \right)^{-1/2} - (\boldsymbol{C}_{\boldsymbol{a}}^{\top}\boldsymbol{C}_{\boldsymbol{a}})^{-1/2} \right\|$$

$$\leqslant \kappa \|\boldsymbol{C}_{\boldsymbol{a}}\| \frac{4\kappa^2\varepsilon}{\sigma_{\min}^2(\boldsymbol{C}_{\boldsymbol{a}})} \leqslant \frac{4\kappa^4\varepsilon}{\sigma_{\min}(\boldsymbol{C}_{\boldsymbol{a}})},$$

Table 2: Gradient for each different loss function

| Loss function | $\nabla\varphi(\boldsymbol{q})$ for 1D problem (73) | $\nabla\varphi(\boldsymbol{Z})$ for 2D problem[13] (74) |
|---|---|---|
| $\ell^1$-loss | $\frac{1}{np}\sum_{i=1}^{p}\breve{\overline{\boldsymbol{y}}}_i \circledast \operatorname{sign}\left(\overline{\boldsymbol{y}}_i \circledast \boldsymbol{q}\right)$ | $\frac{1}{n^2p}\sum_{i=1}^{p}\breve{\overline{\boldsymbol{Y}}}_i \circledast \operatorname{sign}\left(\overline{\boldsymbol{Y}}_i \circledast \boldsymbol{Z}\right)$ |
| Huber-loss | $\frac{1}{np}\sum_{i=1}^{p}\breve{\overline{\boldsymbol{y}}}_i \circledast h'_\mu\left(\overline{\boldsymbol{y}}_i \circledast \boldsymbol{q}\right)$ | $\frac{1}{n^2p}\sum_{i=1}^{p}\breve{\overline{\boldsymbol{Y}}}_i \circledast h'_\mu\left(\overline{\boldsymbol{Y}}_i \circledast \boldsymbol{Z}\right)$ |
| $\ell^4$-loss | $-\frac{1}{np}\sum_{i=1}^{p}\breve{\overline{\boldsymbol{y}}}_i \circledast \left(\overline{\boldsymbol{y}}_i \circledast \boldsymbol{q}\right)^{\odot 3}$ | $-\frac{1}{n^2p}\sum_{i=1}^{p}\breve{\overline{\boldsymbol{Y}}}_i \circledast \left(\overline{\boldsymbol{Y}}_i \circledast \boldsymbol{Z}\right)^{\odot 3}$ |

and

$$\left\|\boldsymbol{R}\boldsymbol{Q}^{-1}\right\| \;\leqslant\; 1 + \left\|\boldsymbol{I} - \boldsymbol{R}\boldsymbol{Q}^{-1}\right\| \;\leqslant\; 1 + \frac{4\kappa^4\varepsilon}{\sigma_{\min}(\boldsymbol{C}_{\boldsymbol{a}})}$$

hold with probability at least $1 - p^{-c_1 n\theta} - n^{-c_2}$. Similarly, by Lemma H.3,

$$
\begin{aligned}
\left\|\boldsymbol{I} - \left(\boldsymbol{R}\boldsymbol{Q}^{-1}\right)^{-1}\right\| &= \left\|\boldsymbol{I} - \boldsymbol{C}_{\boldsymbol{a}}\left(\boldsymbol{C}_{\boldsymbol{a}}^\top\boldsymbol{C}_{\boldsymbol{a}}\right)^{-1/2}\left(\frac{1}{\sqrt{\theta np}}\sum_{i=1}^{p}\boldsymbol{C}_{\boldsymbol{y}_i}^\top\boldsymbol{C}_{\boldsymbol{y}_i}\right)^{1/2}\boldsymbol{C}_{\boldsymbol{a}}^{-1}\right\| \\
&\leqslant \kappa \cdot \left\|\frac{1}{\theta np}\sum_{i=1}^{p}\boldsymbol{C}_{\boldsymbol{y}_i}^\top\boldsymbol{C}_{\boldsymbol{y}_i}\right\|^{1/2} \cdot \left\|\left(\frac{1}{\theta np}\sum_{i=1}^{p}\boldsymbol{C}_{\boldsymbol{y}_i}^\top\boldsymbol{C}_{\boldsymbol{y}_i}\right)^{-1/2} - \left(\boldsymbol{C}_{\boldsymbol{a}}^\top\boldsymbol{C}_{\boldsymbol{a}}\right)^{-1/2}\right\| \\
&\leqslant \kappa \cdot \frac{4\kappa^2\varepsilon}{\sigma_{\min}^2(\boldsymbol{C}_{\boldsymbol{a}})} \cdot (1+\varepsilon)^{1/2}\left\|\boldsymbol{C}_{\boldsymbol{a}}\right\| \;\leqslant\; \frac{8\kappa^4\varepsilon}{\sigma_{\min}(\boldsymbol{C}_{\boldsymbol{a}})},
\end{aligned}
$$

and

$$\left\|\left(\boldsymbol{R}\boldsymbol{Q}^{-1}\right)^{-1}\right\| \;\leqslant\; 1 + \left\|\boldsymbol{I} - \left(\boldsymbol{R}\boldsymbol{Q}^{-1}\right)^{-1}\right\| \;\leqslant\; 1 + \frac{8\kappa^4\varepsilon}{\sigma_{\min}(\boldsymbol{C}_{\boldsymbol{a}})}$$

Thus, replace $\delta = \frac{4\kappa^4\varepsilon}{\sigma_{\min}(\boldsymbol{C}_{\boldsymbol{a}})}$, we obtain the desired result. ∎

# I  Algorithms and Implementation Details

In this section, we provide detailed descriptions of our algorithms. First, we introduce the details Riemannian (sub)gradient descent method for 1D problem. Second, we discuss about subgradient methods for solving the LP rounding problem. Finally, we provide more details about how to solve problems in 2D.

For the purpose of implementation efficiency, we describe the problem and algorithms based on circulant convolution, which is slightly different from the main sections. Because our gradient descent method works for any sparse promoting loss function (other than Huber loss), in the following we describe the problem and the algorithm in a more general form. However, it should be noted that our analysis is only specified for Huber loss in the following sections.

## I.1  Riemannian (sub)gradient descent methods

Here, we consider (sub)gradient descent for optimizing a more general problem

$$\min_{\boldsymbol{q}} \;\; \varphi(\boldsymbol{q}) := \frac{1}{np}\sum_{i=1}^{p}\psi(\boldsymbol{C}_{\boldsymbol{y}_i}\boldsymbol{P}\boldsymbol{q}), \quad \text{s.t. } \|\boldsymbol{q}\| = 1,$$

**Algorithm 1** Riemannian (sub)gradient descent algorithm

---

**Input:** observation $\{\boldsymbol{y}_i\}_{i=1}^m$
**Output:** the vector $\boldsymbol{q}_\star$,

Precondition the data by $\overline{\boldsymbol{y}}_i = \boldsymbol{y}_i \circledast \boldsymbol{v}$, with $\boldsymbol{v} = \left( \frac{1}{\theta np} \sum_{i=1}^p |\boldsymbol{y}_i|^{\odot 2} \right)^{\odot -1/2}$.

Initialize the iterate $\boldsymbol{q}^{(0)}$ and stepsize $\tau^{(0)}$.
**while** not converged **do**
    Update the iterate by

$$\boldsymbol{q}^{(k+1)} = \mathcal{P}_{\mathbb{S}^{n-1}}\left( \boldsymbol{q}^{(k)} - \tau^{(k)} \operatorname{grad} \varphi(\boldsymbol{q}^{(k)}) \right).$$

    Choose a new stepsize $\tau^{(k+1)}$, and set $k \leftarrow k+1$.
**end while**

---

911    where $\psi(\boldsymbol{z})$ can be $\ell^1$-loss ($\psi(\boldsymbol{z}) = \|\boldsymbol{z}\|_1$), Huber-loss ($\psi(\boldsymbol{z}) = H_\mu(\boldsymbol{z})$), and $\ell^4$-loss ($\psi(\boldsymbol{z}) =$
912    $-\|\boldsymbol{z}\|_4^4$). The preconditioning matrix $\boldsymbol{P}$ can be written as

$$\boldsymbol{P} = \boldsymbol{C}_{\boldsymbol{v}}, \quad \boldsymbol{v} = \boldsymbol{F}^{-1}\left( \left( \frac{1}{\theta np} \sum_{i=1}^p |\widehat{\boldsymbol{y}}_i|^{\odot 2} \right)^{\odot -1/2} \right),$$

913    where $\widehat{\boldsymbol{y}}_i = \boldsymbol{F}\boldsymbol{y}_i$, so that

$$\boldsymbol{C}_{\boldsymbol{y}_i}\boldsymbol{P} = \boldsymbol{C}_{\boldsymbol{y}_i}\boldsymbol{C}_{\boldsymbol{v}} = \boldsymbol{C}_{\boldsymbol{y}_i \circledast \boldsymbol{v}} = \boldsymbol{C}_{\overline{\boldsymbol{y}}_i}, \quad \overline{\boldsymbol{y}}_i = \boldsymbol{y}_i \circledast \boldsymbol{v}.$$

914    Therefore, our problem can be rewritten as

$$\min_{\boldsymbol{q}} \ \varphi(\boldsymbol{q}) := \frac{1}{np} \sum_{i=1}^p \psi(\overline{\boldsymbol{y}}_i \circledast \boldsymbol{q}), \quad \text{s.t. } \|\boldsymbol{q}\| = 1. \tag{73}$$

915    Starting from an initialization, we solve the problem via Riemannian (sub)gradient descent,

$$\boldsymbol{q}^{(k+1)} = \mathcal{P}_{\mathbb{S}^{n-1}}\left( \boldsymbol{q}^{(k)} - \tau^{(k)} \cdot \operatorname{grad} \varphi(\boldsymbol{q}^{(k)}) \right),$$

916    where $\tau^{(k)}$ is the stepsize, and the Riemannian (sub)gradient is

$$\operatorname{grad} \varphi(\boldsymbol{q}) = \mathcal{P}_{\boldsymbol{q}^\perp} \nabla \varphi(\boldsymbol{q}),$$

917    which is defined on the *tangent space*[14] $T_{\boldsymbol{q}}\mathbb{S}^{n-1}$ at a point $\boldsymbol{q} \in \mathbb{S}^{n-1}$. Table 2 lists the calcu-
918    lation of (sub)gradients $\nabla \varphi(\boldsymbol{q})$ for different loss functions. For each iteration, the projection
919    operator $\mathcal{P}_{\mathbb{S}^{n-1}}(\boldsymbol{z}) = \boldsymbol{z}/\|\boldsymbol{z}\|$ retracts the iterate back to the sphere. Let $\odot$ denotes entry-wise
920    power/multiplication, the overall algorithm is summarized in Algorithm 1.

921    **Initialization.** In our theory, we showed that starting from a random initialization drawn uniformly
922    over the sphere,

$$\boldsymbol{q}^{(0)} = \boldsymbol{d}, \quad \boldsymbol{d} \sim \mathcal{U}(\mathbb{S}^{n-1}),$$

923    for Huber-loss, Riemannian gradient descent method provably recovers the target solution. On the
924    other hand, we could also cook up a data-driven initialization by choosing a row of $\boldsymbol{C}_{\overline{\boldsymbol{y}}_i}$,

$$\boldsymbol{q}^{(0)} = \mathcal{P}_{\mathbb{S}^{n-1}}\left( \boldsymbol{C}_{\overline{\boldsymbol{y}}_i}^\top \boldsymbol{e}_j \right)$$

925    for some randomly chosen $1 \leqslant i \leqslant p$ and $1 \leqslant j \leqslant n$. By observing

$$\boldsymbol{C}_{\overline{\boldsymbol{y}}_i} \approx \boldsymbol{C}_{\boldsymbol{x}_i}\boldsymbol{C}_{\boldsymbol{a}}\left( \boldsymbol{C}_{\boldsymbol{a}}^\top \boldsymbol{C}_{\boldsymbol{a}} \right)^{-1/2}, \quad \boldsymbol{q}^{(0)} \approx \mathcal{P}_{\mathbb{S}^{n-1}}\left( \left( \boldsymbol{C}_{\boldsymbol{a}}^\top \boldsymbol{C}_{\boldsymbol{a}} \right)^{-1/2} \boldsymbol{C}_{\boldsymbol{a}}^\top s_j \left[ \breve{\boldsymbol{x}}_i \right] \right),$$

926    we have

$$\boldsymbol{C}_{\overline{\boldsymbol{y}}_j} \boldsymbol{q}^{(0)} \approx \alpha \boldsymbol{C}_{\boldsymbol{x}_i}\boldsymbol{C}_{\boldsymbol{a}}(\boldsymbol{C}_{\boldsymbol{a}}^\top \boldsymbol{C}_{\boldsymbol{a}})^{-1}\boldsymbol{C}_{\boldsymbol{a}}^\top s_\ell \left[ \breve{\boldsymbol{x}}_i \right] = \alpha \boldsymbol{C}_{\boldsymbol{x}_j} s_\ell \left[ \breve{\boldsymbol{x}}_i \right].$$

927    This suggests that our particular initialization $\boldsymbol{q}^{(0)}$ is acting like $s_\ell \left[ \breve{\boldsymbol{x}}_i \right]$ in the rotated domain. It is
928    sparse and possesses several large spiky entries more biased towards the target solutions. Empirically,
929    we find this data-driven initialization often works better than random initializations.

**Choice of stepsizes.** For Huber and $\ell^4$ losses, we can choose a fixed stepsize $\tau^{(k)}$ for all iterates to guarantee linear convergence. For subgradient descent of $\ell^1$-loss, it often achieves linear convergence when we choose a geometrically decreasing sequence of stepsize $\tau^{(k)}$ [48]. Empirically, we find that the algorithm converges much faster when Riemannian linesearch is deployed (see Algorithm 2).

---

**Algorithm 2** Riemannian linesearch for stepsize $\tau$

---

**Input:** $\quad \boldsymbol{a}, \boldsymbol{x}, \tau_0, \eta \in (0.5, 1), \beta \in (0, 1),$
**Output:** $\quad \tau, \mathcal{R}_{\boldsymbol{a}}^{\mathcal{M}}\left(-\tau \boldsymbol{P}_{T_{\mathcal{M}}} \nabla \psi_{\boldsymbol{x}}(\boldsymbol{a})\right)$
&emsp; Initialize $\tau \leftarrow \tau_0,$
&emsp; Set $\widetilde{\boldsymbol{q}} = \mathcal{P}_{\mathbb{S}^{n-1}}\left(\boldsymbol{q} - \tau \operatorname{grad} \varphi(\boldsymbol{q})\right)$
&emsp; **while** $\varphi(\widetilde{\boldsymbol{q}}) \geqslant \varphi(\boldsymbol{q}) - \tau \cdot \eta \cdot \|\operatorname{grad} \varphi(\boldsymbol{q})\|^2$ **do**
&emsp;&emsp; $\tau \leftarrow \beta \tau,$
&emsp;&emsp; Update $\widetilde{\boldsymbol{q}} = \mathcal{P}_{\mathbb{S}^{n-1}}\left(\boldsymbol{q} - \tau \operatorname{grad} \varphi(\boldsymbol{q})\right).$
&emsp; **end while**

---

## I.2 LP rounding

Due to preconditioning or smoothing effects of our choice of loss functions, the Riemannian (sub)gradient descent methods can only produce an approximate solution. To obtain the exact solution, we use the solution $\boldsymbol{r} = \boldsymbol{q}_\star$ produced by gradient methods as a warm start, and solve another phase-two LP rounding problem,

$$\min_{\boldsymbol{q}} \ \zeta(\boldsymbol{q}) := \frac{1}{np} \sum_{i=1}^{p} \|\overline{\boldsymbol{y}}_i \circledast \boldsymbol{q}\|_1 \quad \text{s.t.} \quad \langle \boldsymbol{r}, \boldsymbol{q} \rangle = 1.$$

Since the feasible set $\langle \boldsymbol{r}, \boldsymbol{q} \rangle = 1$ is essentially the tangent space of the sphere $\mathbb{S}^{n-1}$ at $\boldsymbol{q}_\star$, whenever $\boldsymbol{q}_\star$ is close enough to one of the target solutions, one should expect that the optimizer $\boldsymbol{q}_r$ of LP rounding exactly recovers the inverse of the kernel $\boldsymbol{a}$ up to a scaled-shift. To address this computational issue, we utilize a *projected subgradient method* for solving the LP rounding problem. Namely, we take

$$\begin{aligned} \boldsymbol{q}^{(k+1)} &= \boldsymbol{r} + \left(\boldsymbol{I} - \boldsymbol{r}\boldsymbol{r}^\top\right)\left(\boldsymbol{q}^{(k)} - \tau^{(k)} \boldsymbol{g}^{(k)}\right) \\ &= \boldsymbol{q}^{(k)} - \tau^{(k)} \mathcal{P}_{\boldsymbol{r}^\perp} \boldsymbol{g}^{(k)}, \end{aligned}$$

where $\boldsymbol{g}^{(k)}$ is the subgradient at $\boldsymbol{q}^{(k)}$ with

$$\boldsymbol{g}^{(k)} = \frac{1}{np} \sum_{i=1}^{p} \check{\overline{\boldsymbol{y}}}_i \circledast \operatorname{sign}\left(\overline{\boldsymbol{y}}_i \circledast \boldsymbol{q}^{(k)}\right).$$

By choosing a geometrically shrinking stepsizes

$$\tau^{(k+1)} = \beta \tau^{(k)}, \quad \beta \in (0, 1).$$

we show that the subgradient descent linearly converges to the target solution. The overall method is summarized in Algorithm 3.

## I.3 Solving problems in 2D

Finally, we briefly discuss about technical details about solving the MCS-BD problem in 2D, which appears broadly in imaging applications such as image deblurring [13–15] and microscopy imaging [3, 16, 17].

**Problem formulation.** Given the measurements

$$\boldsymbol{Y}_i = \boldsymbol{A} \boxasterisk \boldsymbol{X}_i, \quad 1 \leqslant i \leqslant p,$$

where $\boxasterisk$ denotes 2D convolution, $\boldsymbol{A} \in \mathbb{R}^{n \times n}$ is a 2D kernel, and $\boldsymbol{X}_i \in \mathbb{R}^{n \times n}$ is a sparse activation map, we want to recover $\boldsymbol{A}$ and $\{\boldsymbol{X}_i\}_{i=1}^{p}$ simultaneously. We first precondition the data via

$$\overline{\boldsymbol{Y}}_i = \boldsymbol{Y}_i \boxasterisk \boldsymbol{V}, \quad \boldsymbol{V} = \mathcal{F}^{-1}\left(\left(\frac{1}{\theta n^2 p} \sum_{i=1}^{p} |\mathcal{F}(\boldsymbol{Y}_i)|^{\odot 2}\right)^{\odot -1/2}\right),$$

---

**Algorithm 3** Projected subgradient method for solving the LP rounding problem

---

**Input:**   observation $\{\boldsymbol{y}_i\}_{i=1}^{m}$, vector $\boldsymbol{r}$, stepsize $\tau_0$, and $\beta \in (0,1)$.
**Output:**   the solution $\boldsymbol{q}_\star$,

   Precondition the data by $\overline{\boldsymbol{y}}_i = \boldsymbol{y}_i \circledast \boldsymbol{v}$, with $\boldsymbol{v} = \left( \frac{1}{\theta n p} \sum_{i=1}^{p} |\boldsymbol{y}_i|^{\odot 2} \right)^{\odot -1/2}$.

   Initialize $\boldsymbol{q}^{(0)} = \boldsymbol{r}$, $\tau^{(0)} = \tau_0$
   **while** not converged **do**
      Update the iterate

$$\boldsymbol{q}^{(k+1)} \;=\; \boldsymbol{q}^{(k)} - \tau^{(k)} \mathcal{P}_{\boldsymbol{r}^\perp} \boldsymbol{g}^{(k)}.$$

   Set $\tau^{(k+1)} = \beta \tau^{(k)}$, and $k \leftarrow k+1$.
   **end while**

---

where $\mathcal{F}(\cdot)$ denote the 2D DFT operator. By using the preconditioned data, we solve the following optimization problem

$$\min_{\boldsymbol{Z}} \; \varphi(\boldsymbol{Z}) \;:=\; \frac{1}{n^2 p} \sum_{i=1}^{p} \psi(\overline{\boldsymbol{Y}}_i \circledast \boldsymbol{Z}), \quad \text{s.t. } \|\boldsymbol{Z}\|_F = 1, \tag{74}$$

where $\varphi(\cdot)$ is the loss function (e.g., $\ell^1$, Huber, $\ell^4$-loss), and $\|\cdot\|_F$ denotes the Frobenius norm. If the problem (74) can be solved to the target solution $\boldsymbol{Z}_\star$, then we can recover the kernel and the sparse activation map up to a signed-shift by

$$\boldsymbol{A}_\star \;=\; \mathcal{F}^{-1}\left( \mathcal{F}\left(\boldsymbol{V} \circledast \boldsymbol{Z}_\star\right)^{\odot -1} \right), \qquad \boldsymbol{X}_i^\star \;=\; (\boldsymbol{Y}_i \circledast \boldsymbol{V}) \circledast \boldsymbol{Z}_\star, \; 1 \leqslant i \leqslant p.$$

**Riemannian (sub)gradient descent.**   Similar to the 1D case, we can optimize the problem (74) via Riemannian (sub)gradient descent,

$$\boldsymbol{Z}^{(k+1)} \;=\; \mathcal{P}_F\left( \boldsymbol{Z}^{(k)} - \tau^{(k)} \cdot \operatorname{grad} \varphi(\boldsymbol{Z}^{(k)}) \right),$$

where the Riemannian (sub)gradient

$$\operatorname{grad} \varphi(\boldsymbol{Z}) \;=\; \mathcal{P}_{\boldsymbol{Z}^\perp} \nabla \varphi(\boldsymbol{Z}).$$

The gradient $\nabla \varphi(\boldsymbol{Z})$ for different loss functions are recorded in Table 2. For any $\boldsymbol{W} \in \mathbb{R}^{n \times n}$, the normalization operator $\mathcal{P}_F(\cdot)$ and projection operator $\mathcal{P}_{\boldsymbol{Z}^\perp}(\cdot)$ are defined as

$$\mathcal{P}_F(\boldsymbol{W}) \;:=\; \boldsymbol{W}/\|\boldsymbol{W}\|_F, \quad \mathcal{P}_{\boldsymbol{Z}^\perp}(\boldsymbol{W}) \;:=\; \boldsymbol{W} - \|\boldsymbol{Z}\|_F^{-2} \langle \boldsymbol{Z}, \boldsymbol{W} \rangle \boldsymbol{Z}.$$

The initialization and stepsize $\tau^{(k)}$ can be chosen similarly as the 1D case.

**LP rounding.**   Similar to 1D case, we solve a phase-two linear program to obtain exact solution. By using the solution $\boldsymbol{Z}_\star$ produced by Riemannian gradient descent as a warm start $\boldsymbol{U} = \boldsymbol{Z}_\star$, we solve

$$\min_{\boldsymbol{Z}} \; \frac{1}{n^2 p} \sum_{i=1}^{p} \left\| \overline{\boldsymbol{Y}}_i \circledast \boldsymbol{Z} \right\|_1, \quad \text{s.t. } \langle \boldsymbol{U}, \boldsymbol{Z} \rangle = 1.$$

We optimize the LP rounding problem via subgradient descent,

$$\boldsymbol{Z}^{(k+1)} = \boldsymbol{Z}^{(k)} - \tau^{(k)} \mathcal{P}_{\boldsymbol{U}^\perp} \boldsymbol{G}^{(k)},$$

where we choose a geometrically decreasing stepsize $\tau^{(k)}$ and set the subgradient

$$\boldsymbol{G}^{(k)} \;=\; \frac{1}{n^2 p} \sum_{i=1}^{p} \breve{\overline{\boldsymbol{Y}}}_i \circledast \operatorname{sign}\left( \overline{\boldsymbol{Y}}_i \circledast \boldsymbol{Z}^{(k)} \right).$$

## Footnotes

[13]Here, for 2D problem, $\breve{\boldsymbol{Z}}$ denotes a flip operator that flips a matrix $\boldsymbol{Z} \in \mathbb{R}^{n_1 \times n_2}$ both vertically and horizontally, i.e., $\breve{\boldsymbol{Z}}_{i,j} = \boldsymbol{Z}_{n_1-i+1,n_2-j+1}$.

[14] We refer the readers to Chapter 3 of [47] for more details.