[Reviews · NeurIPS 2019]

Reviewer 1



The paper is very clearly written, the litterature review and introduction are adequate the derivations presented in the main paper are sound and the experiments are insightful and relevant. I have the following comments and remarks: - The proposed use of a Huber loss is in fact only there to simplify the theoretical analysis compared to using a classical L1 loss. The current presentation tends to suggest that the Huber loss may bring some specific algorithmic advantage, but it is not the case since the L1 norm (which is the limiting case of Huber as \mu goes to 0) always performs better or equivalently to Huber experimentally. This fact could be made more explicit by the authors in the introduction and problem formulation. A natural question is then: what would happen when using L1 minimization without LP rounding? Finally, it would be nice to discuss the relationship between the proposed approach and classical cross-relation methods penalized by an L1 term (e.g., Y Lin, J Chen, Y Kim, DD Lee (Neurips 2008), Benichoux, Vincent Gribonval (2013)). - Line 120: "the nonconvex relaxation (5).." : the authors probably mean (4). However, while it is clear that solving (3) lead to exact recovery using the formula under line 121, it is less clear that it is also true for solving the convex relaxation (4). This statement should be justified. - line 157: the definition below is not strictly speaking a partition, for any \xi>0. - Below line 157: the definition is not valid for ||q_{-1}||_\infty=0, i.e., for basis vectors: the authors should explicitely mention whether or not they are part of the set. - Line 160: "..they are closer to e_i.." : in what norm? - The remark on the top of page 6 about the assumption that RQ^{-1} ~ I is not mathematically rigorous and makes one wonder how valid this approximation really is and how it affects all the propositions, which are only stated within this regime. More details would be welcome. - Proposition 3.4. : I guess the assumptions \epsilon>1/5log(n) is needed for the proposition to hold. -Footnote (11): the authors state that they use a Riemannian subgradient method similar to (15) for the L1 loss, but it is not clear how this can be done. -Some implementation details are missing for the experiments, e.g., what values are chosen for \tau, \tau^(k) and \beta. -The paper lacks a short conclusion outlining perspectives and future research directions. - Unfortunately, I could not thoroughly check the 33 pages of involved mathematical proof in the supplementary material. I wonder whether it is reasonnable to expect NeurIPS reviewers to thoroughly check such large proofs, given that many reviewers have many other papers to review in a relatively short amount of time (in contrast, reviewing a long journal article may take several months). Typos: - L99: similar -> similarly - footnote 6: of the standard Huber function - L105: a transpose sign is missing, i.e., C_a^TC_a - L113: amendable for -> amenable to - L117: In contrast, the sphere is ... - L130: by using a random initialization, provably recovers... - L133: only requires the kernel a to be invertible - L150 the problem (5) -> the problem (4) - L197: in stating -> to state - Eq. (14): u should be a q instead - Footnote 9: an equivalent problem to (14) - L214: in general subgradient method -> a general subgradient method - L217: we prove that \Zeta - L218: in the sense that - L219: we use the matrix-vector form of convolutions - L240: producing -> produces - L242: For fix n=500 and p=50 - L248: we repeat the simulation 15 times. - There are additional misuses of articles throughout the paper, please proofread.

Reviewer 2



Originality: This work gives a thorough overview of the state of the art for the problem at hand. It combines the technique of optimization of Riemannian manifolds together with the Huber loss. Quality: The quality of the paper is high. Content is well organized and all the theoretical claims contain a proof provided in the supplementary material. The reviewer was not able to evaluate the correctness of all of the proofs. If the source code was provided, it would have been easier to get a sense of the performance of the proposed method. Clarity: Narrative is clear. Content is organized well. Significance: The solved problem is of great importance. Sparse blind deconvolution is still an open research question. Most the the papers in the domain still balance the performance trade-offs and costs in the terms of number of required samples.

Reviewer 3



Overall the presented result is timely and novel by providing a significant improvement over existing works. I have major concerns with the presentation and discussion on main results and missing references. 1. First of all, it is really important to interpret the meaning of the main result in the context of the problem. The sample complexity is given in terms of the number of channels p, which scales as a polynomial in the length of signal n. In practical applications, we do not expect a very large number for p. Is it really necessary to have p increasing in n? If one recalls classical results on multichannel blind deconvolution with FIR models, two channels suffice. How is the sparsity case fundamentally different from the FIR case? Although this question also applies to previous work, I think this fundamental question is still legitimate and worth a good discussion. Even in Figures 2 and 3, p is very small number compared to n. This is a bit inconsistent with theory. 2. The presentation of the algorithm needs some clarification. The matrix Q is a function of unknown a and hence is not available. In (15), does one need to know Q to perform the last step? I guess it is just for the sake of the analysis. This requires a clarification. 3. In the simulation, the author(s) could have considered illustrating how the reconstruction improves as one applies random initialization, RGD, and LP rounding in a progressive way. This can justify particularly the value of the last step. 4. There are several places where the statements are not clear enough. - eq (2) is hard to parse. - The decomposition of a circulant matrix requires a scale factor when F is unitary. Since the authors used the adjoint in the decomposition, it looks like they considered the unitary F. - I did not follow the sentence 92. How does the nonsmoothness make the analysis difficult? - The comparison in Figure 1 can be explained with more details. How can one see that the landscape is better with Huber loss? - line 132: why is it trivial when theta is less than 1/n? - line 157: The union does not cover the entire space. It does if xi converges to 0. Can it still be a valid partition? - The introduction of (10) might be just for the sake of the proofs. Results are directly stated on (9). The authors might want to reorganize the contents to improve the readability. 5. The following paper considered a similar problem and it will be useful to compare the result to it. Augustin Cosse, A note on the blind deconvolution of multiple sparse signals from unknown subspaces

[Author Response · NeurIPS 2019]

**Reviewer 1.**   We appreciate your invaluable comments and questions. We address your concerns below.

R1.1   **Huber loss vs. $\ell^1$-loss.** Our choice of Huber loss rather than $\ell^1$-loss is to simplify theoretical analysis.
Undoubtedly, $\ell^1$-loss is a more natural sparsity promoting function and performs better than Huber. We will refine our
statement in the revision to make this fact more clear and transparent. When $\ell^1$-loss is utilized, the experiments tend to
suggest that the underlying kernel and signals can be exactly recovered even without LP rounding; see the figure below
in which rounding is not applied to $\ell^1$-loss. However, on theory side, the subgradient of $\ell^1$-loss is *non-Lipschitz* which
introduces tremendous difficulty in controlling suprema of random process and perturbation analysis for preconditioning
for our problem. We leave analyzing $\ell^1$-loss for future research which we will discuss in the final draft.

R1.2   **Relation to cross-relation methods.** We apologize for omitting this literature (e.g., G. Xu et al. TSP'95, Y.
Lin et al. NeurIPS'07, K. Lee et al. SIIMS'18), which consider similar problems that we will include in revision. In
comparison, *(i)* Lin et al. proposed an $\ell^1$-regularized least-squares method based on convex relaxation. The convex
method could suffer similar sparsity limitation as [19], and it limits to 2 channels without theoretical guarantees; *(ii)*
Lee et al. proposed an eigen approach for FIR signal model. Their provable efficient method considered short signal $\mathbf{x}_i$
which lives in given random subspaces, thus it cannot directly handle our case with random sparse nonzero support.

R1.3   **Clarification of the set $\mathcal{S}_\xi^{i\pm}$.** Our original description that the union of $\mathcal{S}_\xi^{i\pm}$ gives a full partition of $\mathbb{S}^{n-1}$ is
inaccurate. Indeed, we consider these sets because their union excludes all saddle points, but covers a large portion of
measure over the sphere for small $\xi$. For each individual set, we will make the inclusion $\mathbf{q} = \mathbf{e}_i$ clear and well-defined.

R1.4   **Closeness of $\mathbf{R}\mathbf{Q}^{-1}$ and I.** We will include the precise approximation error from Lemma H.4 of supplementary.

R1.5   **Algorithmic implementation details.** Experimentally we use linesearch for both the stepsizes in RGD and LP
rounding for optimizing all losses. In revision, we will expand Line 230 with more details and release code.

In revision, we will add a conclusion and address other minor issues without detailed explanation due to space limit.

**Reviewer 2.**   We sincerely thank you for your appreciation of our work. For reproducible research, we will release
well documented code of this work on Github, and correspondingly provide a link in the revised draft.

R2.1   **Smoothing parameter $\mu$.** Here, the parameter $\mu$ in Huber introduces a *tradeoff* between sample complexity and
recovery accuracy. As shown in Theorem 3.1, the sample complexity $p$ depends *inversely* on $\mu$: larger $p$ is required
for smaller $\mu$, and vice versa; on the other hand, Figure 2 (or the revised figure below, as suggested) shows smaller $\mu$
produces higher recovery accuracy in Phase 1. We will discuss this around Theorem 3.1 and experiment section.

**Reviewer 3.**   We really appreciate your constructive criticism and valuable feedbacks, that we address as follows.

R3.1   **Sample Complexity.** We agree there is a large sample complexity *gap* between our theory and practice. From
the degree of freedom perspective (e.g., Eric Moulines et al. TSP'95), a constant $p$ is also seemingly enough in our
case. However, as the problem is highly nonconvex with unknown nonzero supports of $\mathbf{x}_i$s, to have provable efficient
methods, we conjecture that paying extra log factors $p \geq \Omega(\text{poly}\log(n))$ is necessary as stated in Line 56, 139 and
251, which is empirically confirmed by Figure 4. This is similar to recent provable efficient method on FIR model (K.
Lee et al. SIIMS'18). On the other hand, we believe our *far from tight* sample complexity $p \geq \Omega(\text{poly}(n))$ is due to the
looseness in our analysis: *(i)* loose control of summations of dependent random variables, and *(ii)* tiny gradient near the
set boundary for concentration. We will discuss this in revision and leave improvement for future work.

R3.2   **Clarification of rounding with unknown rotations.** We do *not* need to know $\mathbf{Q}$ for solving LP rounding.
Footnote 9 and Appendix I provide more details of the *actual* problem form we are solving. The reason we stated LP
rounding in the rotated space as (14) (typo: $\mathbf{u}$ should be $\mathbf{q}$ in (14)) is *only* for the convenience of introducing subsequent
results. Recall the deduction from (4) to (9), we can *reversely* get back the actual form in Footnote 9 by plugging
$\mathbf{q} = \mathbf{Q}\mathbf{q}'$ into (14) (with an abuse of notations of $\mathbf{q}$ and $\mathbf{q}'$), where $\bar{\mathbf{r}}$ is the *actual solution* of optimizing (4) in Phase 1.
Therefore, $\mathbf{Q}$ is *not* needed. We will make this involved narrative more clear in revision.

R3.3   **Simulation in Figure 2.** Following reviewer's suggestion, in the right figure
we show the convergence in a progressive way for optimizing $\ell^4$ and Huber losses,
with the same setup as in Figure 2. We observe that $(i)$ in Phase 1, the reconstruction
errors stagnate for both $\ell^4$ and Huber losses before rounding is applied, and $(ii)$ in
Phase 2, the projected subgradient method for LP rounding converges *linearly* to
a target solution. For the $\ell^1$-loss, it seems that rounding is *not* necessary. Per our
R1.1 to Reviewer 1, analyzing this behavior is the subject of future work.

R3.4   **Other technical issues.** We briefly address other minor technical issues as
follows: *(i)* Equation (2) is to show *intrinsic* shift-scaling symmetry, so that we can
only solve to a shift equivalence; *(ii)* the DFT matrix $\mathbf{F}$ is unnormalized, as shown
on Line 389 of the supplementary; *(iii)* Figure 1 plots the function values over the sphere, where cooler color denotes
smaller values, and vice versa. The target solutions (red dots) stays much closer to global minima for Huber and $\ell^1$
losses than $\ell^4$-loss; *(iv)* the problem becomes trivial when $\theta \leq 1/n$ because $\theta n = 1$ so that $\mathbf{x}_i$ tends to be a $\delta$-function;
*(v)* We will mention the result by Cosse. As it is a convex method following [19], it may suffer similar limitations.

Due to space limit, for other questions we refer Reviewer 3 to our response to Reviewer 1 (e.g., R1.1, R1.2, and R1.3).

[Meta-Review · NeurIPS 2019]

This is the best paper in my docket. It clearly articulates an interesting blind deconvolution problem, describes a Huber loss function with a nice geometric interpretation, and shows how the proposed method leads to significant improvements over the prior SOTA. Not only is the approach novel, but the derivation and analysis have some novel components as well.